# Annual 30-meter Dataset for Glacial Lakes in High Mountain Asia from 2008 to 2017

Fang Chen[1,2,3], Meimei Zhang[1], Huadong Guo[1,2,3], Simon Allen[4,5], Jeffrey S. Kargel[6], Umesh K. Haritashya[7], Cameron S. Watson[8,9]

[1]Key Laboratory of Digital Earth Science, Aerospace Information Research Institute, Chinese Academy of Sciences, No. 9 Dengzhuang South Road, Beijing 100094, China.

[2]State Key Laboratory of Remote Sensing Science, Aerospace Information Research Institute, Chinese Academy of Sciences, No. 9 Dengzhuang South Road, Beijing 100094, China.

[3]Hainan Key Laboratory of Earth Observation, Aerospace Information Research Institute, Chinese Academy of Sciences, Sanya 572029, China.

[4]Department of Geography, University of Zurich, Zurich, 8057, Switzerland.

[5]Institute for Environmental Sciences, University of Geneva, Geneva, 1205, Switzerland.

[6]The Planetary Science Institute, Tucson, Arizona, 85719, USA.

[7]Department of Geology, University of Dayton, Dayton, Ohio, 45469, USA.

[8]Department of Hydrology & Atmospheric Sciences, University of Arizona, Tucson, Arizona, 85721, USA.

[9]COMET, School of Earth and Environment, University of Leeds, Leeds, LS2 9JT, UK.

*Correspondence to*: Meimei Zhang (zhangmm@radi.ac.cn)

**Abstract.** Atmospheric warming is intensifying glacier melting and glacial lake development in High Mountain Asia (HMA), and this could increase glacial lake outburst flood (GLOF) hazards and impact water resources and hydroelectric power management. There is therefore a pressing need to obtain comprehensive knowledge of the distribution and area of glacial lakes, and also to quantify the variability in their sizes and types at high resolution in HMA. In this work, we developed an HMA glacial lake inventory (Hi-MAG) database to characterize the annual coverage of glacial lakes from 2008 to 2017 at 30-m resolution using Landsat satellite imagery. Our data show that glacial lakes exhibited a total area increase of 90.14 km$^2$ in the period 2008–2017, a +6.90% change relative to 2008 (1305.59 ± 213.99 km$^2$). The annual increases in the number and area of lakes were 306 and 12 km$^2$, respectively, and the greatest increase in the number of lakes occurred in 5400 m elevation, which increased by 249. Proglacial-lake-dominated areas, such as the Nyainqêntanglha and Central Himalaya, where more than half of the glacial lake area (summed over a 1° × 1° grid) consisted of proglacial lakes, showed obvious lake-area expansion. Conversely, some regions of Eastern Tibetan Mountains and Hengduan Shan, where unconnected glacial lakes occupied over half of the total lake area in each grid, exhibited stability or a slight reduction in lake area. Our results demonstrate that proglacial lakes are a main contributor to recent lake evolution in HMA, accounting for 62.87% (56.67 km$^2$) of the total area increase. Proglacial lakes in the Himalaya ranges alone accounted for 36.27% (32.70 km$^2$) of the total area increase. Regional geographic variability of debris cover, together with trends in warming and precipitation over the past few decades, largely explain the current distribution of supraglacial and proglacial lake area across HMA. The Hi-MAG database is available at https://doi.org/10.5281/zenodo.4275164 (Chen et al., 2020), and it can be used for studies of the complex interactions between glaciers, climate, and glacial lakes, and GLOFs, potential downstream risks, and water resources.

**1 Introduction**

High Mountain Asia (HMA), consisting of the whole Tibetan Plateau and adjacent mountain ranges such as the Himalayas, Karakoram, and Pamirs, contains the largest area of mountainous glaciers in the world. Atmospheric warming has resulted in widespread glacier retreat and downwasting in many mountain ranges of HMA (Bolch et al., 2012; Brun et al., 2017), which favors the formation and development of a large number of glacial lakes. However, glacial lakes have been incompletely documented over small time intervals. Glacial lake development varies according to climatic, cryospheric, and lake-specific conditions, including whether the basin geometry is connected to glaciers, and the length of lake/glacier contact (Zhao et al., 2018).

There have been many previously published studies devoted to mapping glacial lakes using remote sensing data over different regions of HMA. Some works have focused on investigating the development of relatively large glacial lakes. Rounce et al. (Rounce et al., 2017) identified 131 glacial lakes in Nepal in 2015 that had an area greater than 0.1 km$^2$. Li et al. (Li et al., 2020) compiled an inventory of glacial lakes ($\geq$0.01 km$^2$) with a spatial resolution of 30 m in the Karakoram mountains. Aggarwal et al. (Aggarwal et al., 2017) shared a new dataset of glacial and high-altitude lakes having an area >0.01 km$^2$ for Sikkim, Eastern Himalaya in the period 1972–2015. Ukita et al. (Ukita et al., 2011) constructed a glacial lake inventory of Bhutan in the Himalayas for the period 2006–2010 based on high-resolution Panchromatic Remote-sensing Instrument for Stereo Mapping (PRISM) and Advanced Visible and Near Infrared Radiometer type 2 (AVNIR-2) data from Advanced Land Observing Satellite (ALOS). Considering small lakes represent less of a GLOF risk, they set 0.01 km$^2$ as the minimum lake size. Ashraf et al. (Ashraf et al., 2012) used Landsat-7 ETM+ images for the 2000–2001 period to delineate glacial lakes greater than 0.02 km$^2$ in the Hindukush–Karakoram–Himalaya region of Pakistan.

Because small glacial lakes are highly variable in their shape, location, and occurrence and are clearly sensitive to the warming climate and glacier wastage, a growing number of scholars have been paying attention to their abundance. Salerno et al. (Salerno et al., 2012) provided a complete mapping of glacial lakes (including lake size less than 0.001 km$^2$) and debris-covered glaciers with a 10-m spatial resolution in the Mount Everest region in 2008. Wang et al. (Wang et al., 2013) utilized Landsat TM/ETM+ images for the years 1990, 2000, and 2010 to map glacial lakes with areas greater than 0.002 km$^2$ in the Tien Shan Mountains. Luo et al. (Luo et al., 2020) examined glacial lake changes (lake area >0.0036 km$^2$) for the entire western Nyainqêntanglha range for five periods between 1976 and 2018 using multi-temporal Landsat images. The International Centre for Integrated Mountain Development provided comprehensive information about the glacial lakes (greater than or equal to 0.003 km$^2$) of five major river basins of the Hindu Kush Himalaya using Landsat images from 2005 (Sudan et al., 2018). Nie et al. (Nie et al., 2017) mapped the distribution of glacial lakes across the entire Himalaya in 2015 using 348 Landsat images at 30-m resolution. They set the minimum mapping unit to 0.0081 km$^2$. Zhang et al. (Zhang et al., 2015) presented a database of glacial lakes larger than 0.003 km$^2$ in the Third Pole for the years 1990, 2000, and 2010.

All of these studies significantly help to fill the data gap relating to information about glacial lakes in the HMA region. At the global scale, Pekel et al. (Pekel et al., 2016) used millions of Landsat satellite images to record global surface water over the past 32 years at 30-m resolution, and many large and visible glacial lakes were also included. More recently, Shugar et al. (Shugar et al., 2020) mapped glacial lakes with areas >0.05 km$^2$ around the world using 254,795 satellite images from 1990 to 2018. Wang et al. (Wang et al., 2020b) developed an inventory of glacial lake with areas greater than 0.0054 km$^2$ across HMA at two time points (1990 and 2018) using manual mapping with 30 m Landsat images. They were the first to introduce a glacial lake inventory at such a large scale, and the data shared will serve as a baseline for further studies related to water resource assessment and glacier hazards.

In summary, a homogeneous, annually resolved inventory and analysis of the spatial and temporal extent of different types of glacial lakes over the entire HMA region is still lacking. In this study, we developed an HMA glacial lake inventory (Hi-MAG) database to characterize the annual coverage of glacial lakes from 2008 to 2017 at 30-m resolution. A total of 40,481 Landsat scenes were processed using the Google Earth Engine (GEE) cloud-computing platform to delineate glacial lakes (located within 10 km of the nearest glacier terminus) larger than nine (e.g., $3 \times 3$) pixels (0.0081 km$^2$) (Nie et al., 2017).

Lakes were manually classified into four categories according to their position relative to the parent glacier or their formation mechanisms (Fig. A1). Category (i), proglacial lakes, are usually connected to the glacier tongue and dammed by glacier ice or unconsolidated or ice-cemented moraines (a mixture of ice, snow, rock, debris, clay, etc.). Proglacial lakes are located next to the glacier terminus and receive melt water directly from their mother glaciers. Category (ii), supraglacial lakes, are ponds that form in depressions on low-sloping parts of the surface of a melting glacier and are dammed by ice or the end-moraine or stagnating glacier snout. Category (iii), unconnected glacial lakes, are not currently directly connected to their parent glaciers but they may to some extent be fed by at least one of the glaciers located in the basin. They may (but not necessarily) have recently detached from ice contact due to glacial recession. Although not directly connected with the parent glaciers, these glacial lakes are also an outcome of glacier melting in response to atmospheric warming. They can supply fresh water to major river systems of the HMA region, and their changes have significant scientific and socioeconomic implications (Nie et al., 2017; Song et al., 2016). Finally, category (iv), ice-marginal lakes, are generally distributed on one side of the glacier tongue, meaning that the lake is dammed by the glacier ice on this side, while on the other side, it is bounded by a lateral moraine. With the increase of atmospheric warming and accelerated melting of glaciers, some glacier tributaries gradually detach from a main trunk glacier. These detachment locations, where glacier melting has been particularly intense, is in some cases also likely to form ice-marginal lakes. We note that such ice-marginal lakes are very common in some parts of the world (e.g., Alaska) but are not common in HMA (Armstrong and Anderson, 2020; Capps et al., 2011). Additionally, purely glacier-dammed lakes are formed by the advance of glaciers and dammed by almost pure glacier ice. Although the dam composition and structure is slightly different between proglacial lakes and glacier-dammed lakes, because they are all located in the front of the glacier tongue and driven by the mother glacier, in the process of appending attributed information to each glacial lake, glacier-dammed lakes were merged into the proglacial lakes category.

Every lake was cross-checked manually for its boundary and attribution. We defined an uncertainty of 1 pixel for the detected glacial lake boundaries and calculated the error in the lake area for the whole HMA region. We also assessed the inventory for climatic and geomorphological influences on lake distribution across HMA.

## 2 Study area and data

### 2.1 Study area

The HMA region refers to a broad high-altitude region in South and Central Asia that covers the whole Tibetan Plateau and adjacent mountain ranges, including the Eastern Hindu Kush, Western Himalaya, Eastern Himalaya, Central Himalaya, Karakoram, Western Pamir, Pamir Alay, Northern/Western Tien Shan, Dzhungarsky Alatau, Western Kunlun Shan, Nyainqêntanglha, Gangdise Mountains, Hengduan Shan, Tibetan Interior Mountains, Tanggula Shan, Eastern Tibetan Mountains, Qilian Shan, Eastern Kunlun Shan, Altun Shan, Eastern Tien Shan, Central Tien Shan, and Eastern Pamir

(Figs. 1 and 6a). It extends from 26°N to 45°N and from 67°E to 105°E, and the altitude of the plateau is about 4500 m on average (Baumann et al., 2009). It is made up of alternating mountains, valleys, and rivers, and the terrain is fragmented, showing a decreasing terrain from northwest to southeast. The HMA region has a series of east–west mountain ranges that occupy most of the area. Among these, Tanggula Shan lies in the central part of HMA, with an altitude of over 6000 m. The heights of the fifteen highest mountains in the Himalayas are greater than 8000 m, while the peaks of the mountains in the northern plateau are greater than 6500 m. The north–south mountain ranges are mainly distributed in the southeast of the plateau and near the Hengduan Mountain area. These two groups of mountains constitute the geomorphic framework and control the basic pattern of the plateau landform. Continuous and discontinuous permafrost have developed on the higher land and north-facing slopes.

The HMA region is the source of several of Asia's major rivers, including the Yellow, Yangtze, Indus, Ganges, Brahmaputra, Irrawaddy, Salween, and Mekong. They play a crucial role in downstream hydrology and water availability in Asia (Immerzeel and Bierkens, 2010). Most glaciers in the Tibetan Plateau are retreating, except for the Western Kunlun (Neckel et al., 2014; Kääb et al., 2015) and the Karakoram, where a slight mass gain is occurring (Bolch et al., 2012; Gardner et al., 2013). Moreover, glaciers in different mountain ranges show contrasting patterns. Local factors (e.g., exposure, topography, and debris coverage) may partly account for these differences, but the spatial and temporal heterogeneity of both the climate and degree of climate change may be the main reason. Glacial lakes are formed and develop temporally with the retreat or thinning of glaciers and are directly or indirectly fed by glacier meltwater. They are located within 10 km of the nearest glacier terminus (Wang et al., 2013; Zhang et al., 2015).

The HMA climate is under the combined and competing influences of the East Asian and South Asian monsoons and of the westerlies (Schiemann et al., 2009). This unique geographical position produces an azonal plateau climate characterized by strong solar radiation, low air temperatures, large daily temperature variations, and small differences between annual mean temperatures (Yao et al., 2012). The annual mean temperature is 1.6°C, with the lowest temperature of −1°C to −7°C occurring in January and the highest temperature of 7°C to 15°C occurring in July. The cumulative annual precipitation is about 413.6 mm.

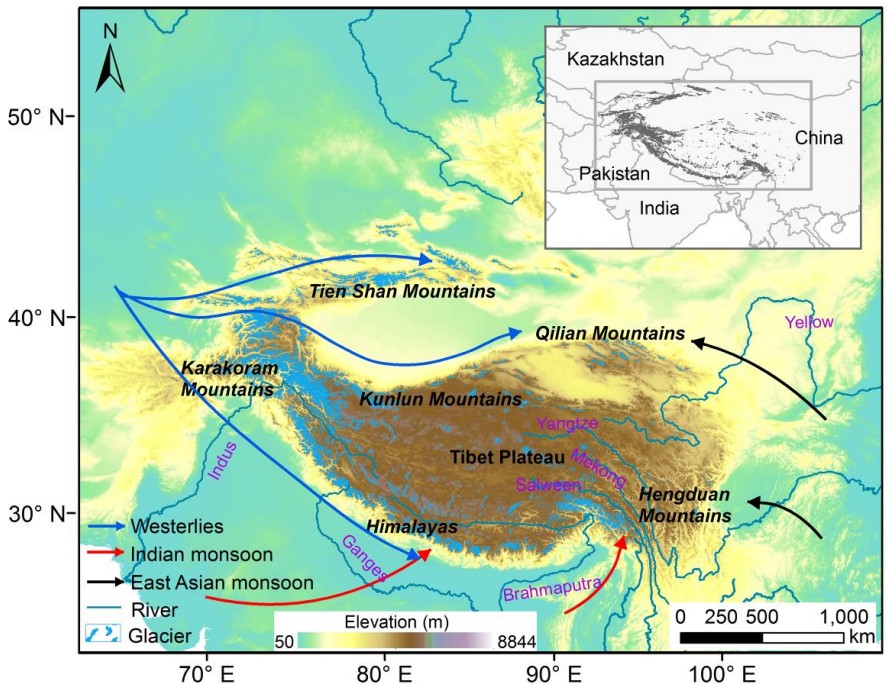

**Fig. 1.** Location of the HMA region. Glacier outlines from the Randolph Glacier Inventory (RGI v5.0), the Second Chinese Glacier Inventory (CGI2), and the Glacier Area Mapping for Discharge from the Asian Mountains (GAMDAM) glacier inventory are drawn in blue.

## 2.2 Dataset

A total of 40,481 satellite images, including Landsat 5 TM imagery during 2008–2011, Landsat 7 ETM+ imagery in 2012, and Landsat 8 OLI during the period 2013–2017, were available in GEE and were used to produce the annual glacial lake maps over the entire HMA (Fig. 2). Here, when Landsat 5 or 8 data were available, Landsat 7 ETM+ imagery with Scan Line Corrector (SLC)-off gaps were generally excluded due to the artefacts induced by the slatted appearance of the original images, but these were exclusively used for the glacial lake mapping in 2012 since no other Landsat data were acquired that year. For the years before 2008, all the available Landsat 5 TM data in each year (e.g., 2004, 2005, 2006, and 2007) do not fully cover the HMA region.

The SLC-off condition of Landsat ETM+ introduces artefacts because the slatted appearance of the original images is occasionally carried into the glacial lake map in 2012. Techniques to fill the SLC-off gaps exist, but these create artificial values that will result in false detections of water (Chen et al., 2011). Considering the strong spatial and temporal variability of glacial lakes such as supraglacial lakes, techniques that merge data from one or more SLC-off fill scenes for generation of a gap-free image require careful use, even when using the thousands of Landsat ETM+ images. It is noted that water mapping using multi-temporal time-series images at large scales usually avoids the use of such techniques (Mueller et al., 2016). Therefore, Landsat 7 ETM+ data with intensive slatted appearance are not suitable for the classification of numerous of glacial lakes. In this study, because the only useable data source for 2012 was from Landsat 7 ETM+, to ensure continuity of annual data from 2008 to 2017, we applied our best efforts to manual extraction of the glacial lakes from the 2012 ETM+ images with the highest possible accuracy.

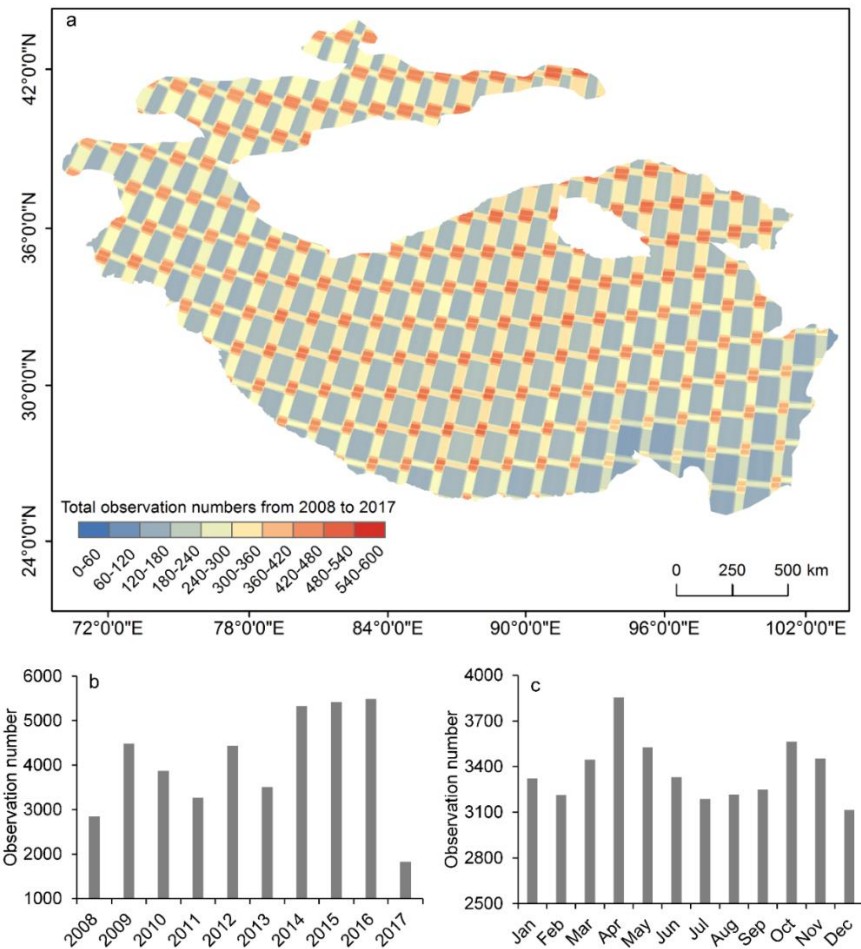

**Fig. 2. (a)** The distribution of total observation numbers from all GEE Landsat scenes, along with these numbers broken down by **(b)** year and **(c)** month.

## 3 Methods

### 3.1 Satellite imagery selection strategy

One effective solution to reduce the influence of seasonal lake fluctuations on the mapping is to map glacial lakes and measure their long-term changes during stable seasons when the lake extents are minimally affected by meteorological conditions and glacier runoff. Here, based on analyses of the mapping times of glacial lakes in different regions, the selected time series of Landsat data were generally from July to November. During this period of each year, the Landsat imagery featured lower perennial snow coverage. Following glacier runoff and precipitation, the area of a glacial lake is large and changes in this area will be small (Nie et al., 2017; Chen et al., 2017; Zhang et al., 2015). These lakes may also reach their maximum extent around the end of the glacier ablation season (June to August) (Gardelle et al., 2013; Liu et al., 2014), except in Central and Eastern Himalaya, where peak ablation extends into post-monsoon September and October. In monsoon-affected areas such as Nepal and Bhutan, monsoon cloud cover from July to mid-September means that clear-sky images can mostly only be obtained from late September to November. The southeastern Tibet region is problematic, not only because the observation season is short, but as a result of abundant cloud cover, which is formed by the warm and humid airflow raised by the topography (Zhang et al., 2020; Umesh et al., 2018; Qiao et al., 2016).

As the most highly variable glacial lakes in the study area, supraglacial lakes change preferentially in the year, showing an increase in area during the pre-monsoon, and rising to their peak area in the early monsoon (June to July) (Miles et al., 2017a; Miles et al., 2017b). Although the selected image seasons are slightly different due to the meteorological conditions in different regions, they all comply with the same criterion that the lakes were in clear-sky images having small snow coverage. This ensured the initial reliability of the mapping of glacial lakes through the GEE cloud-computing platform. If no valid observations could be obtained, then the optimal mapping time needed to be broaden during the whole year.

To further increase data availability, and also as the basis for data selection in the periods beyond the optimum mapping time, we set two criteria for the selection of imagery with valid observations over the potential glacial lake area by using the cloud-score functions in GEE, including (i) cloud cover being less than 20% in the 10-km buffer around each glacier outline of a Landsat scene, or (ii) less than 20% cloud cover for the entire scene. The cloud-score functions in GEE may have significant difficulty in detecting clouds in mountain headwaters with high snow and ice cover, where large amounts of snow and ice are likely to be identified as clouds. However, in this study, it was considered better to use much stricter criteria to filter out a larger number of images with lots of cloud or cloud-lookalike objects (snow/ice) to finally select only images with good observations.

**3.2 Extraction of glacial lake outlines**

For the development of the Hi-MAG database, we applied a systematic glacial lake detection method that comprised two steps: initial glacial lake extraction and subsequent manual refinement of these lake mapping results. The main procedures for glacial lake mapping using Landsat data, as shown in Fig. 3, are as follows. (i) The Landsat top-of-atmosphere data were clipped according to the extent of the glacier buffers and assembled into a time-series dataset. (ii) Poor-quality observations were identified—these included areas affected by cloud, cloud shadow, topographic shadow, and SLC-off gaps. Here, we used the Fmask routine (Zhu and Woodcock, 2012) to detect the clouds and cloud shadows in an imagery. Fmask has the advantage of being able to process a large number of images in a computationally efficient way. Topographic shadows are located in the areas where the sunlight is blocked. Generally, on the dark side of high mountains, the surface gradients are great, and the terrain reliefs are small. Therefore, topographic shadows were masked using the slopes (larger than 10°) and shaded relief values (less than 0.25) calculated from Shuttle Radar Topography Mission (SRTM) data (Li and Sheng, 2012; Quincey et al., 2007). This removes a considerable number of mountain shadows that have the similar spectral reflectance as water bodies. However, the SRTM digital elevation model (DEM) was generated in 2000, which is different from the acquisition time of the Landsat images used for the glacial lake mapping in this study. The derived slopes and shaded relief cannot therefore fully represent the conditions on the date a given Landsat scene was acquired. As a consequence, some lakes that have grown at steep glacier tongues may be masked, and some mountain shadows that interfere with the mapping results of glacial lakes from GEE still remain, leading to the fact that glacial lakes in steep areas are omitted, and residual shadows are misclassified as glacial lakes. As for the SLC-off gaps in the ETM+ images, lakes outside the gaps will be accurately classified, but if they are covered by gaps, then they will be misclassified. Errors caused by striped gaps in Landsat ETM+ data were manually corrected using additional high-quality scenes across the whole year with the assistance of images from adjacent years. (iii) The modified normalized difference water index (MNDWI) was calculated (Xu, 2006). (iv) The potential glacial lake areas were extracted by applying an adaptive MNDWI threshold (Li and Sheng, 2012). The minimum number of water pixels used to define a glacial lake in an image is inconsistent in different studies. For example, Zhang et al. (2015) set the smallest detectable glacial lakes in the Third Pole as being larger than 0.0027 km$^2$ (three

connected pixels) using the Landsat TM/ETM+ data. Nie et al. (2017) selected 0.0081 km$^2$ (nine connected pixels) as the
minimum mapping unit to map glacial lakes in the Himalayas. Other studies have set the minimum threshold areas as
0.001km$^2$ (Salerno et al., 2012), 0.002km$^2$ (Wang et al., 2013), 0.0036km$^2$ (Luo et al., 2020), 0.0054km$^2$ (Wang et al.,
2020b), of 0.01km$^2$ (Li et al., 2020). A smaller minimum mapping unit will detect more glacial lakes. However, the
uncertainty this brings will also be larger than using a larger threshold at the same resolution (Salerno et al., 2012). Our
results demonstrate that a lake area covering fewer than nine water pixels will have an area error of greater than 50% (see
Section 4). Given the uncertainty in the areas of glacial lakes and the spatial resolution of Landsat data, in this study, glacial
lakes larger than nine pixels (≥0.0081 km$^2$) were considered as the minimum mapping unit. (v) Manual inspection and
refinement of individual glacial lake was conducted, and the related attributions were added for each lake.

Based on the automated processing, nearly 60% of glacial lakes in each year can be correctly classified. Of the other
lakes that were not properly classified, 30% were missed and 10% were misclassified. For such a large-scale area that is
characterized by various and complex climatic, geological, and terrain conditions, this classification method is simple but
effective. The results are also reasonable since they provide very low commission errors. To ensure the quality of the
inventory, strict quality control was conducted to visually inspect and correct the mapping errors after the automated
processing using GEE. False lake features, mainly identified as mountain shadows and river segments, were manually
removed by overlapping mapped lake shorelines on the source Landsat imagery and higher-resolution imagery in Google
Earth. Some glacial lakes may be covered by ice and clouds for years, grow at steep glacier tongues, or show heterogeneous
reflectance with the surrounding backgrounds. For these missing glacial lakes, their boundaries were edited further using
ArcGIS. Furthermore, a cross-check and modification was conducted for each glacial lake based on the lake-mapping
results in conjunction with multi-temporal Landsat imagery. Here, all the Landsat imagery that was used for the inspection
was downloaded manually from the United States Geological Survey (USGS) Earth Explorer website
([https://earthexplorer.usgs.gov/](https://earthexplorer.usgs.gov/)). The outputs for each lake polygon include information about the lake type, elevation,
Euclidian distance to the nearest glacier terminus, area, and perimeter. Note that if there was more than one suitable satellite
image in a year, the image with the lowest cloud cover was selected for the calculation of the area and perimeter of a given
lake. Each mountain range was characterized individually by utilizing the mountain boundary shapefile in HMA
([http://geo.uzh.ch/~tbolch/data/regions_hma_v03.zip](http://geo.uzh.ch/~tbolch/data/regions_hma_v03.zip)).

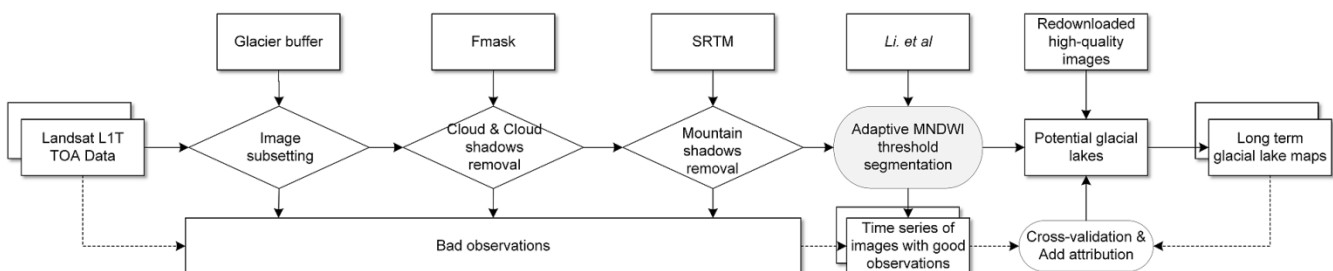

**Fig. 3.** Diagram of the glacial lake mapping workflow.

### 3.3 Yearly lake area changes calculations

Based on the final generated lake inventory data, we used the slope of a linear regression of the lake area (over the grid cells
of 1° × 1°) versus mapping year to qualify the yearly changes in lake area during the study period. The change analysis used

the Theil–Sen estimator, which chooses the median slope of all the derived fitted lines, can effectively represent long-term area changes due to its robustness for trend detection and its insensitivity to outliers. It is also useful for the elimination of effects arising from differences in sensor performance for the mapping of glacial lakes (Sen, 1968; Song et al., 2018).

Although all the lakes were manually checked and edited, due to the limitation of available images and other factors, the conditions for glacial lake mapping were not perfectly consistent for each year. For example, the image dates were not consistent across the whole HMA region because of atmospheric disturbances, and there were also influences from varying lake characteristics, image quality (Bhardwaj et al., 2015; Thompson et al., 2012), ice and shadows that obscured the lakes, which all contributed to detection errors in the lake extent and their annual variation. Generally, these errors were objective and acceptable as a result of the nature of the limited remote sensing data. For this study, because we used time-series data covering a period of ten years for the estimation of annual changes in lake area, and also because the errors only account for a small proportion of the total glacial lake area for each year, the errors in the observed lake area caused by these different effects do not appear to affect the trends in the statistical results. In addition to the Theil–Sen estimator, a Mann–Kendall trend test was used to detect and further confirm the statistical confidence of the linear regression results, and all the estimated trends were found to fall within the 90% confidence intervals. The upper and lower change estimates satisfying the 90% confidence interval for the slope were also derived over the whole HMA region (Fig. A2).

## 4 Cross-validation and uncertainty estimates

Accuracy assessment of the mapping results is difficult due to the lack of field measurements of glacial lakes in continental-scale areas such as HMA. To obtain quality-controlled data, the glacial lake vectors over the entire HMA for the years from 2008 to 2017 were rechecked and reedited individually through dynamic cross-validation by ten trained experts. This was a time-consuming process but was essential for maximizing the quality of the data.

A key factor influencing the estimation of the uncertainty in the glacial lake area measurements is the spatial resolution of the satellite data. In this study, the uncertainty of the glacial lake area was estimated as an error of ±1 pixel on either side of the delineated lake boundary. The percentage error of the area determinations, $A_{er}$, is then proportional to the sensor resolution and is given by (Krumwiede et al., 2014)

$$A_{er} = 100 \cdot (n^{1/2} \cdot m) / A_{gl}, \tag{1}$$

where $n$ is the number of pixels on the boundary of a glacial lake, as approximated by the ratio of the perimeter length and the spatial resolution, $m$ is the area of a pixel in the Landsat image (m$^2$), $A_{gl}$ is the lake area (m$^2$) and the factor 100 converts the value to a percentage.

Assuming an uncertainty of 1 pixel for the detected glacial lake boundaries, we calculated the systematic errors for the whole HMA region, and the results are shown in Fig. 4. For the years between 2008 and 2017, the area uncertainty of each glacial lake generally ranged from 0.30% to 50%, with the mean value falling around 17% and the standard deviation around 11% (Fig. 4a). The maximum and mean values of area uncertainty for the glacial lakes in 2010 were the lowest, while for 2016, the corresponding statistics were the highest. This can be attributed to a number of different factors. The maximum in the area uncertainty of glacial lakes is related to the shape and size of a certain lake, as can be seen from Equation (1). However, its mean value is equal to the sum of the area uncertainties of each glacial lake divided by the total number, which depends on the total number of glacial lakes in a given year, as well as the shape and area of each lake. Furthermore, a close relationship can be found between the area uncertainties and sizes of the glacial lakes (Fig. 4b). Most

of the large glacial lakes (area $\geq 0.04$ km$^2$) have a mean area uncertainty of about 7%. These systematic errors were more significant for the small-sized glacial lakes. We measured glacial lakes down to 0.0081 km$^2$ (nine pixels in Landsat imagery), where systematic errors calculated by Equation (1) were ~50%.

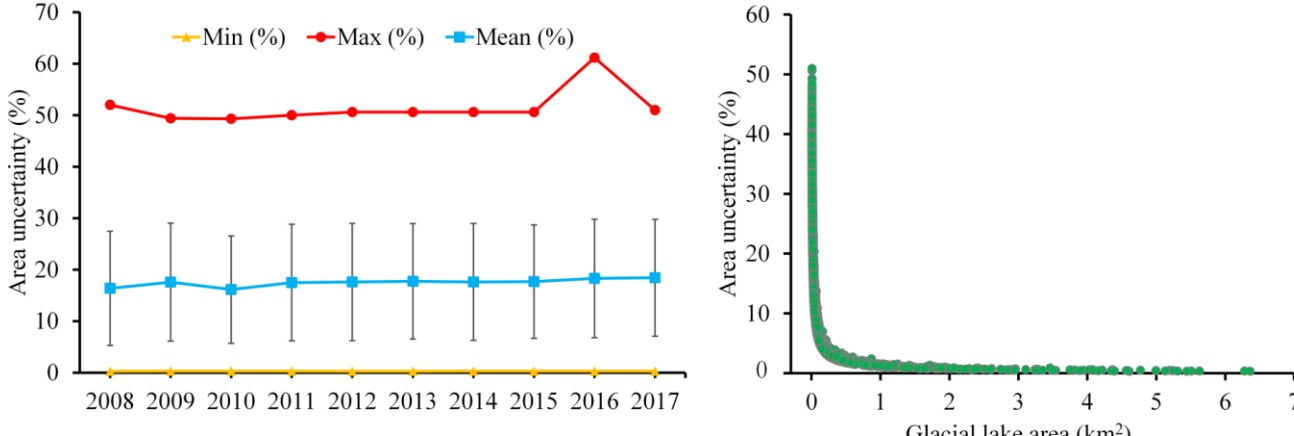

**Fig. 4. (a)** Statistics relating to the uncertainty (%) in the measured areas of glacial lakes for the years from 2008 to 2017. **(b)** Relationship between the area uncertainties and the areas of all the glacial lakes in HMA in 2017.

## 5 Results

### 5.1 Distribution of various types and sizes of glacial lakes

The area coverage of glacial lakes increased by 90.14 km$^2$ in the period 2008–2017, a 6.90% increase relative to 2008 (1305.59 ± 213.99 km$^2$) (Fig. 5a). A Theil–Sen regression fit to all the data showed a mean expansion rate of 12 km$^2$ a$^{-1}$ for the ten-year record, as shown in Fig. 5a. Meanwhile, the estimated changes in glacial lake number from 2008 (12,593 lakes) to 2017 (15,348 lakes) showed an average increase of 306 lakes a$^{-1}$. The steeper percentage increase in the number of lakes (22.33%) compared to the slower expansion of their area (8.79%) based on their linear fit trends shows that many small glacial lakes formed over this period. The number of lakes increased most rapidly in areas beyond 4400 m above sea level (a.s.l.), especially beyond 5300 m (Fig. 5b). The increase in proglacial lakes was concentrated above 4900 m (Fig. 5c). Unconnected glacial lakes grew very slightly in total area below 4400 m (Fig. 5d) but increased notably more at higher elevations. Glaciers are retreating and thinning at ever-higher elevations (Nie et al., 2017), causing the formation of new supraglacial lakes at high elevations, the expansion of existing ice-contact lakes, and the detachment of glaciers from some lakes.

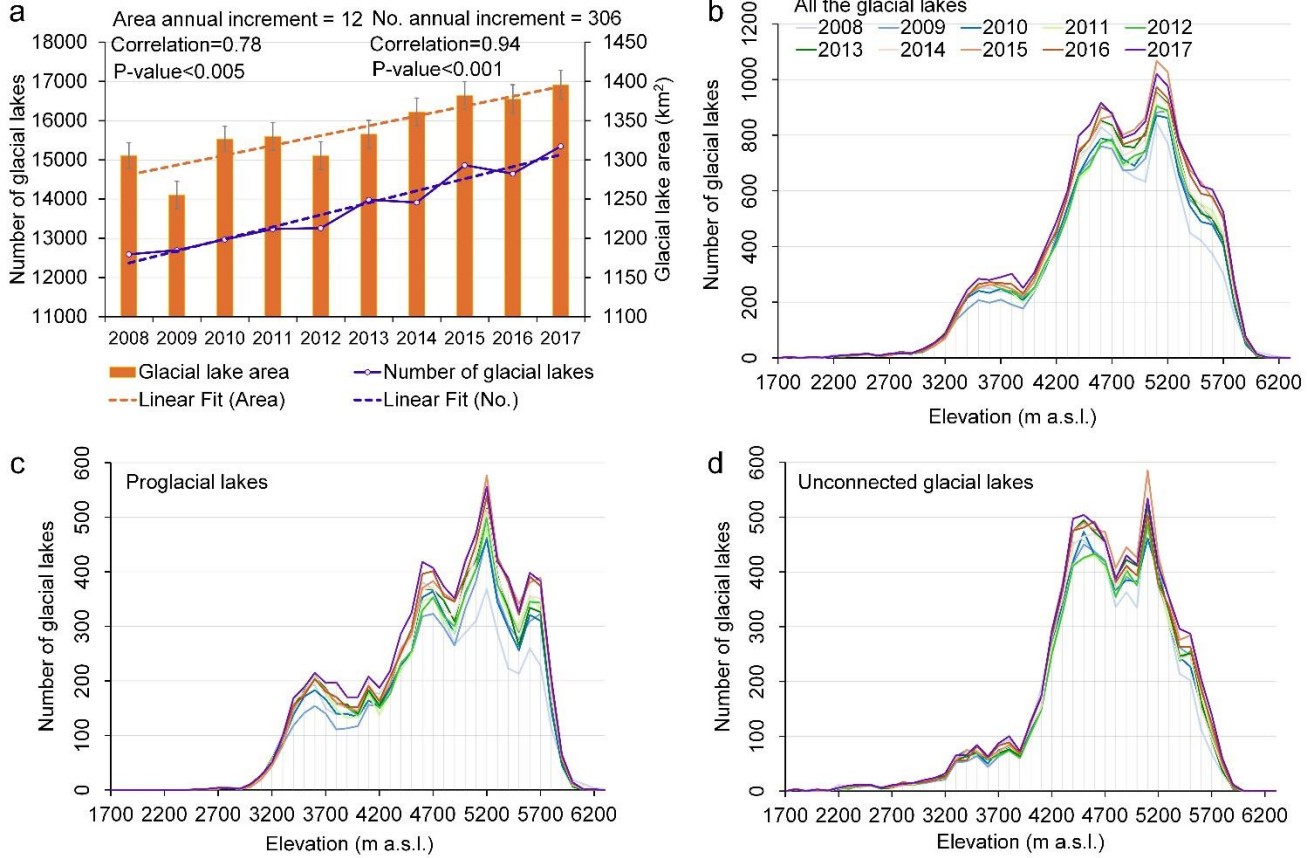

**Fig. 5.** Numbers and total areas of glacial lakes for different years. **(a)** Total number and area of glacial lakes for HMA in the period 2008–2017. The annual increment is the slope of the trend of annual lake area and number. Altitudinal distribution (100-m bin sizes) of lake numbers for **(b)** all glacial lakes, **(c)** proglacial lakes, and **(d)** unconnected glacial lakes.

Annual changes in glacial lakes were further analyzed spatially using a $1° \times 1°$ grid over 22 mountain regions (Fig. 6a) using Theil–Sen regression analysis. An analysis of the mountain-wide lake area loss/gain from 2008 to 2017 was conducted (Table A1). Negative or undiscernible changes in glacial lake area were observed in the Eastern Tien Shan, Eastern Hindu Kush, Hengduan Shan, and Eastern Tibetan Mountains (Fig. 6b), thus reducing the otherwise overall increasing glacial lake area in HMA. The Eastern Hindu Kush lost 2.8 $km^2$ of lake area (Table A1), with a negative area change ($-0.43$ $km^2$ $a^{-1}$) near 35°N, 73°E. Glacial lakes in Nyainqêntanglha and Gangdise Mountains exhibited area loss and gain in different regions. In contrast, Central and Eastern Himalaya and Central Tien Shan showed rapid increases in lake area. Between 2008 and 2017, Central Himalaya's glacial lake area increased by 27.09 $km^2$ (Table A1), exhibiting both a high density of 47 glacial lakes per 100 $km^2$ in 2017 (Fig. A3) and rapid growth, $+0.94$ $km^2$ $a^{-1}$, in lake area due to retreat and thinning of debris-covered glaciers (Song et al., 2016). Moderate area gains occurred along most of the Western Kunlun and Tanggula Shan, e.g., $+0.38$ $km^2$ $a^{-1}$ in Tanggula Shan. The areas of glacial lakes in Pamir Alay, Eastern Pamir, and Eastern Kunlun Shan were spatially and temporally invariant across the whole observation record.

We found that glacial lakes exhibited different expansion trends for different lake types and supraglacial and ice-marginal lakes have relative few coverage areas comparing with proglacial and unconnected lakes (Fig. 6b and Fig. 6c). In the Nyainqêntanglha and Central Himalaya, around half of the glacial lake area consisted of proglacial lakes, where most

growth occurred. In the negative lake-growth (shrinkage) regions of the Eastern Tibetan Mountains and Hengduan Shan,
unconnected glacial lakes were dominantly occupied. As the interaction with a glacier gradually weakens, part of the water
source supplied by that glacier is reduced, and when combined with the effects from atmospheric warming and a decrease in
precipitation, this means that regions mainly consisting of unconnected glacial lakes show a trend of decreasing area.
Proglacial lakes contributed approximately 62.87% (56.67 km$^2$) to the total area increase over HMA (Tables A1 and A2).
Proglacial lakes in Central Himalaya, Eastern Himalaya, and Western Himalaya accounted for 36.27% (32.70 km$^2$) of the
total area increase. In general, proglacial lakes are the main contributor to recent lake evolution in HMA.

We also noted the large area growth of lakes occurred in areas with a relatively large proportion of small glacial lakes,
and this was mainly due to the rapid growth of existing lakes and the formation of new lakes (Fig. 6d). For example, in
some areas of Central and Eastern Himalaya and Nyainqêntanglha that have large annual increases in lake area (greater than
0.23 km$^2$ a$^{-1}$), glacial lakes with a size of less than 0.16 km$^2$ occupied more than 30% of the total area (Table A3). In
particular, in Nyainqêntanglha, the area of small glacial lakes (≤0.16 km$^2$) accounted for 69.47% of the total area.

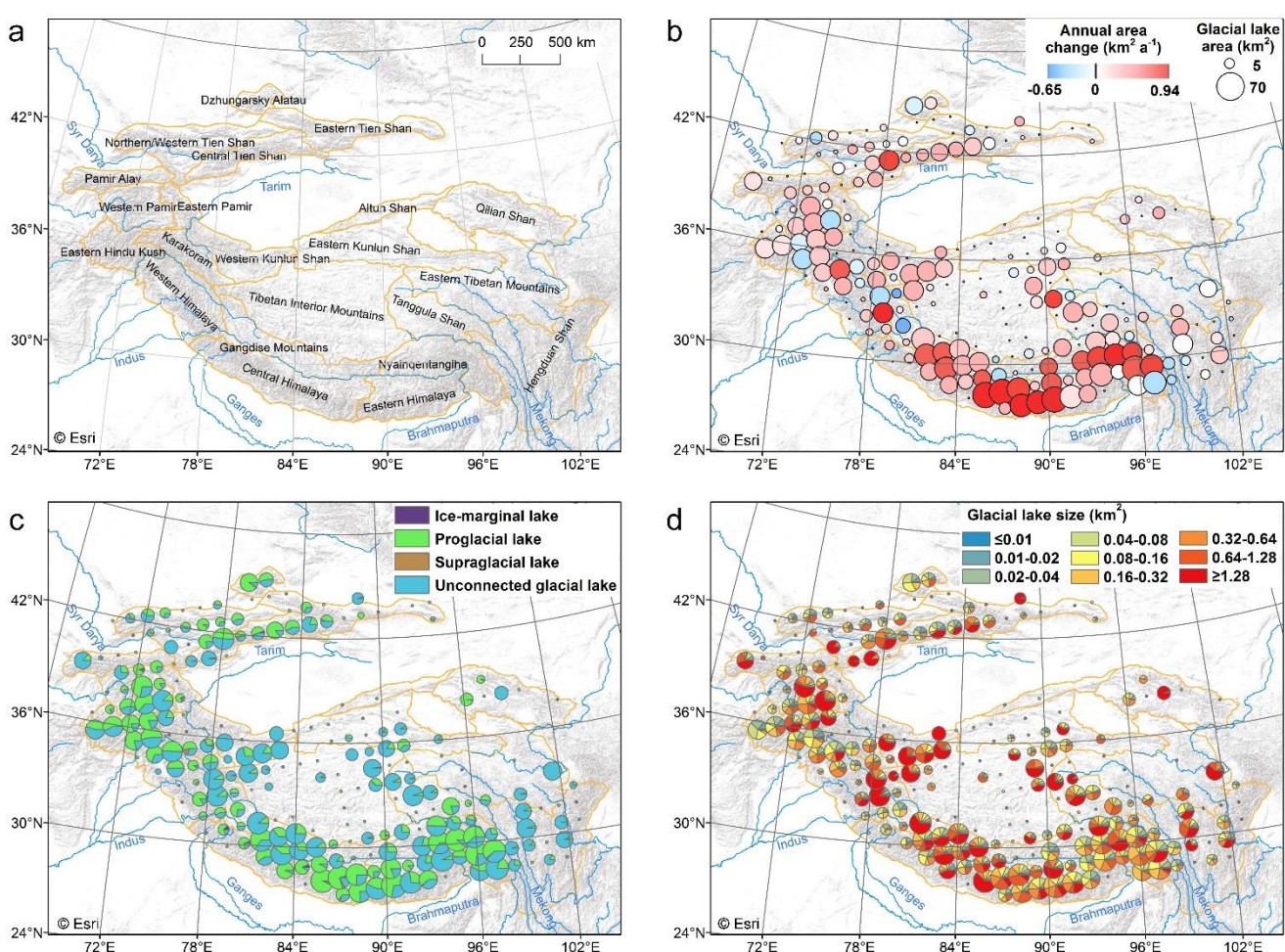

**Fig. 6.** Glacial lake area changes and area distribution. **(a)** Geographic coverage of mountain ranges in HMA. **(b)** Annual
rate of change in lake area (2008–2017) on a 1° × 1° grid. The sizes of the circles represent the total glacial lake area in
2017. **(c)** Proportional areas of four types of glacial lakes in 2017. **(d)** Areas of different sizes of glacial lakes in 2017. The
terrain basemap is sourced from Esri (© Esri).

**5.2 Influencing factors of current distribution of glacial lakes**

To explore factors that have potentially influenced the glacial lake distribution across HMA, we focus on proglacial and supraglacial lakes, for which the changes are closely related with glaciers and expansion is most rapid. Proglacial lakes frequently develop from the enlargement and coalescence of one or more supraglacial lakes (Thakuri et al., 2016; Umesh et al., 2018). Proglacial and supraglacial lake development from 2008 to 2017 is significantly correlated with initial lake area in 2008 ($R = 0.82$, Table A4); larger ice-contact proglacial lakes imply a larger water body in contact with the calving front of the glacier and more rapid retreat (Truffer and Motyka, 2016; King et al., 2019).

For the years before 2008, the year-round Landsat 5 TM data in many years do not fully cover the HMA region. In this study, we constructed the inventory over a ten-year time period. This is shorter than typical glacier response times, which start from a minimum of ten years for short, steep glaciers, to over 150 years for long, debris-covered glaciers (Scherler et al., 2011). Hence, lake expansion is not expected to be coupled with short-term climate trends, particularly for debris-covered glaciers (Umesh et al., 2018). In the inclusion of mass balance forcing of glacial lake changes, the same questions about response times also occur. Hence, rather than focus on the short term evolution of lake expansion, we investigated whether the climate and other factors have influenced the overall distribution of lake area, as observed in 2017.

To investigate the factors influencing the predominance of proglacial and supraglacial lakes, geomorphic, topographic, and climate parameters were correlated with lake area over a $1° \times 1°$ grid, and these were aggregated (taking the mean or sum) for HMA regions. A statistically significant positive correlation exists between lake area and debris-covered glacial area (after Scherler et al. 2018) across HMA ($R = 0.36$, Table A4), confirming the predominance of proglacial and supraglacial lakes forming on debris-covered glacier tongues (Nie et al., 2017). Correlations and significance levels strengthen if the Karakoram is excluded (Table A5). The Karakoram is known as an anomaly of positive glacier mass balances and glacier advances (Gardelle et al., 2012) and also has an anomalously small area of proglacial lakes. Glacier length (RGI-Consortium, 2017) and debris cover are strongly correlated ($R = 0.85$, Table A4), reflecting abundant debris on most large, low-gradient valley glacier tongues in HMA; in turn, there is a statistically significant direct correlation between glacier length and lake area ($R = 0.32$, Table A4), as these tongues provide the ideal conditions for the coalescence of supraglacial ponds and formation of large proglacial lakes (Nie et al., 2017; Richardson and Reynolds, 2000). Glaciers are generally longest and most heavily debris covered in the Hindu Kush Himalaya region (Figs. 7a and 7b).

Some regions have comparable amounts of large debris-covered glaciers but substantial differences in total lake area and area-growth rates (for example, Central Himalaya compared to Central Tian Shan or Western Pamir, Table A5). Regional differences in multi-decadal climate trends could play a role in this observation, with Nyainqêntanglha and the Central and Eastern Himalayan regions all being characterized by rapid warming and decreased precipitation since 1979 (Figs. 7c and 7d), favoring negative glacial mass balances (Brun et al., 2017). This plausibly explains why the lake area is typically larger in these regions relative to adjacent regions further to the west and north (e.g. Western Himalaya) despite often similar glacier characteristics (in terms of debris cover and glacier length) (Figs. 7e and 7f). Furthermore, there is very little debris-covered area but rapid warming in Eastern Himalaya, where proglacial lakes are abundant (Fig. 7f). These results emphasize that the distribution of supraglacial and proglacial lakes across HMA is primarily associated with the presence of large debris-covered glaciers, but regional variability in warming and precipitation trends over the past few decades have likely also had some influence (Shugar and Clague, 2011; Zhao et al., 2019; Umesh et al., 2018; Scherler et al., 2018). These results are consistent with previous findings at regional scales, which have demonstrated a rapid expansion of proglacial lakes on debris-covered glaciers, with expansion in the upstream direction demonstrated to occur primarily through a

process of subsidence at the lake-contact debris-covered glacier tongue (Harrison et al., 2018; Song et al., 2016; Song et al., 2017a).

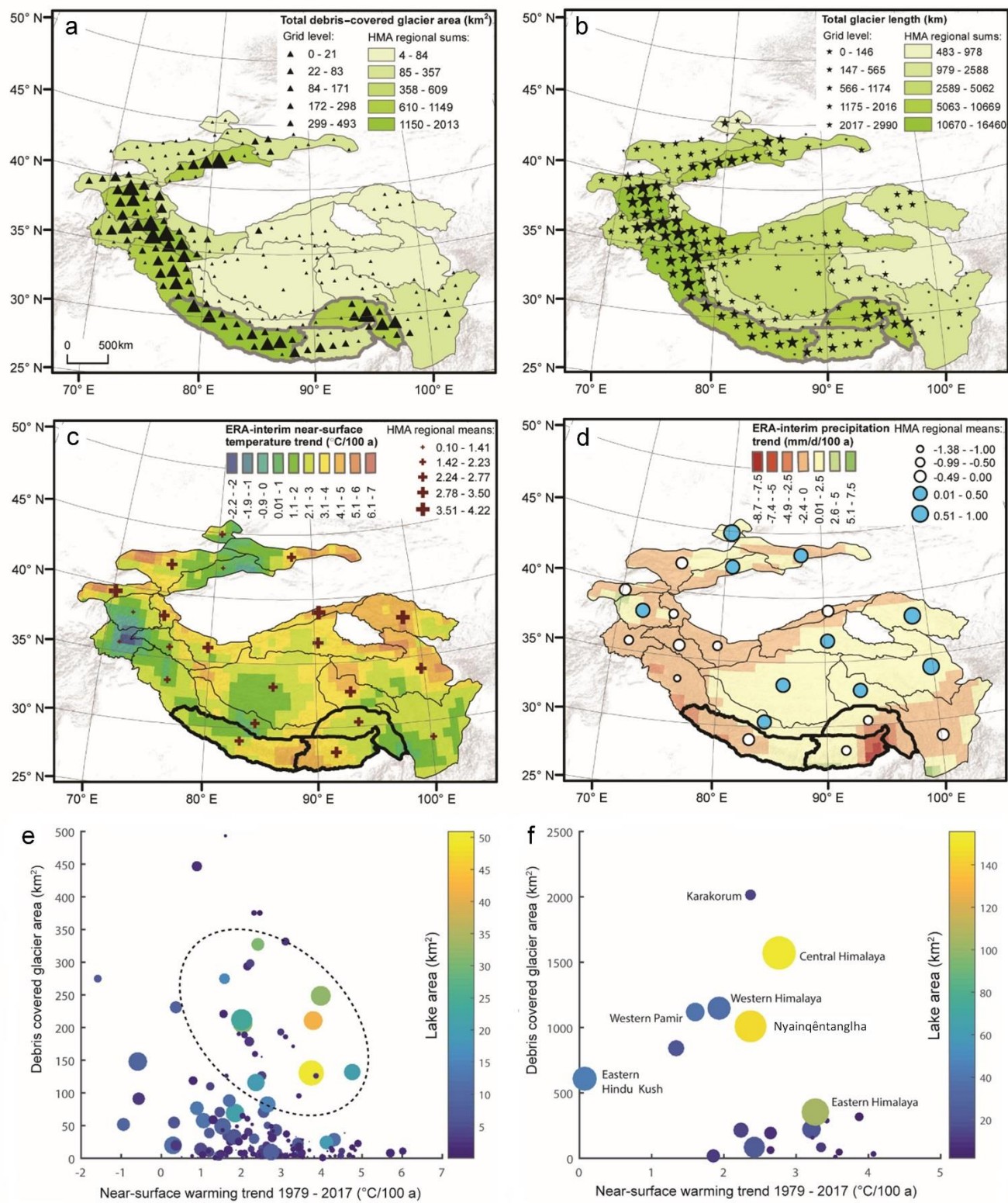

**Fig. 7.** Geomorphic and climatic influences on lake distribution. **(a)** Debris-covered area and **(b)** glacier length aggregated on a 1° × 1° grid. Linear trends in **(c)** temperature and **(d)** precipitation calculated for 1979–2017 from ERA-Interim, including aggregated means over HMA regions. Relationship between total debris-covered area, near-surface temperature

warming, and proglacial and supraglacial lake area of 2017 in (**e**) $1° \times 1°$ grid tiles and (**f**) HMA regions. Some regions discussed in the text are labeled. The lake coverage is high in areas of both rapid warming and high debris cover (**e**, dashed ellipse). Dot sizes are proportional to lake number. See Table A4 for details on data sources.

## 6 Discussions

### 6.1 Comparison with other lake datasets

We compared our dataset with that of Wang et al. (2020) for the closest period (2017 from the Hi-MAG database and 2018 from Wang et al., 2020) over the spatial extent of our HMA region. The differences in the total number and area of lakes between these two datasets are 6206 and 223.97 km$^2$, respectively. We also found that 2077 glacial lakes with a total area of 178.77km$^2$ in our Hi-MAG dataset were not detected by Wang et al. The main reasons for the missed glacial lakes in Hi-MAG are because the interference of some bad observations (cloud or snow), glacial lakes that have dried up or outburst, or were located in the middle of the river.

To test the spatial correlation of the distributions of the glacial lakes in the two datasets, we compared the numbers of glacial lakes and their areas aggregated on a $0.1° \times 0.1°$ grid for the HMA regions. The results for the total glacial lake area, areas of glacial lakes larger than 0.04 km$^2$, and the number of glacial lakes larger than 0.04 km$^2$ are shown in Fig. A4. A clear and strong correlation can be observed for all the statistics between the Hi-MAG dataset and the glacial lake data of Wang et al. Most of the points are distributed around the 1:1 line, which shows that there is great consistency between the two sets of data.

To quantitatively and systematically evaluate the accuracy of our data, we implemented stratified random sampling (Song et al., 2017b; Stehman, 2012), in which the glacial lakes were divided into four strata. The sample sizes were the spatial resolution (30 m) of the data, and the strata were designed as: C0W0 indicates that both the results are non-glacial lakes; C0W1 indicates a non-glacial lake in the present data and a glacial lake in Wang's data; C1W0 indicates a glacial lake in the present data and a non-glacial lake in Wang's data; and C1W1 indicates that both results are glacial lakes.

A total of 4000 points were randomly selected, as shown in Fig. A5. The number of samples for C1W1 and C1W0 were 1300 and 700, respectively, and these numbers have almost the same ratio as that between the total areas for the two strata (1450.50 km$^2$ vs 732.77 km$^2$). Because of the approximate total area with C1W0, we also randomly selected 700 samples from stratum C0W1. The remaining 1300 samples were from C0W0. Every validation sample was visually examined using Landsat imagery and higher-resolution imagery in Google Earth. Sample pixels were interpreted by a regional glacial lake mapping expert, and ambiguous samples were cross-validated by a second observer. If a sample was difficult to interpret, it was marked as ambiguous and excluded from the accuracy assessment. The sample number estimates were produced for each of the four strata (Table A6), and these strata totals were then summed to obtain the total accuracy.

For the 1300 samples that were considered by both datasets to be non-glacial lakes, after the pixel-by-pixel verification, 1215 were found to indeed be non-glacial lakes, while 37 were missed glacial lakes. In contrast, 1260 out of the 1300 samples belonged to the class of glacial lakes, and 25 were misclassified as glacial lakes by both inventories. A total of 307 error pixels were found in the results of Wang et al., constituting about half of the total validation number. For the glacial lakes identified only by our inventory, 678 out of 700 were correctly classified. Our results yielded high overall classification accuracy (88%), user's accuracy (97%), and producer's accuracy (82%) for glacial lake classification using Landsat data.

The Hi-MAG dataset was also compared with other Landsat-based lake inventories (Nie et al., 2017; Pekel et al., 2016; Zhang et al., 2015). The number of lakes in Hi-MAG was found to be 7268 higher and the area was 644.26 km$^2$ greater than the estimation for the Tibetan Plateau (Zhang et al., 2015). The largest discrepancies were found in the Gangdise, Himalaya, and Nyainqêntanglha Mountains in 2010. Across the Himalaya region, we found 476.09 km$^2$ of glacial lakes, 4.57% more than previous estimates in 2015 (Nie et al., 2017). In addition, we qualitatively compared the lake extents between the publicly available high-resolution Global Surface Water (GSW) dataset (Pekel et al., 2016) and our Hi-MAG database summed by mountain range in 2015. For the sake of a reliable comparative analysis, lake polygons in the Hi-MAG dataset were converted into a grid format, and glacial lakes in the GSW were further extracted using the range of glacier buffer (10 km). Hi-MAG detected more glacial lakes in the Himalaya region, Eastern Hindu Kush, and Tien Shan, and fewer in Eastern Pamir and Western Kunlun Shan. Fig. A6 illustrates the differences between our Hi-MAG glacial lake results and the GSW-derived lake area for the whole HMA region.

The glacial lake area observed in our lake dataset in the Eastern Pamir and Western Kunlun Mountains does not conform to the mapped surface water in the GSW for these sub-regions. While there are numerous glacial lakes from an open water perspective, actually part of them are river segments. Additionally, the Himalaya, Eastern Hindu Kush, and some other Tien Shan areas host thousands of glacial lakes that are not readily observable in the GSW dataset. Large discrepancies in mountainous glacial lake estimates preclude a significant consistency between GSW and our Hi-MAG lake data over the HMA region. The region with the highest consistency between GSW and Hi-MAG product is interior Tibet. There is little agreement for Tien Shan, where the weather is rainy and snowy in the region above 3000 m, and large quantities of ancient glacial deposits have accumulated. Here, glacial lakes are characterized by small sizes, and due to the influence of their source glaciers and lake beds, as well as the water depth and sediment inflow, glacial lakes appear to have heterogeneous reflectance in the images. Errors could exist in datasets produced by automated classification, but, as noted, we also conducted detailed manual editing, so we were not relying exclusively on automatic classification. The Karakoram region seems to have fewer glacial lakes in our estimate, owing to the overestimation of surface water on debris-covered glaciers in the GSW dataset.

The low agreement between our Hi-MAG glacial-lake data and the GSW data is mainly due to its lack of systematic glacial lake inventory and mapping capabilities. The lake dynamics and differing climate contexts within HMA may also lead to inconsistencies between the sub-regions. Hi-MAG might have made better use of the optimum satellite imaging season to map glacial lakes, potentially resulting in more complete mapping by avoiding conditions such as periods of lake ice that may confound mapping.

**6.2 Known issues and planned improvements**

There are several important issues and limitations to the datasets produced and the methods used within this study that are important to highlight to potential users. (i) Bodies of water smaller than nine connected pixels (e.g., $1 \times 9$ pixels or $3 \times 3$ pixels, corresponding to $30 \times 270$ m or $90 \times 90$ m, respectively) and those obscured by frozen water surface or loose moraines or hidden by terrain shadows were not included. Broken floating ice or isolated moraines that stood in open water for some time were mapped. Supraglacial lakes such as melt ponds developed on the surface of glaciers present particular challenges because of their small size and highly dynamic properties. Most supraglacial lakes are transient or seasonal, or at least fluctuate seasonally, as they commonly drain and may refill. In fact, this short-duration seasonal water is in general more likely to be underestimated because of temporal discontinuities in the archive and gaps caused by persistent cloud

cover. (ii) The spatial and temporal information reported in the Landsat dataset used in this study complements that acquired in the past. Nevertheless, the biggest limitation to glacial lake mapping from these data are undoubtedly the geographic and temporal discontinuities of the Landsat archive itself. Historical data over the entire HMA before 2008 can be recovered partly from the Landsat 4 TM/MSS, Landsat 5 TM, Landsat 7 ETM+, and partly from SPOT, and other satellite systems, etc., although data access is not always at the full, free, and open level of Landsat. In this regard, ASTER is freely accessible and has a higher resolution than Landsat, but its temporal coverage is very limited in most of HMA. Other Landsat-like moderate resolution multi-spectral sources could be also used to improve and extend the temporal sampling. For example, the European Space Agency's Sentinel 2a satellite launched in 2015 and provides optical imagery at 10-m resolution (Wang et al., 2020a; Yu et al., 2020), which will benefit future research combing all available satellite observations with GEE cloud computing power would make long-term monitoring of changes to HMA's glacial lakes and inland waters possible.

## 7 Data availability

The Hi-MAG database is distributed under the Creative Commons Attribution 4.0 License. The data can be downloaded from the data repository Zenodo at https://doi.org/10.5281/zenodo.4275164 (Chen et al., 2020).

## 8 Conclusions

In conclusion, the Hi-MAG dataset and others have used Earth observation satellite data, especially Landsat imagery, to provide a more consistent delineation of large-scale glacial lake changes. Some remote-sensed glacial lake mapping methods have enabled local-scale area estimation or spatial representation of lake extent and change. Such methods result in relatively good performance for lake areas that remain clear and show homogeneous reflectance in the image, but do not allow for continental-scale mapping of glacial lakes that have spectral interference from other objects such as glaciers, snow, clouds, turbidity, and the sedimentation characteristics of the glacial lake itself, or the atmospheric interference and terrain effects. Automated methods for the extraction of glacial lakes over large-scale areas have been further developed in this work. However, visual interpretation and manual editing is still an effective way to ensure high accuracy of lake inventories and append attributed information for further analysis. Based on an error of ±1 pixel on the lake boundary, the area uncertainty of each glacial lake ranges from 0.30% to 50% for the years between 2008 and 2017, and there is a mean area uncertainty of 17% in the entire HMA region.

Mapping of glacial lakes across the Tibetan Plateau and adjoining ranges reveals a complex pattern of lake occurrence and growth/shrinkage. During the past ten years, 2755 glacial lakes with a total area of 90.14 km$^2$ were increased in the HMA region. Proglacial lakes contributed 62.87% of that increase. We found that most areas in HMA have experienced rapid expansions, Central and Eastern Himalaya and Central Tien Shan showed the most lake area increases (up to +0.94 km$^2$ a$^{-1}$). Negative area changes were observed in the Eastern Tien Shan, Eastern Hindu Kush, Hengduan Shan, and Eastern Tibetan Mountains. The number of lakes grew very rapidly above 4400 m a.s.l., and proglacial lake growth is proceeding at high elevations of above 4900 m, but glacier retreat and lake disconnections are also starting to occur at higher elevations, causing the number and area of both classes to increase. At low elevations, few glaciers remain where proglacial lakes can form, and already detached lakes lack growth mechanisms. Overall, continued growth of glacial lakes can be expected, particularly where large debris-covered tongues remain.

The freely downloadable, detailed Hi-MAG dataset can also be used in future studies to provide a sound and consistent basis on which to quantify critical relationships and processes in HMA, including glacier–climate–lake interactions, glacio-

hydrologic models, GLOFs and potential downstream risks, and water resources.

**Appendix A**

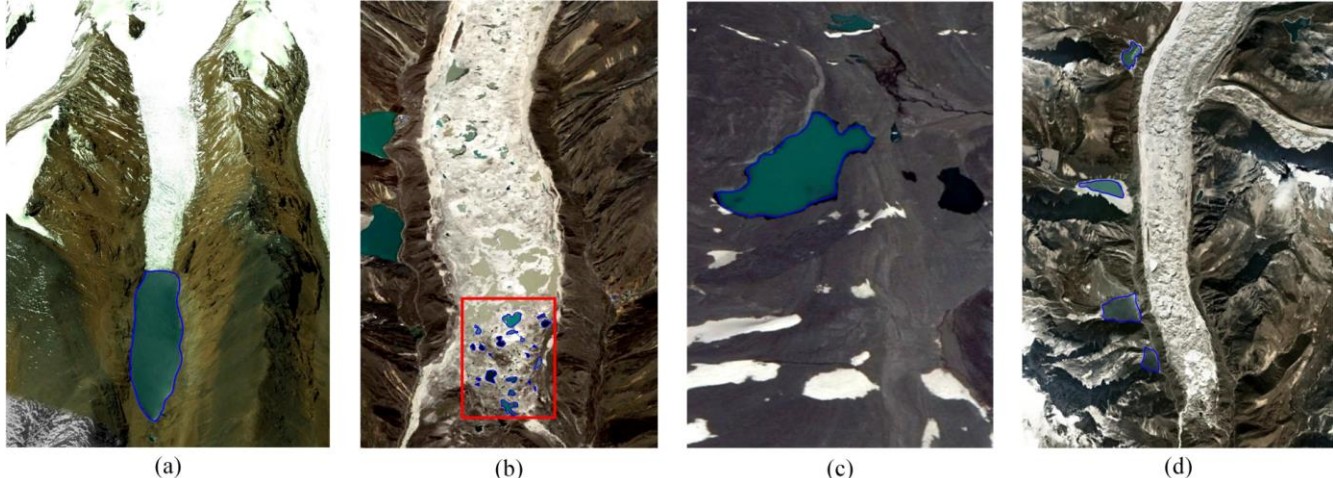

**Fig. A1.** Examples of the various types of glacial lake found in the HMA region: **(a)** proglacial lakes, which are connected to the parent glacier and usually impounded by a debris dam (usually a moraine or ice-cored moraine); **(b)** supraglacial lakes (denoted by the red rectangle), which develop on the glacier surface; **(c)** unconnected glacial lakes; and **(d)** ice-marginal lakes that are distributed on the edge of a glacier. Background images were acquired from © Google Earth, and were obtained in 2009, 2011, 2012, and 2014, respectively. Glacial lake outlines for each type are shown in blue.

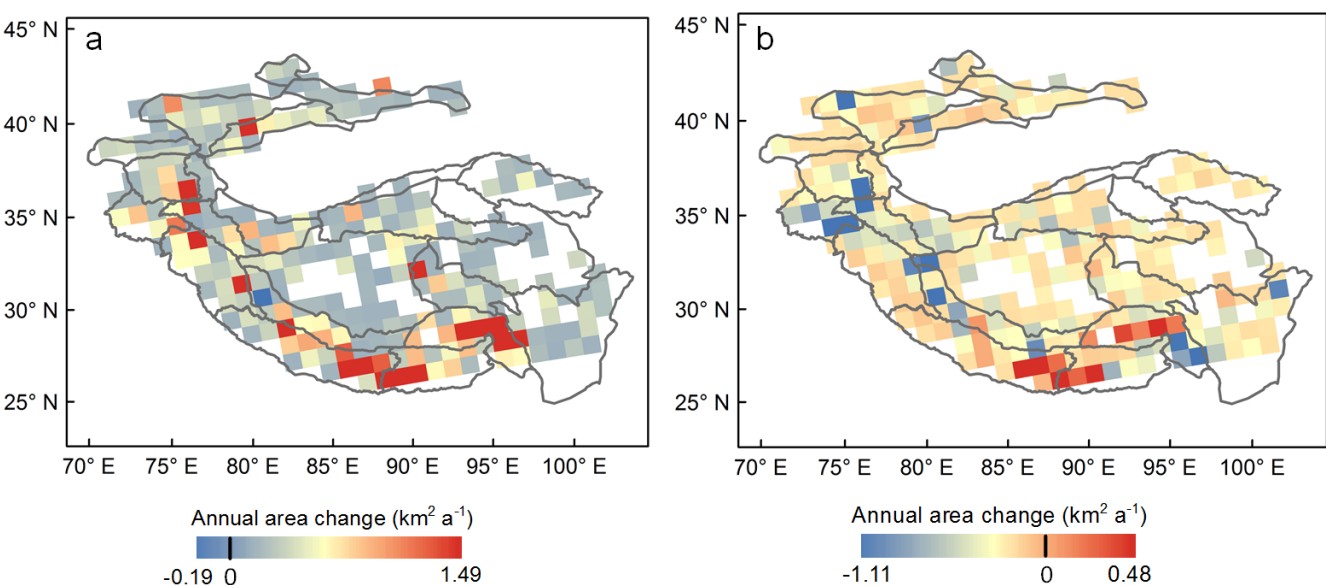

**Fig. A2.** Annual changes in lake area between 2008 and 2017 on a $1° × 1°$ grid. The **(a)** upper and **(b)** lower slopes represent the 90% confidence interval.

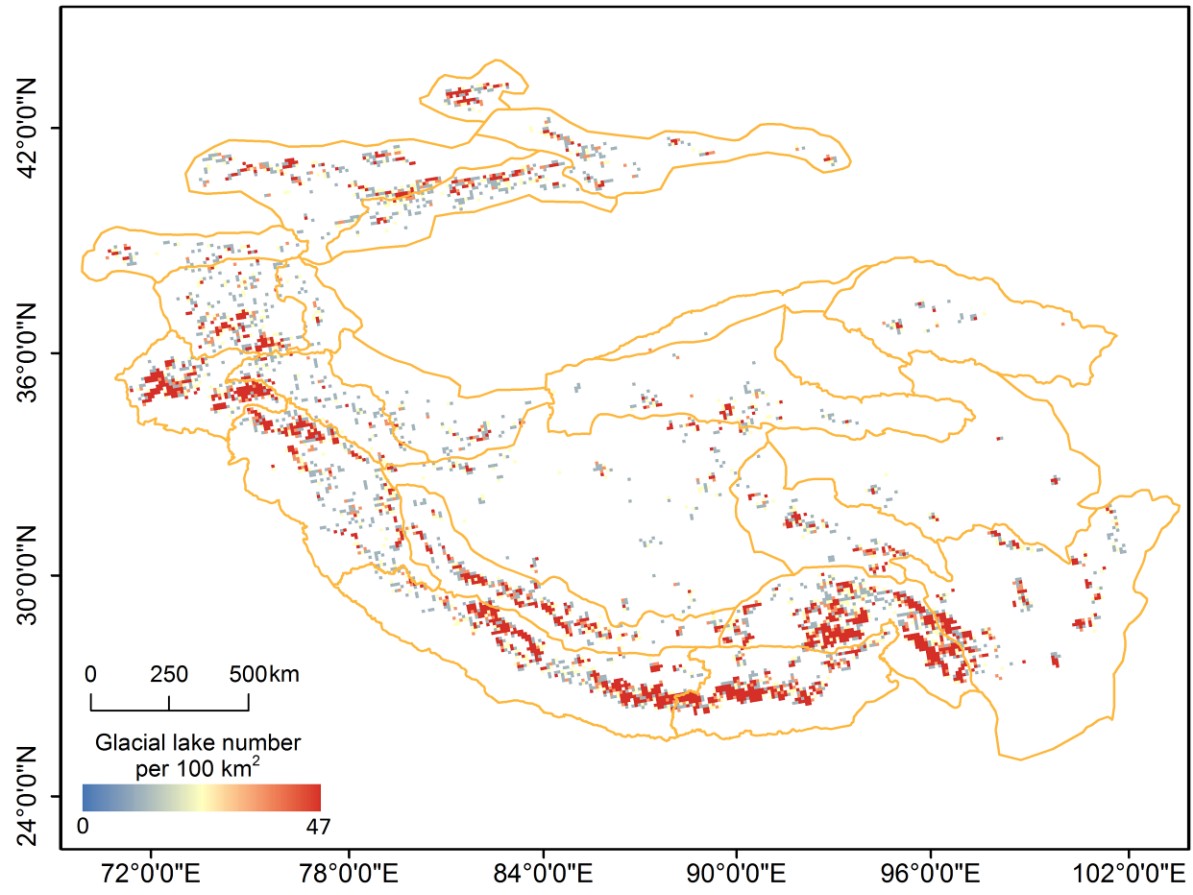

**Fig. A3.** Density (number per 100 km$^2$) distribution of glacial lakes in 2017.

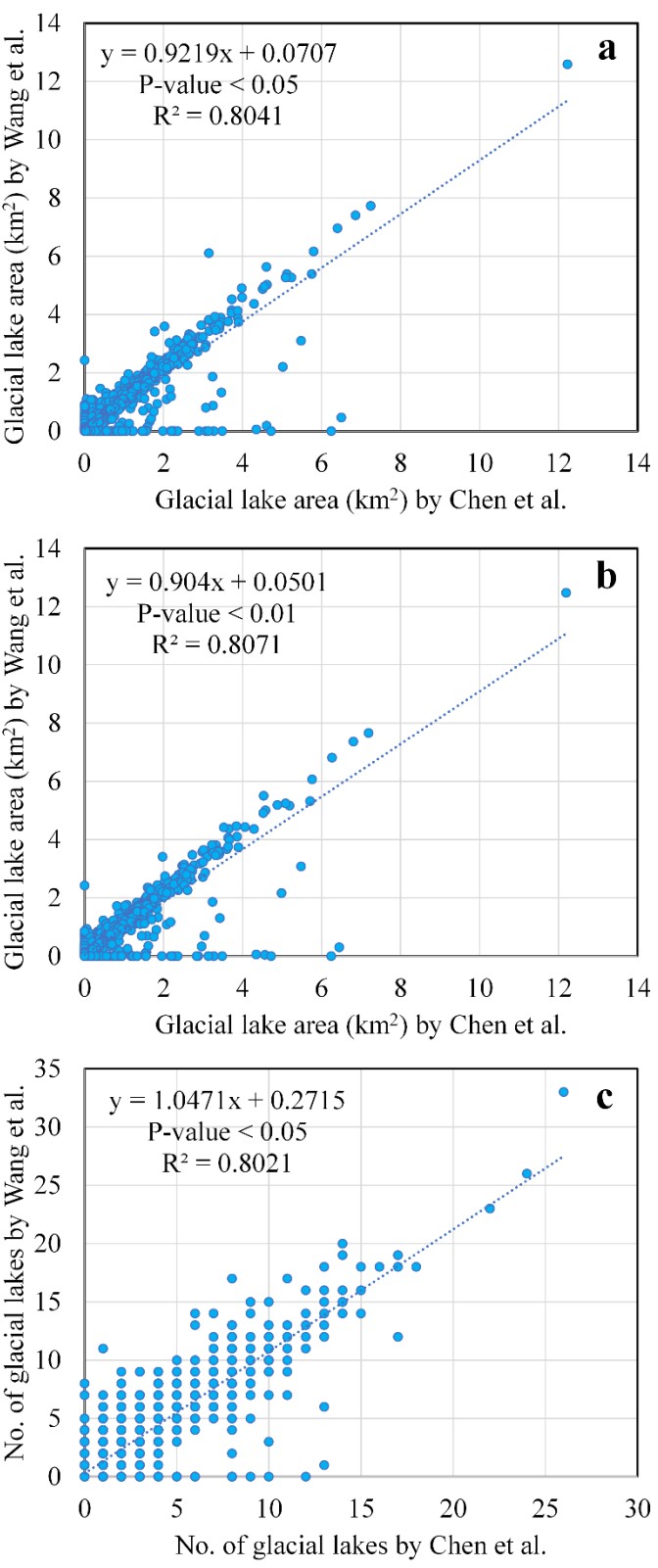

**Fig. A4.** Comparison of the results of **(a)** total glacial lake area, **(b)** areas of glacial lakes larger than 0.04 km$^2$, and **(c)** number of glacial lakes larger than 0.04 km$^2$ summed over a $0.1° \times 0.1°$ grid between the Hi-MAG database and the inventory of Wang et al. (2020).

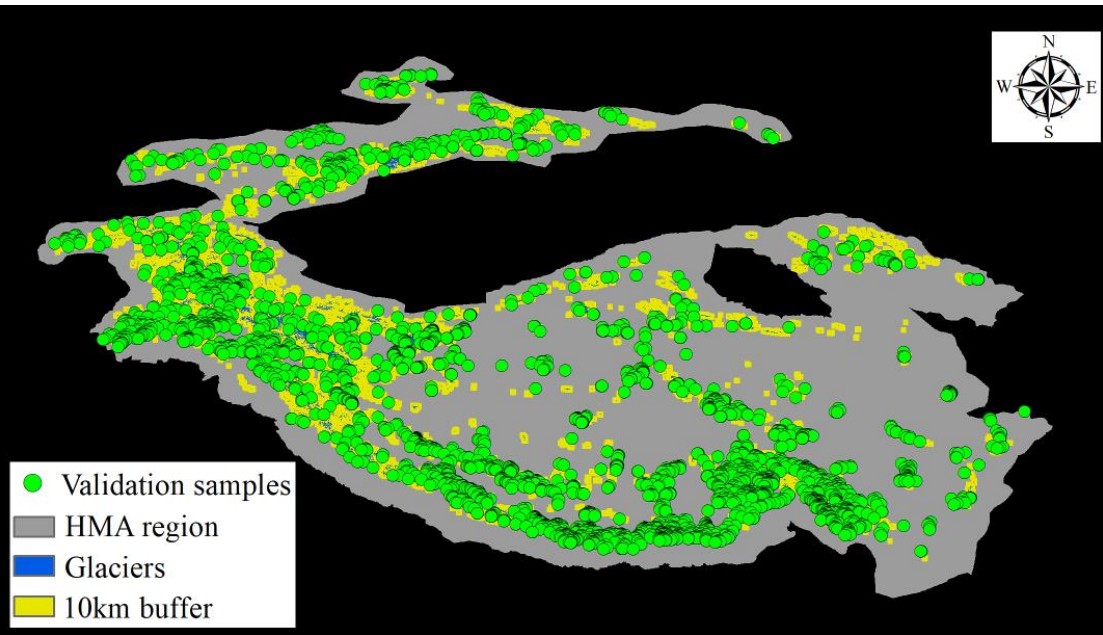

**Fig. A5.** Distribution of validation samples selected using stratified random sampling. Blue polygons are glacier outlines taken from the Randolph Glacier Inventory (RGI v5.0), the Second Chinese Glacier Inventory (CGI2) and the GAMDAM inventory. Yellow polygons refer to buffer areas within 10 km of glacier terminals.

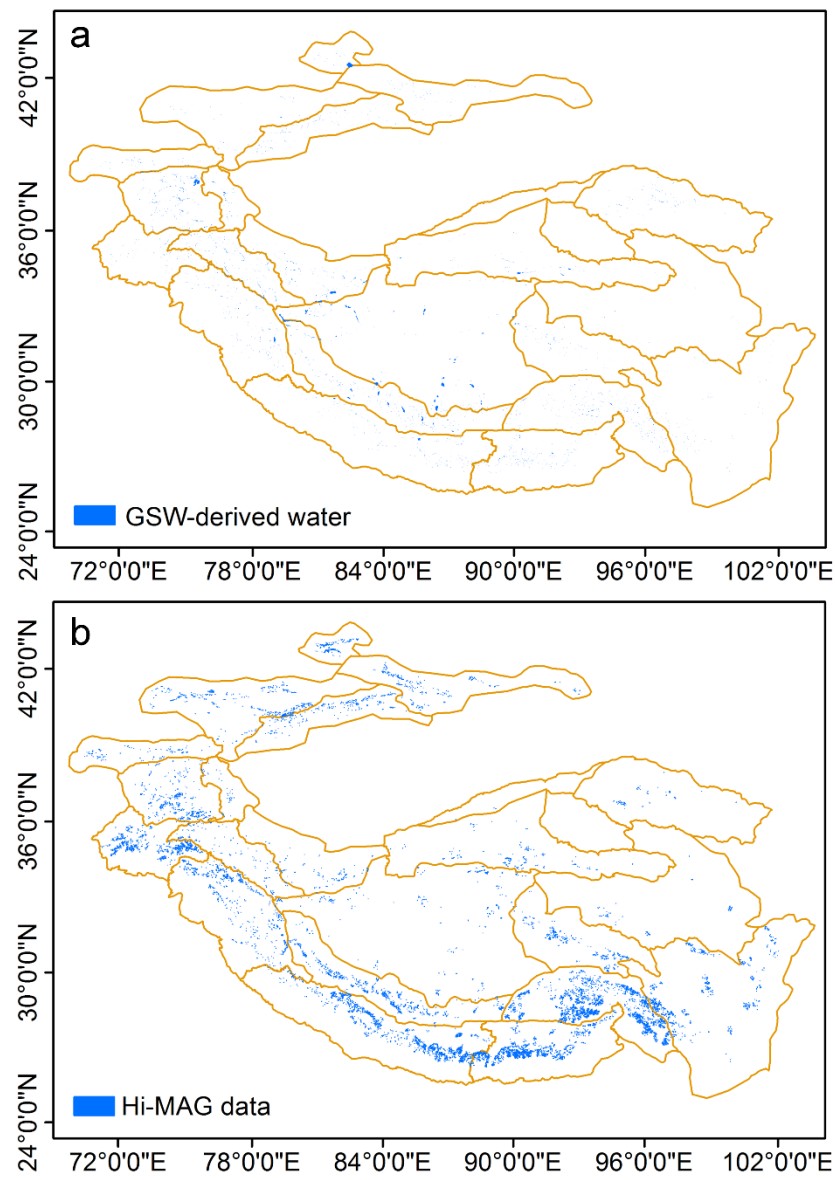

**Fig. A6.** Comparison of the glacial lakes measured in the global maps as in the **(a)** Pekel et al. (2016) and **(b)** our Hi-MAG data.

**Table A1.** Mountain-wide glacial lake number and area per year, and total loss/gain from 2008 to 2017. The unit of area is km².

| Mountain range | 2008 | | 2009 | | 2010 | | 2011 | | 2012 | | 2013 | | 2014 | | 2015 | | 2016 | | 2017 | | Total gain/loss (2008–2017) | |
|---|---|---|---|---|---|---|---|---|---|---|---|---|---|---|---|---|---|---|---|---|---|---|
| | No. | Area | No. | Area | No. | Area | No. | Area | No. | Area | No. | Area | No. | Area | No. | Area | No. | Area | No. | Area | No. | Area |
| Eastern Hindu Kush | 1227 | 73.63 | 856 | 61.40 | 1113 | 68.91 | 1036 | 67.94 | 1015 | 59.80 | 1206 | 67.90 | 1179 | 71.30 | 1172 | 66.36 | 1260 | 65.77 | 1399 | 70.83 | 172 | −2.80 |
| Western Himalaya | 894 | 77.65 | 856 | 70.23 | 756 | 71.25 | 747 | 71.38 | 746 | 68.95 | 779 | 76.00 | 874 | 82.76 | 823 | 78.33 | 999 | 79.51 | 1005 | 87.96 | 111 | 10.31 |
| Eastern Himalaya | 1634 | 164.12 | 1667 | 163.10 | 1675 | 166.49 | 1836 | 172.87 | 1859 | 171.67 | 1910 | 176.54 | 1943 | 174.93 | 2159 | 190.96 | 1954 | 179.09 | 1943 | 179.80 | 309 | 15.68 |
| Central Himalaya | 1312 | 182.54 | 1697 | 179.04 | 1577 | 188.35 | 1847 | 195.77 | 1782 | 195.56 | 1728 | 196.72 | 1850 | 198.52 | 2049 | 206.80 | 2096 | 206.41 | 2182 | 209.63 | 870 | 27.09 |
| Karakoram | 167 | 18.89 | 214 | 16.78 | 206 | 17.91 | 159 | 14.90 | 152 | 13.55 | 182 | 14.76 | 163 | 15.41 | 182 | 17.02 | 250 | 18.26 | 219 | 18.60 | 52 | −0.29 |
| Western Pamir | 481 | 75.53 | 486 | 77.64 | 526 | 83.27 | 548 | 80.86 | 495 | 79.17 | 537 | 79.32 | 550 | 81.20 | 557 | 83.96 | 570 | 81.76 | 624 | 75.82 | 143 | 0.29 |
| Eastern Pamir | 38 | 5.06 | 45 | 5.69 | 48 | 5.81 | 43 | 4.99 | 56 | 5.75 | 50 | 5.43 | 50 | 5.26 | 49 | 5.65 | 58 | 5.66 | 56 | 5.81 | 18 | 0.75 |
| Pamir Alay | 124 | 10.82 | 79 | 9.76 | 100 | 10.93 | 132 | 11.79 | 129 | 11.60 | 127 | 11.74 | 128 | 11.66 | 137 | 11.79 | 130 | 11.54 | 131 | 12.10 | 7 | 1.28 |
| Northern/Western Tien Shan | 474 | 36.16 | 358 | 37.20 | 499 | 36.87 | 541 | 41.71 | 522 | 36.04 | 518 | 36.98 | 551 | 46.06 | 530 | 38.28 | 512 | 37.38 | 626 | 40.91 | 152 | 4.75 |
| Central Tien Shan | 307 | 29.19 | 241 | 29.43 | 305 | 32.65 | 334 | 35.44 | 340 | 35.41 | 335 | 32.36 | 333 | 36.07 | 337 | 35.51 | 441 | 35.97 | 471 | 36.69 | 164 | 7.50 |
| Eastern Tien Shan | 247 | 15.83 | 230 | 17.24 | 241 | 15.99 | 245 | 15.64 | 251 | 15.75 | 259 | 16.15 | 250 | 16.07 | 297 | 17.31 | 241 | 17.99 | 245 | 18.31 | −2 | 2.48 |
| Western Kunlun Shan | 112 | 40.27 | 134 | 38.30 | 121 | 36.92 | 119 | 41.41 | 111 | 39.84 | 136 | 37.09 | 108 | 37.62 | 120 | 41.62 | 126 | 40.54 | 132 | 44.71 | 20 | 4.44 |
| Eastern Kunlun Shan | 180 | 11.92 | 193 | 12.83 | 265 | 20.62 | 237 | 16.46 | 232 | 15.74 | 246 | 15.90 | 244 | 16.72 | 290 | 18.49 | 255 | 17.95 | 248 | 16.90 | 68 | 4.98 |
| Gangdise Mountains | 852 | 99.15 | 908 | 96.46 | 848 | 104.24 | 886 | 96.88 | 961 | 102.48 | 895 | 95.75 | 954 | 96.28 | 991 | 94.64 | 1053 | 96.37 | 1116 | 99.83 | 264 | 0.68 |
| Hengduan Shan | 909 | 62.25 | 1077 | 64.70 | 967 | 65.38 | 867 | 61.29 | 927 | 62.75 | 1064 | 66.53 | 948 | 63.82 | 1023 | 65.45 | 942 | 62.93 | 961 | 63.35 | 52 | 1.10 |
| Tibetan Interior Mountains | 335 | 52.96 | 308 | 44.76 | 349 | 47.59 | 334 | 46.42 | 318 | 46.94 | 334 | 47.73 | 313 | 46.17 | 335 | 48.12 | 318 | 44.74 | 315 | 47.97 | −20 | −4.99 |
| Eastern Tibetan Mountains | 57 | 12.74 | 86 | 13.74 | 86 | 14.39 | 107 | 15.38 | 84 | 13.65 | 100 | 14.35 | 93 | 14.93 | 76 | 13.58 | 85 | 14.53 | 66 | 13.60 | 9 | 0.86 |
| Tanggula Shan | 468 | 39.45 | 318 | 40.99 | 344 | 44.28 | 327 | 43.24 | 347 | 42.17 | 363 | 43.40 | 376 | 44.97 | 461 | 45.53 | 372 | 45.94 | 363 | 44.97 | −105 | 5.52 |
| Qilian Shan | 82 | 7.96 | 78 | 8.67 | 91 | 9.68 | 77 | 9.23 | 76 | 9.19 | 85 | 9.36 | 90 | 10.38 | 86 | 9.90 | 80 | 9.99 | 67 | 8.97 | −15 | 1.01 |
| Dzhungarsky Alatau | 290 | 13.82 | 218 | 11.57 | 233 | 11.75 | 272 | 12.60 | 259 | 12.48 | 269 | 12.68 | 267 | 12.90 | 205 | 11.80 | 240 | 12.44 | 264 | 12.80 | −26 | −1.02 |
| Nyainqêntanglha | 2401 | 275.33 | 2647 | 255.23 | 2614 | 272.95 | 2539 | 273.28 | 2594 | 266.60 | 2856 | 275.46 | 2646 | 277.70 | 2977 | 280.35 | 2660 | 292.24 | 2911 | 285.68 | 510 | 10.35 |

**Table A2.** Mountain-wide annual glacial lake area from 2008 to 2017 for proglacial lakes and unconnected lakes. Supraglacial and ice-marginal lakes have relatively few coverage areas and are not listed in the table. The unit of area is km$^2$. Acronyms are used to represent the names of mountain ranges to save space (Eastern Hindu Kush (EHK), Western Himalaya (WH), Eastern Himalaya (EH), Central Himalaya (CH), Karakoram (K), Western Pamir (WP), Eastern Pamir (EP), Pamir Alay (PA), Northern/Western Tien Shan (N/WT), Central Tien Shan (CT),

Eastern Tien Shan (ET), Western Kunlun Shan (WK), Eastern Kunlun Shan (EK), Gangdise Mountains (G), Hengduan Shan (H), Tibetan Interior Mountains (TIM), Eastern Tibetan Mountains (ETM), Tanggula Shan (T), Qilian Shan (Q), Dzhungarsky Alatau (DA), Nyainqêntanglha (N)).

| | | EHK | WH | EH | CH | K | WP | EP | PA | N/WT | CT | ET | WK | EK | G | H | TIM | ETM | T | Q | DA | N | Total |
|---|---|---|---|---|---|---|---|---|---|---|---|---|---|---|---|---|---|---|---|---|---|---|---|
| Proglacial lakes | 2008 | 46.71 | 34.58 | 91.45 | 133.06 | 10.99 | 34.07 | 4.56 | 4.13 | 17.77 | 14.52 | 9.12 | 5.15 | 1.58 | 24.01 | 11.40 | 3.59 | 0.76 | 9.73 | 3.74 | 11.19 | 133.29 | 605.40 |
| | 2009 | 37.98 | 30.03 | 92.97 | 130.09 | 10.60 | 34.27 | 5.30 | 2.79 | 16.39 | 14.07 | 8.11 | 5.69 | 1.72 | 19.97 | 12.58 | 2.97 | 1.54 | 10.12 | 4.71 | 9.04 | 122.95 | 573.90 |
| | 2010 | 41.68 | 29.73 | 95.11 | 137.49 | 10.55 | 35.72 | 5.02 | 3.57 | 18.09 | 16.70 | 9.18 | 5.73 | 5.66 | 22.27 | 13.55 | 4.72 | 1.68 | 11.52 | 5.25 | 9.24 | 136.04 | 618.48 |
| | 2011 | 42.77 | 32.88 | 100.58 | 143.53 | 9.70 | 34.82 | 4.62 | 4.04 | 20.27 | 15.64 | 8.66 | 4.73 | 3.69 | 20.76 | 12.14 | 3.80 | 1.74 | 10.63 | 4.71 | 10.18 | 136.69 | 626.59 |
| | 2012 | 37.32 | 31.30 | 99.54 | 143.02 | 9.00 | 34.87 | 5.12 | 3.91 | 18.58 | 16.76 | 8.83 | 4.69 | 3.66 | 21.78 | 12.83 | 3.73 | 1.74 | 10.60 | 4.66 | 10.02 | 133.29 | 615.25 |
| | 2013 | 41.67 | 30.58 | 100.97 | 141.25 | 9.22 | 34.57 | 4.71 | 4.08 | 18.28 | 16.72 | 9.20 | 4.95 | 3.51 | 20.56 | 13.55 | 3.99 | 1.67 | 10.60 | 4.74 | 10.07 | 136.58 | 621.47 |
| | 2014 | 43.88 | 36.12 | 98.12 | 145.22 | 9.96 | 35.06 | 4.71 | 4.15 | 20.33 | 17.37 | 8.88 | 5.04 | 4.40 | 21.88 | 12.91 | 4.38 | 1.75 | 10.68 | 5.01 | 10.21 | 136.95 | 637.00 |
| | 2015 | 40.49 | 33.88 | 108.45 | 151.17 | 10.68 | 37.63 | 5.03 | 4.02 | 18.98 | 16.26 | 9.53 | 5.08 | 5.01 | 22.66 | 13.53 | 4.44 | 1.62 | 11.30 | 4.67 | 9.43 | 138.12 | 652.00 |
| | 2016 | 40.52 | 34.20 | 103.31 | 150.59 | 10.92 | 36.13 | 5.02 | 3.83 | 18.12 | 17.63 | 8.31 | 4.67 | 4.03 | 23.29 | 13.03 | 4.08 | 1.74 | 10.75 | 4.73 | 9.99 | 147.83 | 652.73 |
| | 2017 | 43.53 | 35.52 | 103.75 | 152.52 | 11.33 | 37.60 | 5.16 | 4.03 | 20.07 | 18.56 | 8.52 | 5.36 | 4.21 | 23.31 | 13.59 | 4.41 | 1.69 | 10.49 | 4.71 | 10.15 | 143.58 | 662.07 |
| Unconnected lakes | 2008 | 26.82 | 41.86 | 70.98 | 43.82 | 6.33 | 41.24 | 0.47 | 6.70 | 18.38 | 14.56 | 6.69 | 35.10 | 10.35 | 74.83 | 50.60 | 49.08 | 11.98 | 29.50 | 4.23 | 2.57 | 140.42 | 686.53 |
| | 2009 | 23.35 | 39.37 | 69.41 | 45.41 | 4.81 | 43.00 | 0.39 | 6.98 | 20.79 | 15.36 | 9.07 | 32.50 | 11.07 | 76.40 | 52.10 | 41.87 | 12.27 | 30.87 | 3.97 | 2.49 | 131.63 | 673.04 |
| | 2010 | 27.08 | 40.69 | 70.78 | 46.96 | 5.64 | 47.24 | 0.80 | 7.37 | 18.78 | 15.87 | 6.79 | 31.20 | 14.81 | 81.97 | 51.83 | 42.87 | 12.71 | 32.76 | 4.44 | 2.49 | 136.12 | 699.22 |
| | 2011 | 25.00 | 37.62 | 71.65 | 47.79 | 4.44 | 45.79 | 0.31 | 7.75 | 21.42 | 19.76 | 6.96 | 36.65 | 12.78 | 76.13 | 49.16 | 42.63 | 13.64 | 32.59 | 4.52 | 2.34 | 136.05 | 694.98 |
| | 2012 | 22.41 | 36.80 | 71.23 | 47.75 | 3.74 | 44.13 | 0.50 | 7.69 | 17.44 | 18.48 | 6.82 | 35.15 | 12.04 | 80.71 | 49.92 | 43.22 | 11.92 | 31.53 | 4.54 | 2.37 | 132.53 | 680.92 |
| | 2013 | 26.10 | 44.59 | 74.81 | 50.85 | 4.09 | 44.52 | 0.59 | 7.67 | 18.68 | 15.55 | 6.83 | 32.09 | 12.34 | 75.20 | 52.98 | 43.75 | 12.69 | 32.77 | 4.62 | 2.51 | 138.07 | 701.31 |
| | 2014 | 27.27 | 45.69 | 75.92 | 48.65 | 4.31 | 45.85 | 0.43 | 7.52 | 25.69 | 18.54 | 7.08 | 32.55 | 12.29 | 74.40 | 50.92 | 41.74 | 13.19 | 34.26 | 5.38 | 2.60 | 140.03 | 714.33 |
| | 2015 | 25.76 | 43.60 | 81.11 | 50.30 | 5.76 | 46.08 | 0.47 | 7.76 | 19.29 | 19.15 | 7.66 | 36.54 | 13.44 | 71.98 | 51.92 | 43.66 | 11.96 | 34.20 | 5.23 | 2.30 | 141.40 | 719.59 |
| | 2016 | 25.14 | 44.34 | 74.76 | 50.30 | 6.06 | 45.39 | 0.49 | 7.70 | 19.24 | 17.77 | 9.57 | 35.81 | 13.91 | 73.09 | 49.91 | 40.66 | 12.77 | 35.15 | 5.26 | 2.35 | 143.38 | 713.11 |
| | 2017 | 27.19 | 51.42 | 74.96 | 51.59 | 6.54 | 37.95 | 0.46 | 8.06 | 20.81 | 17.52 | 9.66 | 39.34 | 12.68 | 76.53 | 49.76 | 43.57 | 11.91 | 34.45 | 4.27 | 2.56 | 141.22 | 722.44 |

**Table A3.** Areas of different sizes of glacial lakes in 2017 for some regions with large area growth of rates. The unit of area is km$^2$.

| Lake grid ID (Mountain ranges) | 69 (N) | 116 (CH) | 274 (WH) | 71 (N) | 48 (H) | 74 (N) | 72 (N) | 14 (EH) | 13 (EH) | 39 (CH) | 15 (EH) |
|---|---|---|---|---|---|---|---|---|---|---|---|
| ≤0.01 km$^2$ | 0.18 | 0.28 | 0.20 | 0.16 | 0.22 | 0.16 | 0.17 | 0.23 | 0.16 | 0.32 | 0.33 |
| 0.01 km$^2$–0.02 km$^2$ | 0.85 | 1.51 | 1.29 | 0.71 | 1.43 | 1.08 | 1.37 | 1.45 | 1.45 | 1.49 | 2.46 |
| 0.02 km$^2$–0.04 km$^2$ | 1.69 | 2.16 | 2.22 | 1.79 | 3.24 | 2.09 | 2.29 | 2.24 | 2.06 | 2.72 | 4.14 |
| 0.04 km$^2$–0.08 km$^2$ | 1.78 | 3.19 | 2.98 | 2.20 | 5.30 | 3.38 | 4.45 | 2.77 | 2.69 | 3.66 | 7.16 |
| 0.08 km$^2$–0.16 km$^2$ | 1.91 | 5.38 | 3.87 | 2.86 | 4.81 | 4.03 | 5.06 | 3.75 | 4.33 | 5.00 | 13.16 |
| 0.16 km$^2$–0.32 km$^2$ | 1.81 | 4.53 | 2.23 | 2.76 | 4.62 | 5.55 | 5.81 | 2.91 | 3.90 | 5.66 | 11.62 |
| 0.32 km$^2$–0.64 km$^2$ | 1.01 | 5.37 | 1.77 | 1.79 | 3.88 | 1.75 | 3.81 | 5.72 | 3.99 | 7.13 | 12.37 |
| 0.64 km$^2$–1.28 km$^2$ | 0.00 | 2.94 | 0.00 | 1.38 | 2.82 | 2.96 | 4.43 | 0.96 | 7.10 | 8.97 | 7.74 |
| ≥1.28 km$^2$ | 0.00 | 7.22 | 4.19 | 0.00 | 11.46 | 3.17 | 2.59 | 6.07 | 1.40 | 6.06 | 12.00 |
| Total area (km$^2$) | 9.22 | 32.58 | 18.76 | 13.66 | 37.76 | 24.17 | 29.99 | 26.10 | 27.09 | 41.00 | 70.96 |
| Total area (≤0.16 km$^2$) | 6.41 | 12.52 | 10.57 | 7.72 | 14.99 | 10.74 | 13.35 | 10.45 | 10.69 | 13.18 | 27.24 |
| % of Total area (≤0.16 km$^2$) | 69.47 | 38.43 | 56.32 | 56.56 | 39.70 | 44.45 | 44.52 | 40.03 | 39.47 | 32.15 | 38.39 |
| Annual area increase (km$^2$ a$^{-1}$) | 0.23 | 0.28 | 0.28 | 0.29 | 0.32 | 0.41 | 0.42 | 0.49 | 0.70 | 0.74 | 0.94 |

**Table A4.** Summary of correlation coefficients (*R*) for key lake topographic, geomorphic, and climatological parameters, calculated within $1° \times 1°$ grid cells across HMA. Correlation coefficients are bold where $p < 0.05$; (*) indicates $p < 0.01$.

| | Lake area (2008) | Lake area (2017) | Lake change (2008–2017) | Glacier (gl.) area ^ | Debris-covered gl. area | Total gl. length | Mean gl. slope | Mean gl. elevation | Temperature change^^ 1979–2017 | Precipitation change 1979–2017 |
|---|---|---|---|---|---|---|---|---|---|---|
| Lake area (2008) | 1.00 | | | | | | | | | |
| Lake area (2017) | **0.99*** | 1.00 | | | | | | | | |
| Lake change (2008–2017) | **0.82*** | **0.87*** | 1.00 | | | | | | | |
| Glacier (gl.) area | **0.23*** | **0.24*** | **0.22*** | 1.00 | | | | | | |
| Debris-covered gl. area | **0.35*** | **0.36*** | **0.34*** | **0.85*** | 1.00 | | | | | |
| Total gl. length | **0.32*** | **0.32*** | **0.28*** | **0.90*** | **0.85*** | 1.00 | | | | |
| Mean gl. Slope | 0.07 | 0.07 | 0.05 | 0.02 | **0.18** | 0.06 | 1.00 | | | |
| Mean gl. Elevation | 0.12 | 0.14 | **0.17** | 0.11 | 0.00 | 0.05 | **−0.28*** | 1.00 | | |
| Temperature change 1979–2017 | −0.09 | −0.07 | 0.10 | **−0.17** | **−0.25*** | **−0.27*** | 0.00 | 0.07 | 1.00 | |
| Precipitation change | −0.03 | −0.01 | 0.09 | −0.13 | **−0.16** | **−0.15** | **−0.18*** | **0.15** | **0.16** | 1.00 |

545    ^ Glacier data are derived from the Randolph Glacier Inventory (RGI Consortium, 2017), except for debris cover (after Scherler et al., 2018). Climate data are for ERA Interim.

^^ ERA-Interim near surface temperature and precipitation fields for the period 1979–2017 were obtained from the KNMI climate explorer (https://climexp.knmi.nl).

**Table A5.** Regional summary of key topographic, geomorphic, and climatological parameters compared to proglacial and supraglacial lake area in 2017. Correlation coefficients are bold where $p < 0.05$; (*) indicates $p < 0.01$.

| Region | Total area (km²) | Lake area (km²) | Glacier (gl.) area (km²) | Debris-covered gl. area (km²) | Total gl. length (km) | Mean gl. slope (°) | Mean gl. elevation (m) | Temperature change 1979–2017 (°C/century) | Precipitation change 1979–2017 |
|---|---|---|---|---|---|---|---|---|---|
| Central Himalaya | 254886 | 155.7 | 8678 | 1567 | 10669 | 26 | 5542 | 2.77 | −0.25 |
| Central Tien Shan | 105456 | 19.0 | 7270 | 842 | 7415 | 27 | 4181 | 1.35 | 0.05 |
| Dzhungarsky Alatau | 37542 | 10.3 | 521 | 18 | 978 | 24 | 3615 | 1.85 | 0.74 |
| Eastern Himalaya | 164785 | 104.7 | 2838 | 357 | 3614 | 24 | 5484 | 3.26 | −0.84 |
| Eastern Hindu Kush | 95404 | 43.6 | 2938 | 609 | 5062 | 25 | 4856 | 0.08 | −0.86 |
| Eastern Kunlun Shan | 256729 | 4.2 | 2995 | 45 | 3384 | 24 | 5389 | 3.60 | 0.06 |
| Eastern Pamir | 39605 | 5.2 | 2118 | 291 | 2364 | 27 | 5064 | 3.42 | −0.50 |
| Eastern Tibetan Mountains | 333123 | 1.8 | 312 | 12 | 483 | 24 | 5345 | 3.55 | 0.73 |
| Eastern Tien Shan | 140900 | 8.7 | 2332 | 193 | 3977 | 28 | 3974 | 2.65 | 0.17 |
| Gangdise Mountains | 154884 | 23.2 | 1271 | 80 | 2570 | 24 | 5892 | 2.42 | 0.33 |
| Hengduan Shan | 372649 | 13.6 | 1281 | 212 | 2048 | 23 | 5278 | 2.24 | −0.13 |
| Karakoram | 83644 | 11.7 | 21474 | 2013 | 16460 | 31 | 5399 | 2.37 | −0.35 |
| Northern/Western Tien Shan | 187275 | 20.1 | 2262 | 223 | 4138 | 23 | 3943 | 3.22 | −0.36 |

| | | | | | | | | |
|---|---|---|---|---|---|---|---|---|
| Nyainqêntanglha | 172746 | 144.6 | 7047 | 1011 | 8710 | 25 | 5282 | 2.37 | −1.00 |
| Pamir Alay | 71845 | 4.0 | 1847 | 319 | 3441 | 25 | 4109 | 3.88 | −0.27 |
| Qilian Shan | 201699 | 4.7 | 1598 | 30 | 2588 | 26 | 4847 | 4.07 | 0.51 |
| Tanggula Shan | 145064 | 10.6 | 1841 | 84 | 1893 | 21 | 5521 | 3.34 | 0.46 |
| Tibetan Interior Mountains | 526111 | 4.4 | 3815 | 59 | 4179 | 23 | 5927 | 2.64 | 0.31 |
| Western Himalaya | 189494 | 36.3 | 7986 | 1149 | 11974 | 24 | 5180 | 1.93 | −1.24 |
| Western Kunlun Shan | 123388 | 5.3 | 8457 | 159 | 8108 | 26 | 5642 | 3.22 | −0.55 |
| Western Pamir | 109239 | 37.9 | 8417 | 1118 | 11640 | 27 | 4844 | 1.61 | 0.08 |
| Lake area: Correlation Coefficient ($R$) | | | 0.21 | **0.50** | 0.36 | 0.01 | 0.23 | −0.17 | **−0.49** |
| Excl. Karakoram | | | **0.49** | **0.72*** | **0.52** | 0.10 | 0.25 | −0.18 | **−0.50** |

**Table A6.** Statistical results of stratified random sampling.

| Stratum | Total pixel number | Total area (km²) | No. of samples | No. of non-glacial lake samples | No. of glacial lake samples | No. of ambiguous samples |
|---|---|---|---|---|---|---|
| C0W0 | 2,022,448,650 | 1,820,203.78 | 1300 | 1215 | 37 | 48 |
| C0W1 | 925,449 | 832.90 | 700 | 307 | 362 | 31 |
| C1W0 | 814,196 | 732.77 | 700 | 21 | 678 | 1 |
| C1W1 | 1,611,668 | 1450.50 | 1300 | 25 | 1260 | 15 |

**Author contributions.** FC: conceptualization, methodology, lake evolution analysis, project administration, resources, and writing; MMZ: conceptualization, methodology, lake evolution analysis, validation, and writing; HDG: funding acquisition, supervision, and writing; SA: methodology, climate and debris-cover analysis, validation, and writing; JSK: analysis, interpretation, and writing; UH: writing and interpretation; CSW: writing.

**Competing interests.** The authors declare that they have no conflict of interest.

**Acknowledgments.** We thank T. Bolch, and D. Shugar for their contributions to this project in its stages of development; L. Wang, S. G. Xu, Z. Y. Lin, H. Zhao, Y. H. Z. He, T. C. Shan, Z. W. Xu, N. Wang, Z. Z. Yin, and J. X. Wang for the cross-validations of the data that were so integral to this project.

**Financial support.** This work was supported by the Strategic Priority Research Program of the Chinese Academy of Sciences (XDA19030101), the International Partnership Program of the Chinese Academy of Sciences 565 (131211KYSB20170046/131C11KYSB20160061), and the National Natural Science Foundation of China (41871345). SA was supported by the EVOGLAC project under the Swiss National Science Foundation (IZLCZ2_169979/1).

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
