# Peer review of "Annual 30-meter Dataset for Glacial Lakes in High Mountain Asia from 2008 to 2017"

_Earth System Science Data, 2020_

## Referee Comment (RC1) · Anonymous Referee #1 · 18 Jun 2020

Chen et al. mapped glacier lakes in High Mountain Asia in annual time steps from 2008 to 2017 from Landsat imagery. They used an automated image segmentation algorithm to pre-classify >40,000 Landsat images between late spring and early fall. Ten experts manually refined the annual lake inventories. The authors further calculated trends of lake abundance and area in the entire study region and several subregions thereof, and show that the growth in lake area and number could be largely due to many new lakes that occupy the lower size spectrum. Hotspots of total lake growth are the Central Himalaya, Eastern Himalaya, and Western Himalaya. The authors also correlate the size-distribution of glacier lakes with a set of environmental predictors such as temperature change, debris cover, or glacier size, to explain where glacier lakes have formed in the landscape. Debris cover and trends in warming and precipitation

could help to explain the size-distribution and growth of glacier lakes, though regional variances make a more rigorous explanation elusive.

Chen et al. have compiled an important, and possibly the most extensive, data set of glacier lakes in the HMA to date. This inventory could be useful for scientists and practitioners in many disciplines interested in changes of the cryosphere of the HMA. Each year has >14,000 lake polygons, so that it is difficult for me to judge how accurately these lakes have been mapped. Yet, visually comparing the inventories of 2016 and 2017, I noticed that the 2017 data set misses >2 km$^2$ of lakes in a subset 0.3° x 0.2° large. The figure in the attachment shows 11 yellow polygons from 2016 that are not overlapped by blue polygons in 2017. Yet all these lakes are clearly visible in the Landsat scene from 2017-09-26, which is the basemap here. It is surprising that the 'ten trained experts' (L150-151) have not detected this error. Of course, my cross-check was not systematic, and it could be that I just accidentally came across one of very few undetected lakes in this study. Nevertheless, I strongly recommend that the authors again check the inventories for such issues, because a larger quantity of undetected lakes could in fact contribute to the calculated statistics, and the conclusions drawn in this study.

In any case, the tremendous effort that the authors have spent to map several tens of thousands of glacier lakes is, unfortunately, not reflected in the quality of the manuscript. I first wish to emphasize some major points in each section of this manuscript, and outline further details below.

The introduction is very brief and the many dozens of glacier lake inventories that have been compiled in this region before remain largely unmentioned. These inventories are also largely disregarded in the discussion though they could help to discuss the benefits and challenges remaining in this study.

The methods do not introduce the problem of ice-covered lakes that occurs with increasing elevation. Though the authors manually corrected for missing or misclassified
lakes, it remains unclear, how many lakes needed to be corrected manually after automatic classification, and how many lakes are possibly missing in this data base.

The text and the figures in the results (Chapter 5.1 and 5.2) show dozens of numbers and trends, regarding lake areas, abundance, and changes thereof, but remain without error bars or confidence intervals throughout. Yet the authors acknowledge that mapping errors for small lakes in particular can be large, so that it must be cautioned against interpreting these trends. The panels in Fig. 5, for example, show considerable variance, and it remains unclear whether the regression models used to calculate the trends, account for this variance.

In Chapter 5.3 and the discussion, the authors search for environmental predictors that could contribute to the (changing) size-distribution of glacier lakes. Yet many of these correlations, for example with glacier slope or elevation, are presented in tables, but remain untouched in the manuscript. Other correlations such as local glacier mass balances (e.g. Brun et al., 2017) are disregarded, though these could be useful in the light of shrinking glaciers and growing lakes. In general, the discussion does not go far beyond showing these correlations; discussion with previous studies that aimed to explain the size-distribution of glacier lakes remains very limited.

Throughout the manuscript the authors use subjective qualifiers such as 'small', 'large', or 'relatively' that could be systematically filled with the data that they have produced. The orthography, grammar, and style could still deserve much improvement, which is surprising to see given the number of co-authors involved in the writing part of this study.

These major points should be fully addressed in a revised manuscript. Below I detail line-by-line some more points of criticism. Some technical corrections are also included.

L17: 'Atmospheric warming' instead of 'climate change'?

L19: 'incomplete': Can this study claim completeness?

L21-22: 'rapid', 'moderate', 'large', 'faster': please be more specific. From annual lake inventories, it should be straightforward to calculate, for example, the total lake area, number, and absolute changes during the study period.

L23-24: 'Proglacial lake dominated areas'; 'unconnected lake dominated areas': unclear what and where these areas are.

L23: 'significant': how do the authors measure significance here?

L26: 'a main contributor': to? Increase in lake area?

L27: 'an overlooked element'. Not sure whether this can be called a novel finding, given that the prevalence of small lakes in glacier lake inventories has been emphasized before, e.g. in (Khadka et al., 2018; Nagai et al., 2017; Shukla et al., 2018)

L29: 'is' instead of 'are'?

L30: New sentence instead of comma?

L34: 'significant': how do the authors measure significance here?

L35: 'incompletely': same argument as in the abstract. Do the authors mean that glacier lakes had been mapped at large intervals?

L36-37: Please add a reference.

L37-39: The authors could use more suitable references to support their arguments: Salerno et al. used 10 m satellite images and have mapped lakes <0.001 km$^2$; Brun et al. have studied glaciers, not lakes; for the inventories with 'narrow geographic scope', it would be good have a reference for some of the major region such as Karakoram, Sikkim, Bhutan, Nepal, Tien Shan, Nyainqentanglha, etc., to stress this 'patchwork' of glacier lake inventories more clearly. The ~2003 inventory of ICIMOD (Maharjan et al., 2018) and the multi-temporal inventories by Zhang et al. (2015) and by Wang

et al. (2020) are not mentioned at all here, though these studies addresses many of the issues raised here. What about the study from Pekel et al. (2016), who mapped surface water at high resolution globally? In summary, the authors could acknowledge previous work more than in the present form, given that many teams of researchers had aimed to map glacier lakes in the entire HMA (or parts thereof) before...

L52: 'atmospheric warming' instead of 'climate warming'?

L45-56: What is the difference between i) and iv)? For example, given the situation of a lake that is dammed by a moraine, and its parent glacier calving into it: is this a lake 'usually connected to the glacial tongue and dammed by unconsolidated or ice-cemented moraines' (L46) or a lake 'bounded by a lateral moraine on one side and damming glacier ice on the other side'? Furthermore, I miss the category of purely glacier-dammed lakes, such as the one at Kyagar glacier (Round et al., 2017). Maharjan et al. (2018) list more than 20 ice-dammed lakes in the Himalayas.

L55-56: 'Alaska': reference missing

L57: 'for potential automatic mapping errors': The authors did not mention before that they mapped lakes automatically.

L58: What are 'systematic errors'?

L60: The chapter on the study area contains very limited information on the 'study area'. Could be expanded, including the climate and topography of the HMA. This could prepare readers for the observed variability of glacier lake abundance.

L62: 'etc': Please write out all regions; Fig 4a is mentioned before Fig 1.

L62: 'For this study': are there other studies where glacier lakes formed differently?

L63: 'within 10 km' instead of 'within a 10 km'?

L64: 'Approximately 40,481 Landsat series satellites': did the authors mean 'Landsat images from the three Landsat missions'? The term 'approximately' seems a bit odd

here, given this exact number.

L69: What do the authors mean by 'year-round'? Why did they not fill the years 2004, 2005, 2006, and 2007 with images from ETM+, similar to approach for 2012?

L73-74: 'in false detections of water' instead of 'in the false water detections'?

L72-73: 'were accurately classified but usually misclassified otherwise': not fully clear, please elaborate more clearly. Again, this sentence mentions classification, but any methodological background has not been given yet. The authors could consider to shift such sentences at appropriate locations down in the text.

L85-86: 'during stable seasons when lake extents are minimally affected by meteorological conditions and glacier runoff", which is 'from July to November': not sure whether this statement is true. Studies on supraglacial ponds (Miles et al., 2017b, 2017a) show that changes occur preferentially in that part of the year.

L89: 'most-monsoon': did the authors mean 'post-monsoon'?

L95: 'cloud score functions in GEE': did the authors make sure that this cloud score function works well in mountain headwaters with high snow and ice cover? This is a major challenge for cloud detection algorithms (Zhu and Woodcock, 2012).

L95: 'a partial Landsat scene': unclear – please reformulate.

L97: 'criteria' instead of 'criterions'?

L97: 'needs to be broaden': by how many months? Possibly into the next year? Previous studies (Maharjan et al., 2018; Veh et al., 2018) have shown that some lakes could be veiled by clouds and ice for years.

L98-100: Please reformulate this sentence, possibly splitting into two.

L104-106: Repetition, consider deleting.

L109: 'and SLC-off gaps'

L110: 'a more computationally efficient way': than?

L109-L111: I suggest to switch the order of these two sentences.

L112: 'slopes (larger than 10°)': How did the authors make sure that this threshold does not mask water bodies that have grown at steep glacier tongues since the SRTM DEM was shot in 2000?

L112: 'value less than 0.25': Unclear what this value means in physical quantities.

L113: The paper from Quincey is from 2007.

L115: Maybe 'acquired. Some' ?

L116: 'remain' instead of 'remains', and 'was' instead of 'were'?

L118: 'the relative stability': unclear, please reformulate.

L118: '0.0081 km$^2$': the reason for choosing this minimum mapping unit is not fully clear.

L127: 'polygon include' instead of 'vector includes'?

L128: 'distance to the nearest glacier terminus': how is this distance measured? Along the flow path or the Euclidian distance? What happens if there is no glacier upstream, given that the buffer around the glacier can extend into adjacent, yet glacier-free catchments?

L135-146: It remains unclear, how the authors calculated the perimeter of a given lake in a given year, if there is more than one suitable satellite image in this year. Please elaborate more clearly.

L136: 'versus image date': the image dates are not consistent across the study region, not least because of atmospheric disturbances. What do the authors mean here?

L137: delete 'on'? 'chooses the median'?

[Figure]

L140-141: Li et al. (2019) had reported similar variance in the total annual lake area in the Tian Shan, and I am still wondering whether these changes are real (which means that a substantial area of glacier lakes must have shrunk in some years) or whether this is the result of 'adverse weather conditions, varying lake characteristics and image quality'. Given that the authors in this study manually checked and mapped each lake, it is surprising that the current maps still demand smoothing. At least the authors should be able to explain how much of the annual variance can be accounted to any of these three effects, and other effects such as ice or shadow on lakes, which the authors have not mentioned yet.

L146: maybe 'elimination of the effect from differences', or rephrase appropriately. And what do the authors mean with 'differences in the sensor capabilities'?

L151-153: How many lakes (in total, or percent) were manually added to the automatically derived lake inventories? How many needed manual refinement?

L159: So n also contains m? Why is the perimeter not shown as an own variable?

L160: remove 'areal'.

L170: The authors could explain more clearly what they mean by 'precision' here. It seems that 'precision' is a conversion of uncertainty, so that the issue of high uncertainties for small lakes remains.

L171: 'Krumwiede et al. (Krumwiede et al., 2014)': remove double reference.

L173-174: Unclear what the authors wish to say here. What do they mean with 'predictably much better'? And how can they compare precision and accuracy, if these measures are not on the same scale?

L175: The authors could consider plotting the distributions of uncertainties (instead of a table) to highlight differences in the tails of uncertainties in each year? Do the differences in accuracy (e.g. lowest in 2010, highest in 2016) scale with the total area or number of glacier lakes?

L178: The information from this sentence should go into the abstract.

L179: 'A linear least-squares': Why did the authors refrain from using the Theil-Sen-regression here?

L179 & L181: '18 km2 a-1' & '380 lakes a-1': Please add confidence bands. In principle, all numbers and changes in Chapter 5.1 need error bounds because of the mapping uncertainty.

L182: 'Fig. A3': unclear why this figure does not show all lake sizes. This could support the argument of the authors that smaller lakes have increased more in number than large lakes. On the other hand, do these two plot suggest that most of the increase in lake area is tied to the growth of large lakes? Data points in A3 could also have uncertainty bars given the mapping errors?

L183-184: 'The increase of proglacial lakes was concentrated above 4900 m (Fig. 3c)': unclear whether this clearly follows from this plot, could be also at 4600 or 5600 m asl. Plotting the absolute differences of lake number for each elevation band could help.

L186: 'supraglacial lakes': why are these not shown here? Given the downwasting of debris-covered glacier in the Himalayas, for example, this type of lakes could account for a substantial amount of the increase in total lake number, no?

L190-Fig 3a: why have some 1,000 lakes vanished between 2008 and 2009?

L194-195: Which 'non-parametric trend analysis' did the authors use?

L200: 'large patches': unclear formulation, please elaborate.

L201-203: What is the reason for the two bubbles with the large decrease in lake area (Fig. 4b), while all surrounding bubbles show a positive trend? Is this because one (or more) large lake(s) shrunk or burst out?

L210-211: What is happening in regions with shrinking lakes? Do some of these fall dry or burst out?

L211-214: Very nice findings. These numbers could find their way into the abstract?

L215: 'the largest area growth of lakes occurred in areas with relatively large proportion of small glacial lakes': is there a way to measure this notion?

L216-217: 'many large-lake dominated areas exhibited decreased or nearly unchanged lake extent': not sure whether Fig. 4 fully supports this conclusion. In the Eastern Himalayas, for example, small glacier lakes account for less than 10 % for the entire lake area in that region (Table A4), so that most of the growth must be tied to the largest lakes, right?

L218-222: Unclear content in these sentences, please reformulate.

L230-Fig4b: not sure whether annual trends are useful here because, as far as I understood, the authors assume that lake area scales linearly with time in each $1° \times 1°$ bin. Yet the authors provide neither a measure of uncertainty for these annual changes, nor is it clear whether these changes occur linearly in time or that the rate of change could be affected by outliers. Total changes in glacier lake area between 2008 and 2017 could probably make more sense here .

L203: where does Fig. A4 show that the increase in lake area is 'due to retreat and thinning of debris-covered glaciers'?

L207-208: Fig. 4c does not show any trends.

L220-227: OK – and what about the Pamir Alay, Western & Eastern Kunlun, and Eastern Himalaya, where the contribution to the total growth in lake area is much smaller from small lakes compared to large lakes, according to Table A2? What I learned from this table is, that there is some variability in the lake-size distribution, which itself could be more emphasized. I have more the feeling that there are few (or no?) general rules that could help explaining the regional distribution of glacier lakes and associated growth patterns. If that feeling is wrong, I would appreciate simple correlation plots that check for such notions.

L230: Fig 4b: Colors around zero could be changed to white and negative value to a darker shade of blue for more contrast in the bubbles.

L242-244: Could it be that the number of lakes in the smallest size class ($< 0.01$ km$^2$) remains constant in most regions because these lakes climb into the next higher size class from year to year? If this holds true, could this mean that the lowest size class tells us more about the annual production rate of lakes in that region?

L247-248: Binning should be equal, so that size classes are comparable. Now, the smallest class spans 0,0919 km$^2$, the medium 0.01 km$^2$, and the largest 0.02 km$^2$. Linear regression should have confidence bounds, given that the positive trends in some panels (e.g. K, CT, WH, EHK) could be due to outliers. All panels could have the same Y-axes to show differences in lake abundance between regions.

L256: R$^2$=0.53: I couldn't find this number in Table A5. Also, which type of correlation coefficient is calculated here? The caption of table A5 also talks about R, not R$^2$.

L257: 'larger ice-contact proglacial lakes imply larger calving-front interactions': source?

L259: What and how long is 'a typical glacier response time'?

L260-261: Content of this sentence not fully clear, please reformulate.

L264: 'geomorphic, topographic and climate parameters': more specifically?

L266: what does this correlation tell us?

L267: Not the best statistical approach to exclude data points that we don't like.

L267-269: What has the correlation between debris cover and glacier length to do with glacier lakes? Where is a proof for the notion that 'low-gradient glaciers favor supraglacial and proglacial lake formation'?

L272: 'Some adjacent regions': which ones exactly? 'comparable', 'large', 'substantial':

please be more specific.

L273: 'longer-term climate trends': on which scales? Years, decades, centuries?

L274: 'rapid warming': warming seems to be present across the entire study region except the Karakoram. 'decreased precipitation': precipitiation is unchanged in the Eastern and Western Himalaya, where lakes have grown most in the study period. The line of argumentation is not clear here.

L275-277: Unclear, please be more specific ('further west and north', 'similar'?) and reformulate.

L279-282: Vague conclusion – these arguments could explain anything, in the light of the high regional variability in all the predictors that the authors have shown. There is a large body of literature that has worked on the size distribution and regional growth pattern of glacier lakes (e.g. Song et al., 2016, 2017). However, this literature is largely left untouched here. By the way, until here, no references have been given for the datasets used in the correlations.

L282: Dan and CLAGUE, 2011: wrong citation.

L291: 'datasets'.

L293: compared to which dataset? Reminds me also a bit of comparing apples with oranges, given these studies used different minimum mapping units, sizes of the study regions, buffers around lakes, types of lakes analysed and so on.

L296: 'a much larger lake area': how much larger?

L300: How did the authors convert the gridded data from GSW to polygons that match the Hi-MAG database?

L300-301: 'more glacial lakes in the Himalaya, Eastern Hindu Kush, and Tien Shan, and fewer in Eastern Pamir and Western Kunlun Shan': please be more specific.

L304: 'significant number': how do the authors measure significance here?

L305: 'only part of lakes are formed by glacier meltwater': how do the authors measure the contribution of glacier meltwater to the total volume of a glacier lake?

L304: the GSW does not 'define' glacier lakes per se, but maps surface water. Please clarify.

L310: 'errors could exist in either dataset': a surprising notion given that 10 independent experts have mapping glacier lakes for the Hi-MAG database. Also, what causes these 'similar reflectance from the adjacent land surfaces' to be so special in the Tien Shan compared to all other regions?

L311-312: 'Karakoram regions have fewer glacial lakes in our estimate': because? The GSW has overestimates surface water on debris covered glaciers, may this helps explaining the difference?

L343-346: 'relatively good performance for areas having simple lake characteristics and environmental backgrounds'; 'diverse climatic conditions, physical properties and surrounding environments': please be more specific.

L360: Please add the year when these images were shot. In each panel, the authors could also show the lake polygons mapped for this year for each lake type.

L366: Please avoid using rainbow color scales.

L406: Why does this need to be acknowledged given that these researchers are co-authors in this study?

References

Brun, F., Berthier, E., Wagnon, P., Kääb, A. and Treichler, D.: A spatially resolved estimate of High Mountain Asia glacier mass balances from 2000 to 2016, Nat. Geosci., 10(9), 668–673, doi:10.1038/ngeo2999, 2017.

Khadka, N., Zhang, G. and Thakuri, S.: Glacial Lakes in the Nepal Himalaya: Inventory and Decadal Dynamics (1977–2017), Remote Sens., 10(12), 1913, doi:10.3390/rs10121913, 2018. Li, J., Warner, T. A., Wang, Y., Bai, J. and Bao, A.: Mapping glacial lakes partially obscured by mountain shadows for time series and regional mapping applications, Int. J. Remote Sens., 40(2), 615–641, doi:10.1080/01431161.2018.1516314, 2019.

Maharjan, S. B., Mool, P., Lizong, W., Xiao, G., Shrestha, F., Shrestha, R., Khanal, N., Bajracharya, S., Joshi, S. and Shai, S.: The status of glacial lakes in the Hindu Kush Himalaya-ICIMOD Research Report 2018/1 (2018)., ICIMOD Res. Rep., (2018/1), 2018.

Miles, E. S., Steiner, J., Willis, I., Buri, P., Immerzeel, W. W., Chesnokova, A. and Pellicciotti, F.: Pond Dynamics and Supraglacial-Englacial Connectivity on Debris-Covered Lirung Glacier, Nepal, Front. Earth Sci., 5, 69, doi:10.3389/feart.2017.00069, 2017a.

Miles, E. S., Willis, I. C., Arnold, N. S., Steiner, J. and Pellicciotti, F.: Spatial, seasonal and interannual variability of supraglacial ponds in the Langtang Valley of Nepal, 1999–2013, J. Glaciol., 63(237), 88–105, doi:10.1017/jog.2016.120, 2017b.

Nagai, H., Ukita, J., Narama, C., Fujita, K., Sakai, A., Tadono, T., Yamanokuchi, T. and Tomiyama, N.: Evaluating the Scale and Potential of GLOF in the Bhutan Himalayas Using a Satellite-Based Integral Glacier–Glacial Lake Inventory, Geosciences, 7(3), 77, doi:10.3390/geosciences7030077, 2017.

Pekel, J.-F., Cottam, A., Gorelick, N. and Belward, A. S.: High-resolution mapping of global surface water and its long-term changes, Nature, 540(7633), 418–422, doi:10.1038/nature20584, 2016.

Round, V., Leinss, S., Huss, M., Haemmig, C. and Hajnsek, I.: Surge dynamics and lake outbursts of Kyagar Glacier, Karakoram, The Cryosphere, 11(2), 723–739, doi:10.5194/tc-11-723-2017, 2017.
Shukla, A., Garg, P. K. and Srivastava, S.: Evolution of Glacial and High-Altitude Lakes in the Sikkim, Eastern Himalaya Over the Past Four Decades (1975–2017), Front. Environ. Sci., 6, 81, doi:10.3389/fenvs.2018.00081, 2018.

Song, C., Sheng, Y., Ke, L., Nie, Y. and Wang, J.: Glacial lake evolution in the southeastern Tibetan Plateau and the cause of rapid expansion of proglacial lakes linked to glacial-hydrogeomorphic processes, J. Hydrol., 540, 504–514, doi:10.1016/j.jhydrol.2016.06.054, 2016.

Song, C., Sheng, Y., Wang, J., Ke, L., Madson, A. and Nie, Y.: Heterogeneous glacial lake changes and links of lake expansions to the rapid thinning of adjacent glacier termini in the Himalayas, Geomorphology, 280, 30–38, doi:10.1016/j.geomorph.2016.12.002, 2017.

Veh, G., Korup, O., Roessner, S. and Walz, A.: Detecting Himalayan glacial lake outburst floods from Landsat time series, Remote Sens. Environ., 207, 84–97, doi:10.1016/j.rse.2017.12.025, 2018.

Wang, X., Guo, X., Yang, C., Liu, Q., Wei, J., Zhang, Y., Liu, S., Zhang, Y., Jiang, Z. and Tang, Z.: Glacial lake inventory of High Mountain Asia (1990-2018) derived from Landsat images, Earth Syst. Sci. Data, 1–23, doi:https://doi.org/10.5194/essd-2019-212, 2020.

Zhang, G., Yao, T., Xie, H., Wang, W. and Yang, W.: An inventory of glacial lakes in the Third Pole region and their changes in response to global warming, Glob. Planet. Change, 131, 148–157, doi:10.1016/j.gloplacha.2015.05.013, 2015.

Zhu, Z. and Woodcock, C. E.: Object-based cloud and cloud shadow detection in Landsat imagery, Remote Sens. Environ., 118, 83–94, doi:10.1016/j.rse.2011.10.028, 2012.

Please also note the supplement to this comment:
https://essd.copernicus.org/preprints/essd-2020-57/essd-2020-57-RC1-supplement.pdf

[Figure]

**Supplement:**

---

## Referee Comment (RC2) · Anonymous Referee #2 · 19 Jun 2020

Chen et al. used Google Earth Engine to map glacial lakes in High Mountain Asia (HMA) from 2008 to 2017 with Landsat imagery. Their data is given in annual time steps, which so far is the highest temporal resolution of glacial lake inventory for the HMA. Thus, this kind of dataset if with fine quality could be particularly useful for scientific researches in changes of the cryosphere of the HMA as well as for related assessments and evaluations on the hydrological responds and glacial lake outburst flood risks under a changing climate in the HMA.

Noticed that recently there published a similar dataset produced by Wang et al. (2020), which includes two periods (1990 and 2018) of glacial inventory for the HMA and currently is also under review for the ESSD. The later one used a more traditional method and probably involved more extensive manually inspection during their investigations. When comparing these two datasets for the closest period (2017 of Cheng et al. and 2018 of Wang et al. 2020), I found there is a very large discrepancy (Table 1) in their results, although they have claimed that they used similar (not the same) area threshold (0.0081 $km^2$ of Chen et al. 2020 and 0.0054 $km^2$ of Wang et al. 2020) and distance threshold (within a 10 km from the nearest glacier terminus) for the lake mapping.

**Table 1: Differences between two datasets of HMA glacial lake inventory**

| Source | Year | Total Numbers | Area_Sum ($km^2$) | Area_Max ($km^2$) | Area_Mean ($km^2$) | Area_Min ($m^2$) | Notes |
|--------|------|---------------|---------------------|---------------------|----------------------|--------------------|-------|
| Wang et al. 2020 | 2018 | 28953 | 1955.939 | 6.465 | 0.067 | 5400 | Altai mountains included |
| Wang et al. 2020 | 2018 | 22219 | 1672.479 | 6.465 | 0.075 | 8100 | Altai mountains excluded and area larger than 8100 $m^2$ |
| Chen et al. 2020 | 2017 | 14477 | 1635.939 | 26.598 | 0.113 | 8100 | Altai mountains excluded |

In general, dataset of Chen et al. has missed a quite number of glacial lakes when comparing their results with the Wang et al. 2020's. In addition, they excluded the Altay and Sayan mountains, which actually should be included for an inventory study for the HMA. Even if excludes Altay form Wang et al.'s result, there still exist remarkable discrepancy in total numbers (nearly 7800 lakes) and total area (~37 $km^2$) between each other (Table 1). I agree with the Reviewer #1 that the missing inventory by Chen et al. is far from the range of uncertainty, but it was indeed errors due to the lack of systematic experts' check through the results, which were greatly depended on the methods they used when applying their procedures in Google Earth Engine.

The current attribute table of this inventory is too sample, that it even did not give an ID for each lake. Glacial lakes should be indexed with unique ID that could be used to connect with RGI or GLIMS glacier inventory dataset. In addition, the abbreviations used in the dataset (PGL, UCL and SGL) were totally not mentioned in the main text of the paper, we don't know what they meant.

For a dataset paper, it should avoid including any further analysis on the data (for example, the inter-annual variability of lake area presented in the section 5.2), especially when their results exist large uncertainty or errors. The authors should pay more attention to how to control the quality of their dataset. Although the Google Earth Engine offer the opportunity to calculate lake

inventory with a higher temporal resolution, I still strongly recommend they check through their result for each year, or maybe improve their methods to avoid errors as much as possible. Then a comparison between their results with existing datasets (such as datasets published by Zhang et al. 2015 or Wang et al. 2020) is necessary for audients judgement of the data quality. They did have do some comparison between their results with the Global Surface Water (GSW) dataset, but both these two were calculated by Google Earth Engine.

**References:**

Zhang, G., Yao, T., Xie, H., Wang, W. and Yang, W.: An inventory of glacial lakes in the Third Pole region and their changes in response to global warming, Glob. Planet. Change, 131, 148–157, doi:10.1016/j.gloplacha.2015.05.013, 2015.

Wang, X., Guo, X., Yang, C., Liu, Q., Wei, J., Zhang, Y., Liu, S., Zhang, Y., Jiang, Z. and Tang, Z.: Glacial lake inventory of High Mountain Asia (1990-2018) derived from Landsat images, Earth Syst. Sci. Data, 1–23, doi:https://doi.org/10.5194/essd-2019- 12, 2020.

---

## Author Response (AR1)

**Response to Reviewers' comments to manuscript essd-2020-57**

**"Annual 30-meter Dataset for Glacial Lakes in High Mountain Asia from 2008 to 2017"**

**Dear Editors and Reviewers:**

Thank you a lot for your kind and careful reviewing. Your suggestions give us important and constructive perspective on this manuscript, and help to improve the manuscript greatly. We have fully considered all the comments of you, and have substantially revised our manuscript according to your comments. A point-by-point response to the outstanding comments raised is attached to this manuscript. The major changes are summarized as follows:

1. We have put considerable effort to update data of glacial lakes for the ten year records, and manually append their attribute information. The related statistics, figures and analysis in the article have also been modified based on the new lake inventory.

2. We have comprehensively investigated the existing works about glacial lake inventory in the Section 1. Introduction, and quantitatively analysed and compared these inventories with ours in the Section 6. Discussions, to clearly show benefits and challenges remaining in this study.

3. Detailed explanations about the mapping of some problematic ice-covered lakes have been given in the Section 3.2. The number of missed or misclassified lakes that need to be manually corrected were also described clearly.

4. A thorough and quantitative uncertainty analysis of lake area was added in the Section 4 of revised manuscript, the error bars for the lake area, and confidence intervals for the estimated trends were also added throughout the paper.

5. We have carefully modified the language deficiencies, imprecise expressions, and provided more detailed interpretations and conclusions.

The changes have been highlighted in colored text in the revised manuscript. In the following, we provide point by point responses to the outstanding comments and suggestions provided by the Anonymous Reviewers. We are indebted to you for your outstanding and constructive comments, which greatly helped us to improve the technical quality and presentation of our manuscript. Once again, thank you very much for your comments and suggestions.

**Response to Comments by Reviewer#1:**

1. *Chen et al. mapped glacier lakes in High Mountain Asia in annual time steps from 2008 to 2017 from Landsat imagery. They used an automated image segmentation algorithm to pre-classify >40,000 Landsat images between late spring and early fall. Ten experts manually refined the annual lake inventories. The authors further calculated trends of lake abundance and area in the entire study region and several subregions thereof, and show that the growth in lake area and number could be largely due to many new lakes that occupy the lower size spectrum. Hotspots of total lake growth are the Central Himalaya, Eastern Himalaya, and Western Himalaya. The authors also correlate the size-distribution of glacier lakes with a set of environmental predictors such as temperature change, debris cover, or glacier size, to explain where glacier lakes have formed in the landscape. Debris cover and trends in warming and precipitation could help to explain the size-distribution and growth of glacier lakes, though regional variances make a more rigorous explanation elusive.*

Response:

We greatly appreciate the suggestions of the Reviewer for his/her accurate summary of the main contributions of our work and for the outstanding recommendations provided. Following the Reviewer's very pertinent recommendations, in the revised manuscript and the following text we have made modifications according to these questions. We thank again the Reviewer for his/her careful handling of our manuscript and for the constructive suggestions provided, which greatly helped us improve the technical quality and presentation of our manuscript.

2. *Chen et al. have compiled an important, and possibly the most extensive, data set of glacier lakes in the HMA to date. This inventory could be useful for scientists and practitioners in many disciplines interested in changes of the cryosphere of the HMA. Each year has >14,000 lake polygons, so that it is difficult for me to judge how accurately these lakes have been mapped.* **(1)****Yet, visually comparing the inventories of 2016 and 2017, I noticed that the 2017 data set misses >2 km² of lakes in a subset 0.3° x 0.2° large. The figure in the attachment shows 11 yellow polygons from 2016 that are not overlapped by blue polygons in 2017. Yet all these lakes are clearly visible in the Landsat scene from 2017-09-26, which is the basemap here. It is surprising that the 'ten trained experts' (L150-151) have not detected this error.** *Of course, my cross-check was not systematic, and it could be that I just accidentally came across one of very few undetected lakes in this study. Nevertheless,* **(2)****I strongly recommend that the authors again check the inventories for such issues, because a larger quantity of undetected lakes could in fact contribute to the calculated statistics, and the conclusions drawn in this study***.*

Response:

We gratefully thank the Reviewer for his/her comments and suggestions. In this study, we produced the glacier lake dataset in High Mountain Asia in annual time steps from 2008 to 2017. The number of glacial lakes in each year is more than 12,000, totally, the ten year records have nearly 140,000 lake polygons. To improve work efficiency, obtain the accurate boundary of the glacial lakes, and meanwhile, minimize the subjective judgment errors of the operatives, we applied a systematic glacial lake detection method that combined two steps from initial glacial lake extraction and subsequently manual refinement of lake mapping results.

The initial glacial lake extraction using GEE can make sure that approximately half of all the lakes are automatically extracted. To make annual data more complete and accurate, manual inspection and refinement of individual glacial lake is necessary to supplement some missing lakes and correct the mapping errors such as mountain shadows and river segments. The glacial lake vector over the entire HMA for the years from 2008 to 2017 has been rechecked and reedited individually through dynamic cross-validation by ten trained experts. Besides, the related attribution (i.e., lake type, elevation, distance to the nearest glacier terminus, area and perimeter) were manually added for each lake. It should be noted that the type of each individual glacial lake was carefully judged according to the different formation mechanisms or growth stages. Thus the whole processing is time consuming and require a considerable amount of work. We are terrible sorry for missing lakes that larger than 2 $km^2$ in the Landsat scene from 2017-09-26. Although the lakes >2 $km^2$ are not that much, they may still contribute to the calculated statistics, and the conclusions drawn in some sub-regions.

Therefore, following the Reviewer's very pertinent recommendations, we have carefully examined the lake data for the ten year by supplementing glacial lakes that have an area of greater than 0.04 $km^2$. The area of 0.04 $km^2$ is chosen as the threshold for re-revision of the glacial lake dataset for three reasons: **i)** glacial lakes with small sizes are more likely to be confused with surroundings due to its less effective spectral, textural and spatial information in comparison with those of relatively large glacial lakes, which results in the **large uncertainty of area** (Please see the Section 4, small glacial lakes with the area of less than 0.04$km^2$ have the mean area uncertainty of about 25.7 %). **ii)** small glacial lakes are highly variable in their locations, shapes and size, and also the optimal images with valid observations over the potential glacial lake area is very limited **in each year**. The image dates are not consistent across the whole HMA region because of atmospheric disturbances, also the influences from image quality, ice and shadow that obscured the lakes, this creates a great deal of uncertainty about the number and extent of small glacial lakes **for the such an annually resolved inventory**. **iii)** given the amount of work required to update the inventory and also revise our manuscript before the deadline.

Our team have been working hardly to update data of glacial lakes larger than 0.04 $km^2$ in the ten year, and add their attribute information. Correspondingly, the related statistics, figures and analysis in the article have also been modified based on the new lake inventory. We tried our best to improve the data and manuscript, and hope that the revised manuscript will meet with approval. As an annual time steps data over the HMA region for the ten-year time period, the produced data may not be complete and perfect enough, yet our team will continuously update and share more and better glacial lake data in the future.

3. *In any case, the tremendous effort that the authors have spent to map several tens of thousands of glacier lakes is, unfortunately, not reflected in the quality of the manuscript. I first wish to emphasize some major points in each section of this manuscript, and outline further details below.*

Response:

We gratefully thank the Reviewer for his/her constructive comments. This paper focus on obtaining the comprehensive knowledge of the distribution, area of glacial lakes, and also quantification of variability in their size and type at high resolution in HMA. We develop a HMA Glacial Lake Inventory (Hi-MAG) database to characterize the annual coverage of glacial lakes from 2008 to 2017 at 30 m resolution. This is the first glacial lake inventory across the HMA with annual temporal resolution, it can provide details for different types of glacial

lakes and evolution patterns. Although the method of lake mapping was automatic, for quality control every lake polygon was inspected and was manually edited where needed. The related attribution were also added for each lake. This is a huge amount of work for the mapping of nearly 140,000 glacial lakes over ten time periods from 2008 to 2017.

In the last version of manuscript, the mapping work was not fully reflected in the quality of the manuscript indeed. Therefore, in the revised manuscript, we have made considerable efforts to fill that defect, which is mainly manifested in the four aspects including **supplement of glacial lake data, quantitative uncertainty analysis of lake area and adding confidence intervals for trends, details about mapping procedure, comparison with other lake inventories**. We have made the following improvements in our study:

1) We have been working hardly to update data of glacial lakes larger than 0.04 $km^2$ in the ten year, and add their attribute information. Based on the new lake inventory, we have also carefully modified the tables, figures, the related analysis and made reliable conclusions.

2) A thorough and quantitative uncertainty analysis of lake area was added in the Section 4 of revised manuscript. Given the mapping uncertainty of glacial lakes and thus for the further improvement of reliability of calculated statistics, the error bars for the lake area, and confidence intervals for the estimated trends were also added throughout the paper.

3) Detailed descriptions about the key procedures for glacial lake mapping were provided, such as the amount of work required for the manual editing, and mapping of some problematic ice-covered lakes.

4) To clearly show benefits and challenges remaining in this study, and also for more rigorous cross-validation, we firstly comprehensively investigated the existing works about glacial lake inventory. Then in the **Section 6.1** of revised manuscript, we have conducted a deep analysis and comparison between our Hi-MAG dataset and glacial lake inventory from Wang et al., 2020, who mapped glacial lake larger than 0.0054 $km^2$ in a much larger High Mountain Asia region in two time periods (1990 and 2018), and made their data public recently. The comparative discussions with other studies (e.g. Nie et al., 2017;Zhang et al., 2015;Pekel et al., 2016a) were used as a general reference.

Furthermore, we have carefully considered the major points in each section, and more points of criticism line-by-line provided by the Reviewer, and have substantially revised our manuscript according to your comments in the following response and revised manuscript.

4. *The introduction is very brief and the many dozens of glacier lake inventories that have been compiled in this region before remain largely unmentioned. These inventories are also largely disregarded in the discussion though they could help to discuss the benefits and challenges remaining in this study.*

5. *L37-39: The authors could use more suitable references to support their arguments: Salerno et al. used 10 m satellite images and have mapped lakes <0.001 $km^2$; Brun et al. have studied glaciers, not lakes; for the inventories with 'narrow geographic scope', it would be good have a reference for some of the major region such as Karakoram, Sikkim, Bhutan, Nepal, Tien Shan, Nyainqentanglha, etc., to stress this 'patchwork' of glacier lake inventories more clearly. The ~2003 inventory of ICIMOD (Maharjan et al., 2018) and the multi-temporal inventories by Zhang et al. (2015) and by Wang et al. (2020) are not*

*mentioned at all here, though these studies addresses many of the issues raised here. What about the study from Pekel et al. (2016), who mapped surface water at high resolution globally? In summary, the authors could acknowledge previous work more than in the present form, given that many teams of researchers had aimed to map glacier lakes in the entire HMA (or parts thereof) before. . .*

Response:

We gratefully thank the Reviewer for his/her careful reading of the manuscript and for pointing out these issues. Many previously published researches have devoted to the glacial lakes mapping with remotely sensed data over the different regions of HMA, their inventories have proved to be important data resources for the spatiotemporal analysis of glacial lakes and better understanding the response of glacial lakes to climate change and glacier retreat. In the last version of manuscript, the introduction section is very brief indeed, which lacked the detailed descriptions and related references about many glacial lake inventories complied in the HMA region before. Following the Reviewer's very pertinent recommendations, in the revised manuscript, we have comprehensively investigated the existing works in the **Section 1. Introduction**. The detailed modifications are as follows:

Many previously published researches have devoted to the glacial lakes mapping with remotely sensed data over the different regions of HMA. Some works mainly focus on the investigation of the development of relatively large glacial lakes. Rounce et al. identified 131 glacial lakes in Nepal in 2015 that are greater than 0.1 $km^2$ (Rounce et al., 2017). Li et al. compiled an inventory of glacial lakes ($\geq$ 0.01 $km^2$) with a spatial resolution of 30 m in the Karakoram mountains (Li et al., 2020). Aggarwal et al. shared a new dataset of glacial and high-altitude lakes that have an area > 0.01 $km^2$ for Sikkim, Eastern Himalaya from 1972–2015 (Aggarwal et al., 2017). Ukita et al. constructed a glacial lake inventory of Bhutan in the Himalaya from the period 2006-2010 based on high-resolution PRISM and AVNIR-2 data from ALOS. Considering small lakes present less of a GLOF risk. They set 0.01 $km^2$ as the minimum lake size (Ukita et al., 2011). Ashraf et al. used Landsat-7 ETM+ images for the 2000-2001 period to delineate glacial lakes greater than 0.02 $km^2$ in the Hindukush-Karakoram-Himalaya (HKH) Region of Pakistan (Ashraf et al., 2012). Because small glacial lakes experience highly variable in their shape, location, and occurrence, and were clearly sensitive to the warming climate and glacier wastage, a growing number of scholars have paied attention to the abundance of small glacial lakes. Salerno et al. provided a complete mapping of glacial lakes (including lake size less than 0.001 $km^2$) and debris-covered glaciers with 10-m spatial resolution in the Mount Everest region in 2008 (Salerno et al., 2012). Wang et al. utilized Landsat TM/ETM+ images for the years 1990, 2000 and 2010 to map glacial lakes with area more than 0.002 $km^2$ in Tien Shan Mountains (Wang et al., 2013a). Luo et al. examined glacial lake changes (lake area >0.0036 $km^2$) for the entire western Nyainqentanglha range for the five periods between 1976 and 2018 using multi-temporal Landsat images (Luo et al., 2020). International Centre for Integrated Mountain Development (ICIMOD) provided comprehensive information about the glacial lakes (greater than or equal to 0.003 $km^2$) of five major river basins of the Hindu Kush Himalaya (HKH) using Landsat images for the year 2005 (Sudan et al., 2018). Nie et al.

mapped the distribution of glacial lakes across the entire Himalaya in the year of 2015 using a total of 348 Landsat images at 30 m resolution. They set the minimum mapping unit to 0.0081 $km^2$ (Nie et al., 2017). Zhang et al. presented a database of glacial lakes larger than 0.003 $km^2$ in the Third Pole for the years ~1990, 2000, and 2010 (Zhang et al., 2015). All these researches greatly help to fill the data gap of glacial lakes information in the HMA region. At the global scale, Pekel et al. used millions of Landsat satellite images to record global surface water over the past 32 years at 30-metre resolution (Pekel et al., 2016b), many large and visible glacial lakes were also included. More recently, Shugar et al. mapped glacial lakes with areas >0.05 $km^2$ around the world using 254,795 satellite images from 1990 to 2018 (Shugar et al., 2020). Wang et al. developed a glacial lake inventory (with size of larger than 0.0054 $km^2$) across the High Mountain Asia at two time periods (1990 and 2018) using manual mapping on 30 m Landsat images (Wang et al., 2020b). They firstly introduce glacial lake inventory at such a large-scale, the data shared will be served as a baseline for the further studies related to water resource assessment or glacier hazard risk.

Besides, in the **Section 6.1 Comparison with other lake dataset** of the revised manuscript, we have deeply analysed and discussed about the major differences between the existing inventories and ours, to clearly show benefits and challenges remaining in this study. The detailed modifications are as follows:

[revised manuscript text omitted]

6.  *The methods do not introduce the problem of ice-covered lakes that occurs with increasing elevation. Though the authors manually corrected for missing or misclassified lakes, it remains unclear, how many lakes needed to be corrected manually after automatic classification, and how many lakes are possibly missing in this data base.*

    *L151-153: How many lakes (in total, or percent) were manually added to the automatically derived lake inventories? How many needed manual refinement?*

Response:

We gratefully thank the Reviewer for his/her outstanding recommendations provided. In this study, to reduce the influence of seasonal lake fluctuations for the mapping, one effective solution is to map glacial lakes and measure their long-term changes during stable seasons when lake extents are minimally affected by meteorological conditions and glacier runoff. Here the selected time series of Landsat data were generally from July to November. During this period of each year, the Landsat imagery featured less perennial snow coverage. The lakes also reached their maximum extent, specifically around the end of the glacier ablation season (June to August). These criteria were used for the selection of imagery for the initial glacial lake extraction using GEE, manual inspection and refinement of individual glacial lake were then conducted to visually inspect and correct the mapping errors. Hi-MAG might have made better use of the optimum satellite imaging season to map glacial lakes, potentially resulting in more complete mapping by avoiding conditions— such as periods of lake ice— that may confound mapping. For the images without valid observations over the potential glacial lake area, then optimal mapping time needs to be broaden during the whole year.

However, some lakes may be permanently coved by ice with the increasing elevation. Besides, it is also difficult to map glacial lakes in environments with similar reflectance from the adjacent land surfaces, Tien Shan is such a typical area, where mapping errors could exist in the dataset. On these conditions, a detailed manual editing is essential to map these glacial lakes. So we were not relying exclusively on automatic classification. Here the ice-covered glacial lakes that were missed in the automated processing were manually edited using additional high-quality scenes downloaded from USGS Earth Explorer website during the whole year with assistance of images from adjacent years.

As for the automatic classification using GEE, there are four main procedures including (i) the clipped Landsat images by the extent of the glacier buffers and assembled into a time-series dataset; (ii) identified and masked some poor quality observations such as cloud, cloud shadow, topographic shadow; (iii) calculated MNDWI; and (iv) extracted the potential glacial lake areas by applying adaptive MNDWI threshold. Based on the automated processing, nearly 60% glacial lakes in each year can be correctly classified, of the other lakes that were not properly classified, 30% were missed and 10% were misclassified (mainly mountain shadows and river segments). For such a large-scale area that characterized by various and complex climatic, geological and terrain conditions, this classification method is simple but effective, the results are also reasonable since it provides very low commission errors. As shown in Figure I, glacial lake outlines extracted using the automated classification method in our study fit the real boundary of the glacial lake very well, while manually delineated glacial lake outlines are largely influenced by people's subjective experience and manual operation, resulting in overestimation for most part of a glacial lake region, and underestimation for a few areas. Moreover, results from manual digitization show poor performance for the delineation of the some glacial lakes with complex curved shapes.

[Figure]

Manually delineated lake outlines from Wang et al. 2020
Automatically extracted glacial lake outlines

Figure I. Examples of extraction of glacial lake outlines using the automated classification method in our study and manual digitization from Wang et al. 2020.

However, an estimated 30% errors in the initially classified map were from the missing small glacial lakes. Glacial lakes with small sizes are more likely to be confused with surroundings due to its less effective spectral, textural and spatial information in comparison with those of relatively large glacial lakes. Because the automated method is mainly based on the spectral features for the glacial lake mapping over the large mountainous area, the spectral information provided by the small glacial lakes will be incomprehensive and insufficient for the accurate detection of this kind of glacial lakes under various and complex environmental conditions.

Therefore, visual interpretation and manual editing is still an effective way to ensure the high accuracy of lake inventories.

Finally, we have added some text in the **Section 3.2 Adaptive glacial lake mapping method** to give a detailed explanations about the mapping of some problematic ice-covered lakes that occurs with increasing elevation. The number of missed or misclassified lakes that need to be manually corrected were also described clearly.

The detailed modifications are as follows:

Based on the automated processing, nearly 60% glacial lakes in each year can be correctly classified, of the other lakes that were not properly classified, 30% were missed and 10% were misclassified. For such a large-scale area that characterized by various and complex climatic, geological and terrain conditions, this classification method is simple but effective, the results are also reasonable since it provides very low commission errors. To ensure the quality of inventory, strict quality control was conducted to visually inspect and correct the mapping errors after the automated processing using GEE. False lake features, mainly identified as mountain shadows and river segments, were manually removed by overlapping mapped lake shorelines on the source Landsat imagery and higher-resolution imagery in Google Earth. For missing glacial lakes, such as some lakes may be covered by ice and clouds for years, grow at steep glacier tongues, and lakes show heterogeneous reflectance with the surrounding backgrounds, the lake boundaries were edited further using ArcGIS.

7. *The text and the figures in the results (Chapter 5.1 and 5.2) show dozens of numbers and trends, regarding lake areas, abundance, and changes thereof, but remain without error bars or confidence intervals throughout. Yet the authors acknowledge that mapping errors for small lakes in particular can be large, so that it must be cautioned against interpreting these trends. The panels in Fig. 5, for example, show considerable variance, and it remains unclear whether the regression models used to calculate the trends, account for this variance.*

Response:

We greatly appreciate the Reviewer's valuable comments very much. In the Section 5.1 and 5.2 of the last version of manuscript, lake area and calculated trends lacked the related error bar and confidence intervals, respectively, which are not rigorous indeed. In the revised manuscript, we have calculated and added the absolute area error and relative error for the lake area, and significance level for the estimated trends throughout the paper.

Besides, as for the estimation of area error, most of the large glacial lakes (area $\geqslant 0.04$km$^2$) have the relative error of about 7%. We also measured glacial lake down to 0.0081 km$^2$ (nine pixels in Landsat imagery), where relative errors calculated were ~50%. It can be found that this systematic error was more significant for the small-sized glacial lakes. We gratefully thank the Reviewer for pointing out important issue about interpreting the trends for small glacial lakes cautiously. We have carefully modified the whole manuscript to make sure the statistics and related analyses all focus on large-sized and all sizes of glacial lakes over the whole HMA region or different mountain ranges to discuss spatial and temporal variability for the 10-year record, and meantime, to avoid analysing and explaining the trends and number for small-sized

glacial lakes. Fig. 5 shows the inter-annual variations in the number of small glacial lakes for different HMA mountain regions. Given the large uncertainty of area and number of small glacial lakes, and in order to make the analysis and logic of the whole article more rigorous, we have removed the Section 5.2 in the revised manuscript. Please see the revised manuscript for more detailed information.

8. *In Chapter 5.3 and the discussion, the authors search for environmental predictors that could contribute to the (changing) size-distribution of glacier lakes. Yet many of these correlations, for example with glacier slope or elevation, are presented in tables, but remain untouched in the manuscript. Other correlations such as local glacier mass balances (e.g. Brun et al., 2017) are disregarded, though these could be useful in the light of shrinking glaciers and growing lakes. In general, the discussion does not go far beyond showing these correlations; discussion with previous studies that aimed to explain the size-distribution of glacier lakes remains very limited.*

Response:

To the extent possible we have tried to improve the explanation and context provided for our results comparing lake distribution with primary glacial, geomorphic and climatic factors. Our main objective in this section is not to provide fundamental new understanding of lake formation processes (which would be outside the scope of this journal), but rather, utilise our unprecedented large lake dataset to confirm relationships and processes that have previously been suggested based on regional observations and studies. We have improved the reference to these previous studies and reworded sentences that were vague or unclear to the reviewer.

9. *Throughout the manuscript the authors use subjective qualifiers such as 'small', 'large', or 'relatively' that could be systematically filled with the data that they have produced. The orthography, grammar, and style could still deserve much improvement, which is surprising to see given the number of co-authors involved in the writing part of this study.*

Response:

We gratefully thank the Reviewer for his/her careful reading of the manuscript and for pointing out this issue. In the revised manuscript, we have put considerable effort to revise the whole text, to avoid using these subjective qualifiers such as 'small', 'large', or 'relatively', and improve the presentations of the relevant data. Moreover, we have also carefully modified the language deficiencies, imprecise expressions, some simplistic interpretations and style, and make the manuscript in a better design.

10. *These major points should be fully addressed in a revised manuscript. Below I detail line-by-line some more points of criticism. Some technical corrections are also included.*

    *L17: 'Atmospheric warming' instead of 'climate change'?*

Response:

Thanks for this good suggestion. We have replaced the words "climate change" by "Atmospheric warming" in the revised manuscript.

11. *L19: 'incomplete': Can this study claim completeness?*

Response:

We apologize for using this inappropriate expression. Our research is also incomplete in some respects, for instance, the incomplete delineation of some quite small glacial lakes ($<0.04 km^2$) over ten year period; besides, although our Hi-MAG dataset was produced in annual time steps from 2008 to 2017 (which has high temporal resolution), ten years is still a short span, shorter than typical glacier response times to the atmospheric warming, lake expansion can decouple from short-term climate fluctuations. Therefore, inspired by this original report, our research group will continue to update and optimize the released glacial lake data in the near future.

To make this expression much more accurate and rigorous, we have rewritten the related sentences in the revised manuscript. The detailed modifications are as follows:

There is a pressing need for obtaining the comprehensive knowledge of the distribution, area of glacial lakes, and also quantification of variability in their size and type at high resolution in HMA.

*12. L21-22: 'rapid', 'moderate', 'large', 'faster': please be more specific. From annual lake inventories, it should be straightforward to calculate, for example, the total lake area, number, and absolute changes during the study period.*

Response:

We gratefully thank the Reviewer for his/her constructive comments. By calculating the corresponding statistics of the number and area of glacial lakes, we have carefully revised the contents about 'rapid', 'moderate', 'large' and 'faster' to make them more specific. The detailed modifications are as follows:

Annual increase in lake number and area are 306 glacial lakes and 12 $km^2$, respectively, and maximum increased lake number occurred in 5400 m elevation, which increased by 249.

*13. L23-24: 'Proglacial lake dominated areas'; 'unconnected lake dominated areas': unclear what and where these areas are.*

Response:

We are so sorry for these unclear expressions. Proglacial lake dominated areas mainly refer to the areas (over 1°×1° grid) that around more than half of the glacial lake area consisted of proglacial lakes. In this study, the Nyainqentanglha and Central Himalaya are the typical representatives of the proglacial lake dominated areas. Similarly, in the regions of Eastern Tibetan Mountains and Hengduan Shan, the unconnected glacial lakes occupied over half of the total lake area in 1°×1° grid, were dominantly occupied.

Besides, to make it much more clearly, in the revised manuscript, we have modified this sentence by providing more details about what and where the proglacial lake dominated and unconnected lake dominated areas are. The detailed modifications are as follows:

Proglacial lake dominated areas such as the Nyainqentanglha and Central Himalaya, where around more than half of the glacial lake area (summed over 1°×1° grid) consisted of proglacial lakes showed obvious lake area expansion, while in the regions of Eastern Tibetan Mountains and Hengduan Shan,

*14. L23: 'significant': how do the authors measure significance here?*

Response:

We greatly appreciate the Reviewer's valuable comments very much. For the glacial lake area time series in each $1°×1°$ grid, we applied the Theil-Sen estimator to derive the slope (annual change) of the trend. The change estimates satisfy the 90% confidence interval for the slope. While in the last version of manuscript, areas showed 'significant' lake area expansion originally meant that the areas have fast or obvious expansion. For example, in the Nyainqentanglha and Central Himalaya, where around more than half of the glacial lake area (summed over $1°×1°$ grid) consisted of proglacial lakes, showed obvious lake area expansion. From Fig. 5(b), the annual area change in these areas are larger than +0.30 $km^2 a^{-1}$, present deep red in color bar.

Moreover, we are sorry for the improper use of the word 'significant'. We have used "obvious" instead of "significant'" to make it easy to understand. The detailed modifications are as follows:

"where around more than half of the glacial lake area (summed over $1°×1°$ grid) consisted of proglacial lakes showed obvious lake area expansion,"

*15. L26: 'a main contributor': to? Increase in lake area?*

Response:

We gratefully thank the Reviewer for his/her careful reading of the manuscript and for pointing out this issue. Proglacial lakes contributed approximately 62.87% of total area increase (56.67 $km^2$) over HMA (Table A1 and A2), serving as a main contributor to the increase in lake area. We have supplemented this content in the revised manuscript.

*16. L27: 'an overlooked element'. Not sure whether this can be called a novel finding, given that the prevalence of small lakes in glacier lake inventories has been emphasized before, e.g. in (Khadka et al., 2018; Nagai et al., 2017; Shukla et al., 2018)*

Response:

We gratefully thank the Reviewer for providing us with important references from Khadka et al., 2018, Nagai et al., 2017, and Shukla et al., 2018, they definitely have a certain guiding significance for our study. In these studies, the delineated small glacial lakes were widespread in the different sub-regions of HMA. Results shown that small glacial lakes are highly variable both in the number and spatial extent, and are quite sensitive to the atmospheric warming. The importance of these small glacial lakes has become increasingly prominent, and people pay more and more attention to them. Therefore, in the last version of manuscript, the related statement about small glacial lakes are 'an overlooked element' to recent lake evolution in HMA is not a novel finding **indeed**. In our study, to further emphasize that small glacial lakes play an important role for the total number changes and area expansion of glacial lakes in **the**

**entire HMA**, we have modified the words 'an overlooked element' to 'an important element' in the revised manuscript.

*17. L29: 'is' instead of 'are'?*

Response:

Thanks for this suggestion. We have replaced 'are' by 'is' in the revised manuscript.

*18. L30: New sentence instead of comma?*

Response:

Thanks for this good suggestion. We have corrected the related sentences to make it clearly in the revised manuscript. The detailed modifications are as follows:

"it can be used for studies on the complex interactions between glacier, climate and glacial lake, glacial lake outburst floods, potential downstream risks and water resources."

*19. L34: 'significant': how do the authors measure significance here?*

Response:

We greatly appreciate the comment of the Reviewer, and we are terrible sorry again for the inappropriate use of 'significant', which may confuse the readers. Similar to your comment 14, the word 'significant' here is different from the concept of statistical significance, it means that obvious or widespread, namely, atmospheric warming has resulted in widespread glacier retreat and downwasting in many mountain ranges of the HMA. In the revised manuscript, we have completely modified this sentence to make it easy to understand. The detailed modifications are as follows:

"Atmospheric warming has resulted in widespread glacier retreat and downwasting in many mountain ranges of the HMA (Brun et al., 2017b;Bolch et al., 2012b),"

*20. L35: 'incompletely': same argument as in the abstract. Do the authors mean that glacier lakes had been mapped at large intervals?*

Response:

We gratefully thank the Reviewer for his/her valuable comments and suggestions. Yes, the word 'incompletely' does mean in the previously published researches, glacial lakes had been mapped at relatively large time intervals, not at annual time scale. That is, glacial lakes have been incompletely documented at small time intervals in the previous studies. To explain the 'incompletely' much more clearly, we have corrected this sentence in the revised manuscript. The detailed modifications are as follows:

"yet glacial lakes have been incompletely documented at small time intervals."

*21. L36-37: Please add a reference.*

Response:

Thanks for this suggestion. We have added a reference after this sentence. The detailed modifications are as follows:

"Glacial lake development varies according to climatic, cryospheric, and lake-specific conditions, such as basin geometry that is either connected to glaciers or unconnected, and the length of the lake/glacier contact (Zhao et al., 2018)."

*22. L52: 'atmospheric warming' instead of 'climate warming'?*

Response:

Thanks for this good suggestion. We have replaced the words "climate warming" by "atmospheric warming" in the revised manuscript.

*23. L45-56: What is the difference between i) and iv)? For example, given the situation of a lake that is dammed by a moraine, and its parent glacier calving into it: is this a lake 'usually connected to the glacial tongue and dammed by unconsolidated or ice-cemented moraines' (L46) or a lake 'bounded by a lateral moraine on one side and damming glacier ice on the other side'? Furthermore, I miss the category of purely glacier-dammed lakes, such as the one at Kyagar glacier (Round et al., 2017). Maharjan et al. (2018) list more than 20 ice-dammed lakes in the Himalayas.*

Response:

We gratefully thank the Reviewer for his/her careful reading of the manuscript and for pointing out this issue. In this study, all the glacial lakes were manually classified into four categories according to their position relative to the parent glacier or their formation mechanisms. Among these four types of glacial lakes, proglacial lakes are located next to the glacier terminus and receive melt water directly from their mother glaciers. These lakes usually connected to the glacier tongue and dammed by **glacier ice, unconsolidated or ice-cemented moraines** (mixture of ice, snow, rock, debris and clay, etc.), as shown in Figure I(a). Ice-marginal lakes are not very common in HMA, these lakes are generally distributed on one side of the glacier tongue, which means that the lake is dammed by the glacier ice on this side. While on the other side, it is bounded by a **lateral moraine** (Figure I(b)). With the increase of atmospheric warming and accelerated melting of glacier, some glacier tributary gradually detaches from a main trunk glacier. The detached location, where glacier melting has been particularly intense, in some case is also likely to form ice-marginal lakes. Based on the above analyses, if a lake is dammed by a moraine, and its parent glacier is calving into it, which means that the lake is contacted with its parent glacier terminus, this lake can be called a proglacial lake. Besides, in our study, purely glacier-dammed lakes are typically one kind of glacial lakes formed by the advance of glaciers and dammed by glacier ice, thus usually directly connected to the glacier tongue (as shown in Figure I(c), Kyagar glacial-dammed lake). Although the dam composition and structure is slightly different between the proglacial lakes and glacier-dammed lakes, dam of proglacial lakes is composed by the ice, rock, clay, etc., while the dam of glacier-dammed lakes is made up of almost pure glacier ice, because they are all located in the front of the glacier tongue and driven by the mother glacier, in the process of appending attributed information to each glacial lake, glacier-dammed lakes were merged into class of proglacial lakes.

Maharjan et al., 2018 adopted a two-level classification that is based on lake dam type and lake forms for the glacial lakes in the Hindu Kush Himalaya. The first level, based on dam type,

broadly classifies the lakes into four types: 1) moraine-dammed lake, 2) ice-dammed lake, 3) bedrock-dammed lake, and 4) other dammed lake. The second level, based on lake form, classifies the lakes into seven subtypes. **Ice-dammed lakes** are defined as lakes dammed by glacier ice, including lakes on the surface of a glacier or lake dammed by glaciers in the tributary/trunk valley, or between the glacier margin and valley wall, or at the junction of two glaciers. The criteria for classifying a glacial lake by Maharjan are completely different from our study. Given that our study area covers the entire HMA region, which includes ensemble of mountain ranges and favoured different properties of glacial lakes, all the lakes were classified mainly with respect to their position relative to the glacier and their formation mechanisms.

[Figure]

| (a) | (b) | (c) |

Figure I. Examples of (a) pro-glacial lakes, (b) ice-marginal lakes, and (c) glacier-dammed lakes (Kyagar glacial-dammed lake).

It should be noted that, in the last version of manuscript, the composition of dam also includes the glacier ice, which has been omitted in the definition of proglacial lakes. In addition, ice-marginal lakes and glacier-dammed lakes were also not introduced completely and clearly. Therefore, in the revised manuscript, we have thoroughly modified the concepts of different categories of glacial lakes in the Section I. Introduction. The detailed modifications are as follows:

Lakes were manually classified into four categories according to their position relative to the parent glacier or their formation mechanisms (Fig. A1): i) proglacial lakes, usually connected to the glacier tongue and dammed by glacier ice, unconsolidated or ice-cemented moraines (mixture of ice, snow, rock, debris and clay, etc.), proglacial lakes are located next to the glacier terminus and receive melt water directly from their mother glaciers; ii) supraglacial lakes - this is where ponds form in depressions on low-sloping parts of the surface of a melting glacier and are dammed by ice or the end-moraine or stagnating glacier snout; iii) unconnected glacial lakes, which are glacial lakes not directly connected to their parent glaciers at the present time but which, to some extent, may be fed by at least one of the glaciers located in the basin and may have been (but not necessarily are) recently detached from ice contact due to glacial recession. Although not directly connected with the parent glaciers, these glacial

lakes are also the outcome of glacier melting in response to atmospheric warming, they can supply fresh water to major river systems of the HMA region, and their changes have significant scientific and socio-economic implications (Nie et al., 2017;Song et al., 2016); and (iv) ice-marginal lakes, these lakes are generally distributed on one side of the glacier tongue, which means that the lake is dammed by the glacier ice on this side. While on the other side, it is bounded by a lateral moraine. With the increase of atmospheric warming and accelerated melting of glacier, some glacier tributary gradually detaches from a main trunk glacier. The detached location, where glacier melting has been particularly intense, in some case is also likely to form ice-marginal lakes. We note that such ice-marginal lakes are very common in some parts of the world (e.g., Alaska (Armstrong and Anderson, 2020;Capps et al., 2011)) but are not common in HMA. Besides, purely glacier-dammed lakes are typically one kind of glacial lakes formed by the advance of glaciers and dammed by almost pure glacier ice. Although the dam composition and structure is slightly different between the proglacial lakes and glacier-dammed lakes, because they are all located in the front of the glacier tongue and driven by the mother glacier, in the process of appending attributed information to each glacial lake, glacier-dammed lakes were merged into class of proglacial lakes.

*24. L55-56: 'Alaska': reference missing*

Response:

Thanks for this suggestion. We have added the corresponding references in the revised manuscript. The detailed modifications are as follows:

"We note that such ice-marginal lakes are very common in some parts of the world (e.g., Alaska (Armstrong and Anderson, 2020;Capps et al., 2011)) but are not common in HMA."

*25. L57: 'for potential automatic mapping errors': The authors did not mention before that they mapped lakes automatically.*

Response:

We gratefully thank the Reviewer for his/her comments and suggestion. As for this sentence, we did not really mention before that the lakes were mapped automatically. We have modified this sentence to make it much more accurate in the revised manuscript. The detailed modifications are as follows:

"Every lake was cross-checked manually for its boundary and attribution."

*26. L58: What are 'systematic errors'?*

Response:

Thanks for this good suggestion. In this study, systematic errors refer to the error estimate of glacial lake area. Assuming an uncertainty of 1 pixel for the detected glacial lake boundaries, we calculated the lake area error for the whole HMA region according to the equation (1). To

avoid conceptual confusion and for better understanding, we have corrected 'systematic errors' to 'lake area error' in the revised manuscript.

*27. L60: The chapter on the study area contains very limited information on the 'study area'. Could be expanded, including the climate and topography of the HMA. This could prepare readers for the observed variability of glacier lake abundance.*

Response:

We greatly appreciate the Reviewer's valuable comments very much. For preparing the readers for the observed variability of glacial lakes in the HMA region, following the Reviewer's very pertinent recommendation, we have rewritten and expanded the chapter on the study area by adding the map of the HMA and comprehensive descriptions about the climate and topography. The detailed modifications are as follows:

**2.1 Study area**

The term HMA refers to a broad high-altitude region in South and Central Asia that covers the whole Tibetan Plateau and adjacent mountain ranges, including the Eastern Hindu Kush, Western Himalaya, Eastern Himalaya, Central Himalaya, Karakoram, Western Pamir, Pamir Alay, Northern/Western Tien Shan, Dzhungarsky Alatau, Western Kunlun Shan, Nyainqentanglha, Gangdise Mountains, Hengduan Shan, Tibetan Interior Mountains, Tanggula Shan, Eastern Tibetan Mountains, Qilian Shan, Eastern Kunlun Shan, Altun Shan, Eastern Tien Shan, Central Tien Shan, Eastern Pamir (Fig. 1 and Fig. 4a). It extends from 26°N to 45°N and from 67°E to 105°E, the altitude of the plateau is about 4500 m on average (Baumann et al., 2009). It's made up of alternating mountains, valleys and rivers, the terrain is fragmented, showing a decreasing trend from Northwest to Southeast. HMA has a series of East-West mountains that occupy most of the area. Among these, Tanggula Shan lies in the central part of the HMA, with an altitude of over 6000 m. The height of fifteen highest mountains in the Himalayas are more than 8000 m, while the peaks of the mountains in the northern plateau are more than 6500 m. The North-South mountains are mainly distributed in the southeast of the plateau and near the Hengduan Mountain area. These two groups of mountains constitute the geomorphic framework and control the basic pattern of the plateau landform. Continuous and discontinuous permafrost have developed on the higher land and north facing slopes.

HMA is the source of several of Asia's major rivers, including the Yellow, Yangtze, Indus, Ganges, Brahmaputra, Irrawaddy, Salween, and Mekong. They play a crucial role in downstream hydrology and water availability in Asia (Immerzeel and Bierkens, 2010). Most glaciers in the Tibetan Plateau are retreating, except for the Western Kunlun (Neckel et al., 2014;Kääb et al., 2015) and the Karakoram, where a slight mass gain is occurring (Bolch et al., 2012a;Gardner et al., 2013). Moreover, glaciers in different mountain ranges show contrasting patterns. Local factors (e.g., exposure, topography, and debris coverage) may partly account for these differences but the spatial and temporal heterogeneity of both the climate and degree of climate change may be the main reason. Glacial lakes are formed and developed temporally with the retreat or thinning of glaciers and are directly or indirectly fed by glacier meltwater, they are located within 10 km from the nearest glacier terminus (Zhang et al., 2015;Wang et al., 2013b).

The HMA climate is under the combined and competitive influences of the East Asian and South Asian monsoons and of the westerlies (Schiemann et al., 2009). This unique geographical position produces an azonal plateau climate characterized by strong solar radiation, low air temperatures, large daily temperature variations and small differences between annual mean temperatures (Yao et al., 2012).

The annual mean temperature is 1.6 °C, with the lowest temperature of −1 − −7 °C occurring in January and the highest temperature of 7 − 15°C occurring in July. The cumulative precipitation is about 413.6 mm a year.

[Figure]

**Fig. 1. The location of the High Mountain region of Asia (HMA). Glacier outlines from the Randolph Glacier Inventory (RGI v5.0), the Second Chinese Glacier Inventory (CGI2) and the GAMDAM inventory are drawn in sky blue.**

*28. L62: 'etc': Please write out all regions; Fig 4a is mentioned before Fig 1.*

Response:

Thanks for your suggestion. In the revised manuscript, we have written out the names of all the mountain ranges in the HMA and adjusted the position of the Fig. 1 and Fig. 4a. The detailed modifications are as follows:

"The term HMA refers to a broad high-altitude region in South and Central Asia that covers the whole Tibetan Plateau and adjacent mountain ranges, including the Eastern Hindu Kush, Western Himalaya, Eastern Himalaya, Central Himalaya, Karakoram, Western Pamir, Pamir Alay, Northern/Western Tien Shan, Dzhungarsky Alatau, Western Kunlun Shan, Nyainqentanglha, Gangdise Mountains, Hengduan Shan, Tibetan Interior Mountains, Tanggula Shan, Eastern Tibetan Mountains, Qilian Shan, Eastern Kunlun Shan, Altun Shan, Eastern Tien Shan, Central Tien Shan, Eastern Pamir (Fig. 1 and Fig. 4a)."

*29. L62: 'For this study': are there other studies where glacier lakes formed differently?*

Response:

We are terrible sorry for making this mistake. In both our study and other studies, glacial lakes are all formed and developed temporally with the retreat or thinning of glaciers and are directly

or indirectly fed by glacier meltwater. We have deleted the words 'For this study' in the revised manuscript.

*30. L63: 'within 10 km' instead of 'within a 10 km'?*

Response:

Thanks for your suggestion. We have used 'within 10 km' instead of 'within a 10 km' in the revised manuscript.

*31. L64: 'Approximately 40,481 Landsat series satellites': did the authors mean 'Landsat images from the three Landsat missions'? The term 'approximately' seems a bit odd here, given this exact number.*

Response:

We greatly appreciate the suggestion of the Reviewer. Landsat series satellites scenes mean images from the three Landsat missions including Landsat 5 TM, Landsat 7 ETM+ and Landsat 8 OLI. In the revised manuscript, we have modified this sentence by explaining 'Landsat series satellites' clearly and giving the exact number of images used in this study. The detailed modifications are as follows:

==40,481 satellite images including Landsat 5 TM imagery during 2008 to 2011, Landsat 7 ETM+ imagery in 2012, and Landsat 8 OLI during 2013 to 2017,== were available in GEE and were used to produce the annual glacial lake maps over the entire HMA (Fig. 2).

*32. L69: What do the authors mean by 'year-round'? Why did they not fill the years 2004, 2005, 2006, and 2007 with images from ETM+, similar to approach for 2012?*

Response:

We gratefully thank the Reviewer for his/her constructive comments. Here the 'year-round' Landsat 5 TM data means all the available Landsat 5 TM images in each year. For the years before 2008, all the available Landsat 5 TM images in a single year do not fully cover the entire HMA region, please see Figure II for the examples of the annual Landsat 5 TM dataset available from 2004 to 2007 in GEE.

Similar for the images used for 2012 glacial lake mapping, Landsat 7 ETM+ images can be used as auxiliary data into the extraction of glacial lake outlines over the whole HMA region before 2008. However, the SLC-off condition of Landsat ETM+ introduces artefacts because the slatted appearance of the original images is occasionally carried into the glacial lake map in 2012. Techniques to fill the SLC-off gaps exist, but these create artificial values that will result in false detections of water. It is noted water mapping using multi-temporal time series images at large scales usually avoided the use of such techniques. Therefore, Landsat 7 ETM+ data with intensive slatted appearance is not really a good and suitable data for the classification of numerous of glacial lakes.

In this study, because the only useable data source for the year of 2012 is from Landsat 7 ETM+, to ensure continuity of annual data from 2008 to 2017, we have tried our best to extract the glacial lakes from the 2012 ETM+ images as accurately as possible. Errors caused by

striped gaps of Landsat ETM+ were manually corrected using additional high-quality scenes during the whole year with assistance of images from adjacent years.

[Figure]

Figure II. All the available Landsat 5 TM dataset covering the whole HMA region for the years of (a) 2004; (b) 2005; (c) 2006 and (d) 2007.

Finally, in the revised manuscript, we have carefully modified related content by giving more detailed explanations about 'year-round' and why did not fill the years 2004, 2005, 2006, and 2007 with images from ETM+. The detailed modifications are as follows:

For the years before 2008, all the available Landsat 5 TM data in each year (e.g., 2004, 2005, 2006, and 2007) do not fully cover the HMA region.

Therefore, Landsat 7 ETM+ data with intensive slatted appearance is not really a good and suitable data for the classification of numerous of glacial lakes. In this study, because the only useable data source for the year of 2012 is from Landsat 7 ETM+, to ensure continuity of annual data from 2008 to 2017, we have tried our best to manually extract the glacial lakes from the 2012 ETM+ images as accurately as possible.

Errors caused by striped gaps of Landsat ETM+ were manually corrected using additional high-quality scenes during the whole year with assistance of images from adjacent years.

*33. L73-74: 'in false detections of water' instead of 'in the false water detections'?*

Response:

Thanks for this good suggestion. We have replaced 'in the false water detections' by 'in false detections of water' in the revised manuscript.

*34. L72-73: 'were accurately classified but usually misclassified otherwise': not fully clear, please elaborate more clearly. Again, this sentence mentions classification, but any methodological background has not been given yet. The authors could consider to shift such sentences at appropriate locations down in the text.*

Response:

We gratefully thank the Reviewer for his/her outstanding recommendations provided. In this study, lakes out of the gaps 'were accurately classified but usually misclassified otherwise' can be interpreted as lakes out of the gaps were accurately classified, but if glacial lakes are covered by gaps, they will be misclassified. Besides, this sentence is related with the lake classification indeed, which should belong to the content of Section 3. Methods other than 2.2 Dataset. We have moved such sentences to the **3.2 Extraction of glacial lake outlines** in the Section 3 of the revised manuscript.

*35. L85-86: 'during stable seasons when lake extents are minimally affected by meteorological conditions and glacier runoff", which is 'from July to November': not sure whether this statement is true. Studies on supraglacial ponds (Miles et al., 2017b, 2017a) show that changes occur preferentially in that part of the year.*

Response:

We greatly appreciate the Reviewer's valuable comments very much. To reduce the influence of seasonal lake fluctuations for the mapping, one effective solution is to map glacial lakes and measure their long-term changes during stable seasons. Zhang et al., 2015 conducted an inventory of glacial lakes in **the Third Pole** for ~1990, 2000, and 2010 using Landsat TM/ETM+ data (Zhang et al., 2015). The Landsat data selected for lake delineation were generally from the months July through November of ~1990, 2000, and 2010. During this period in each year, Landsat images featured less perennial snow coverage, and lake has a larger area following glacier-induced runoff and monsoon-induced precipitation. Nie et al., 2017 generated glacial lake extents in **the Himalaya** for 1990, 2000, 2005 and 2010 using automatic object-oriented mapping and human inspection methods, and in 2015 from Landsat 8 images using an advanced automated adaptive lake mapping method (Nie et al., 2017). Similarly, to reduce the effect of seasonal variability, images in similar seasons were selected, mainly in autumn and early winters (September to December), when glacial lake change is minor after the ablation period of glaciers. As the most highly variable glacial lakes in the study area, supraglacial lakes change preferentially in the year. For example, as shown in Miles et al., 2017a and Miles et al., 2017b, supraglacial pond cover in the **Langtang valley** shows an increase during the pre-monsoon, **rises to a peak in the early monsoon (June to July)**, drops during the post-monsoon and decreases to negligible during the winter months (Miles et al., 2017a;Miles et al., 2017b). Based on the above analyses about the selected lake mapping time in the different regions, in this study, the selected time series of Landsat data were generally from July to November when lake extents are minimally affected by meteorological conditions and glacier runoff. Most of the lakes also reached their maximum extent, specifically around the end of the glacier ablation season (June to August). If valid observations can not be obtained during this time span, then optimal mapping time needs to be broaden.

Furthermore, we gratefully thank the Reviewer for providing us with important references from Miles et al., 2017a and Miles et al., 2017b, they definitely have a certain guiding significance for our study. We have added these articles as references and providing some

explanations about the selected optimal mapping time in the **Section 3.1 Satellite imagery selection strategy** of the revised manuscript. The detailed modifications are as follows:

Here the selected time series of Landsat data were generally from July to November based on the analyses of mapping time of glacial lakes in the different regions. During this period of each year, the Landsat imagery featured less perennial snow coverage. Change in area of glacial lake is minor and has a large area following the glacier runoff and precipitation (Zhang et al., 2015;Nie et al., 2017).

As the most highly variable glacial lakes in the study area, supraglacial lakes change preferentially in the year, showing an increase in area during the pre-monsoon, and rising to the peak area in the early monsoon (June to July) (Miles et al., 2017a;Miles et al., 2017b).

*36. L89: 'most-monsoon': did the authors mean 'post-monsoon'?*

Response:

We are terrible sorry for this inaccurate expression. 'most-monsoon' do indeed mean 'post-monsoon'. We have corrected 'most-monsoon' to 'post-monsoon' in the revised manuscript.

*37. L95: 'cloud score functions in GEE': did the authors make sure that this cloud score function works well in mountain headwaters with high snow and ice cover? This is a major challenge for cloud detection algorithms (Zhu and Woodcock, 2012).*

Response:

We gratefully thank the Reviewer for his/her outstanding recommendations provided. We realize that this is a very important point that we may not explain clearly in the last version of manuscript. In this study, the selected time series of Landsat data for the glacial lake mapping were generally from July to November. During this period, the Landsat imagery featured less perennial snow coverage. Change in area of glacial lake is minor and has a large area following the glacier runoff and precipitation. This is the initial condition for the selection of images for glacial lake mapping in the HMA region. Although the selected image seasons are slightly different due to the meteorological conditions in different regions, they all comply with the same criterion that lake area were in clear-sky images and has small snow coverage, which will ensure the initial reliability of the mapping glacial lakes through GEE cloud computing platform. For the annual delineation of glacial lakes, the number of selected images is quite limited based on this prerequisite. If no valid observations can be obtained, then optimal mapping time needs to be broaden in the whole year.

**To further increase data availability, and also as the basis for data selection in the periods beyond the optimum mapping time**, we set two criteria for the selection of imagery with valid observations over the potential glacial lake area by using the cloud score functions in GEE, including (i) cloud cover is less than 20% in the 10 km buffer around each glacier outlines of a Landsat scene, or (ii) less than 20% cloud cover for the entire scene. As the reviewer pointed out that cloud score functions in GEE may face a big challenge to detect clouds in mountain headwaters with high snow and ice cover, large amounts of snow and ice are also likely to be identified as clouds using these cloud score functions. However, in this study, it is a much stricter criteria to filter out more images with lots of cloud-lookalike (snow/ice) bad observations and finally choose images with valid observations over the

potential glacial lake area. In addition to the selection of images for the extraction of glacial lakes by setting optimal mapping time and criteria of cloud score mentioned above, manual inspection and refinement of individual glacial lake is still an essential way to ensure the quality of inventory using high-quality scenes that were downloaded manually from USGS Earth Explorer website.

Finally, to give more comprehensive analysis about how cloud score functions in GEE were used for the image selection, we have completely modified the related content in **Section 3.1 Satellite imagery selection strategy** of the revised manuscript. The detailed modifications are as follows:

As the most highly variable glacial lakes in the study area, supraglacial lakes change preferentially in the year, showing an increase in area during the pre-monsoon, and rising to the peak area in the early monsoon (June to July) (Miles et al., 2017a;Miles et al., 2017b). Although the selected image seasons are slightly different due to the meteorological conditions in different regions, they all comply with the same criterion that lake area were in clear-sky images and has small snow coverage. This will ensure the initial reliability of the mapping glacial lakes through GEE cloud computing platform. If no valid observations can be obtained, then optimal mapping time needs to be broaden during the whole year.

To further increase data availability, and also as the basis for data selection in the periods beyond the optimum mapping time, we set two criteria for the selection of imagery with valid observations over the potential glacial lake area by using the cloud score functions in GEE, including (i) cloud cover is less than 20% in the 10 km buffer around each glacier outlines of a Landsat scene, or (ii) less than 20% cloud cover for the entire scene. The cloud score functions in GEE may face a big challenge to detect clouds in mountain headwaters with high snow and ice cover, large amounts of snow and ice are likely to be identified as clouds. However, in this study, it is a much stricter criteria to filter out more images with lots of cloud-lookalike (snow/ice) and finally choose images with good observations.

*38. L95: 'a partial Landsat scene': unclear – please reformulate.*

Response:

Thanks for this suggestion. Here 'a partial Landsat scene' refers to a part of a Landsat image that located in the 10 km buffer around each glacier outlines. We have modified this related sentence to make it clearly in the revised manuscript. The detailed modifications are as follows:

(i) cloud cover is less than 20% in the 10 km buffer around each glacier outlines of a Landsat scene.

*39. L97: 'criteria' instead of 'criterions'?*

Response:

Thanks for this suggestion. We have replaced 'criterions' by 'criteria' in the revised manuscript.

*40. L97: 'needs to be broaden': by how many months? Possibly into the next year? Previous studies (Maharjan et al., 2018; Veh et al., 2018) have shown that some lakes could be veiled by clouds and ice for years.*

Response:

We greatly appreciate the Reviewer's valuable comments very much. During the image processing using GEE, the automated selection of images for the extraction of glacial lakes is based on the optimal mapping time and criteria of cloud score. For the images without valid observations over the potential glacial lake area, then optimal mapping time needs to be broaden **during the whole year**.

As for the lakes that could be covered by clouds and ice for years, which will results in mapping errors in the dataset. On these conditions, a detailed manual editing is essential to map these glacial lakes. So we were not relying exclusively on automatic classification. Here the cloud or ice-covered glacial lakes that were missed in the automated processing were manually edited using additional high-quality scenes downloaded from USGS Earth Explorer website during the whole year with assistance of images from adjacent years.

Besides, in the section 3.1 and 3.2 of the revised manuscript, we have modified related content to clearly show the broaden time and how to map lakes that coved by ice and cloud for years. The detailed modifications are as follows:

==If no valid observations can be obtained, then optimal mapping time needs to be broaden during the whole year.==

==For missing glacial lakes, such as some lakes may be covered by ice and clouds for years, grow at steep glacier tongues, and lakes show heterogeneous reflectance with the surrounding backgrounds, the lake boundaries were edited further using ArcGIS.==

*41. L98-100: Please reformulate this sentence, possibly splitting into two.*

Response:

Thanks for this suggestion. We have reformulated this sentence by splitting it into two sentences. The detailed modifications are as follows:

==Although the selected image seasons are slightly different due to the meteorological conditions in different regions, they all comply with the same criterion that lake area were in clear-sky images and has small snow coverage. This will ensure the initial reliability of the mapping glacial lakes through GEE cloud computing platform.==

*42. L104-106: Repetition, consider deleting.*

Response:

Thanks for this suggestion. We have deleted this sentence in the revised manuscript.

*43. L109: 'and SLC-off gaps'*

Response:

Thanks for this suggestion. We have added the word 'and' before 'SLC-off gaps' in the revised manuscript.

*44. L110: 'a more computationally efficient way': than?*

Response:

We are so sorry for this inappropriate expression. We have modified this sentence to make it much rigorous. The detailed modifications are as follows:

 Fmask has the advantage of being able to process a large number of images ==in a computationally efficient way==.

*45. L109-L111: I suggest to switch the order of these two sentences.*

Response:

Thanks for this good suggestion. We have switched the order of these two sentences in the revised manuscript.

*46. L112: 'slopes (larger than 10°)': How did the authors make sure that this threshold does not mask water bodies that have grown at steep glacier tongues since the SRTM DEM was shot in 2000?*

Response:

We greatly appreciate the Reviewer's valuable comments very much. Topographic shadows were masked using the slopes (larger than 10°) and shaded relief maps (value less than 0.25) calculated from SRTM data (Li and Sheng, 2012). This will remove considerable mountain shadows that have the similar spectral reflectance with water bodies. However, SRTM DEM was generated in 2000, which is different from the acquisition time of Landsat images used for the glacial lake mapping in this study, the derived slopes and shaded relief cannot fully represent the conditions on the date a given Landsat scene is acquired. As a consequence, some lakes that have grown at steep glacier tongue may be masked, and some mountain shadows that interfere with the mapping results of glacial lakes from GEE still remain, leading to the fact that glacial lakes in some steep areas are omitted, and some residual shadows are misclassified as glacial lakes. On these conditions, in order to further improve the accuracy of delineated glacial lake outlines, manual refinement of initial lake mapping results is necessary to supplement some missing lakes and correct the mapping errors such as mountain shadows using time series of high-quality Landsat images downloaded manually from USGS Earth Explorer website. For missing glacial lakes that have grown at steep glacier tongues, the lake boundaries were edited further using ArcGIS. False lake features, mainly identified as mountain shadows, were manually removed by overlapping mapped lake shorelines on the source Landsat imagery and higher-resolution imagery in Google Earth.

 Besides, in the revised manuscript, we have added some content in the **Section 3.2 Extraction of glacial lake outlines** to clearly illustrate mapping errors induced by slope threshold of SRTM DEM and how to correct the mapping errors. The detailed modifications are as follows:

 ==However, SRTM DEM was generated in 2000, which is different from the acquisition time of Landsat images used for the glacial lake mapping in this study, the derived slopes and shaded relief cannot fully represent the conditions on the date a given Landsat scene is acquired. As a consequence, some lakes that have grown at steep glacier tongue may be masked, and some mountain shadows that interfere with==

the mapping results of glacial lakes from GEE still remain, leading to the fact that glacial lakes in the steep areas are omitted, and residual shadows are misclassified as glacial lakes.

False lake features, mainly identified as mountain shadows and river segments, were manually removed by overlapping mapped lake shorelines on the source Landsat imagery and higher-resolution imagery in Google Earth. For missing glacial lakes, such as some lakes may be covered by ice and clouds for years, grow at steep glacier tongues, and lakes show heterogeneous reflectance with the surrounding backgrounds, the lake boundaries were edited further using ArcGIS.

*47. L112: 'value less than 0.25': Unclear what this value means in physical quantities.*

Response:

We gratefully thank the Reviewer for his/her careful reading of the manuscript and for pointing out these issues. According to the results from Li and Sheng, 2012, DEM-derived surface gradients of most glacial lakes are less than 10°, and the shaded reliefs are larger than 0.25 (Li and Sheng, 2012;Quincey et al., 2007). Generally, a lake is a flat surface at a constant water level, and the surface gradients across the lake are very slight. In contrast, shadow features are usually located on the dark side of the high mountains, where the sunlight is blocked, so their surface gradients are great and the terrain reliefs are small. Therefore, slope and shaded relief maps derived from DEMs can be used to mask the areas of topographic shadows. With these topographic features, the mountain shadow areas are defined as the places whose surface slopes are larger than 10° and shaded relief values are less than 0.25. Furthermore, we have added some text in the revised manuscript to give a clear explanation about meaning of 0.25 in physical quantities. The detailed modifications are as follows:

Topographic shadows are located in the areas where the sunlight is blocked, generally on the dark side of the high mountains, their surface gradients are great and the terrain reliefs are small. Therefore, topographic shadows were masked using the slopes (larger than 10°) and shaded relief values (less than 0.25) calculated from SRTM data (Li and Sheng, 2012;Quincey et al., 2007).

*48. L113: The paper from Quincey is from 2007.*

Response:

We are terrible sorry for this mistake, we have corrected the year of this paper in the revised manuscript.

*49. L115: Maybe 'acquired. Some'?*

Response:

Thanks for this suggestion. We have modified it as 'acquired. Some' in the revised manuscript.

*50. L116: 'remain' instead of 'remains', and 'was' instead of 'were'?*

Response:

Thanks for this suggestion. We have replaced 'remains' by 'remain' and 'were' by 'was' in the revised manuscript.

*51. L118: 'the relative stability': unclear, please reformulate.*

*L118: '0.0081 km$^2$': the reason for choosing this minimum mapping unit is not fully clear.*

Response:

We gratefully thank the Reviewer for his/her constructive comments. The minimum number of pixels that used to define a glacial lake in the image was inconsistent in different studies. For example, Zhang et al., 2015 set the smallest detectable glacial lakes in the Third Pole of larger than 0.0027 km$^2$ (three connected pixels) using the Landsat TM/ETM+ data. Nie et al., 2017 selected 0.0081 km$^2$ (nine connected pixels) as the minimum mapping unit to map glacial lakes in the Himalaya. Other studies set the minimum threshold areas as 0.001km$^2$ (Salerno et al., 2012), 0.002km$^2$ (Wang et al., 2013a), 0.0036km$^2$ (Luo et al., 2020), 0.0054km$^2$ (Wang et al., 2020b), 0.01km$^2$ (Li et al., 2020) for the identification of glacial lakes and analysis of their spatial and temporal variations.

A smaller minimum mapping unit will detect a greater number of glacial lakes, however, it will also bring proportionally larger uncertainty than large lakes at the same resolution. This is because the uncertainty of measurement depends on the linear error and the perimeter of glacial lakes (Salerno et al., 2012). Our results demonstrate that a lake area covering fewer than nine lake water pixels have an area error of larger than 50% (Please see the **Section 4 Cross-validation and uncertainty estimate** in the revised manuscript). Given the area uncertainty of glacial lakes and the spatial resolution of Landsat data, in this study, glacial lakes larger than nine pixels ($\geqslant$ 0.0081 km$^2$) were considered as the minimum mapping unit.

Finally, we have rewritten the related content to provide a fully clear reasons about the selection of 0.0081km$^2$ as the minimum mapping unit. The detailed modifications are as follows:

==To define a glacial lake in the image, the minimum number of water pixels was inconsistent in different studies. For example, Zhang et al., 2015 set the smallest detectable glacial lakes in the Third Pole of larger than 0.0027 km$^2$ (three connected pixels) using the Landsat TM/ETM+ data. Nie et al., 2017 selected 0.0081 km$^2$ (nine connected pixels) as the minimum mapping unit to map glacial lakes in the Himalaya. Other studies set the minimum threshold areas as 0.001km$^2$ (Salerno et al., 2012), 0.002km$^2$ (Wang et al., 2013a), 0.0036km$^2$ (Luo et al., 2020), 0.0054km$^2$ (Wang et al., 2020b), 0.01km$^2$ (Li et al., 2020) for the identification of glacial lakes and analysis of their spatial and temporal variations. A smaller minimum mapping unit will detect more glacial lakes, however, the uncertainty it brings is also larger than large lakes at the same resolution (Salerno et al., 2012). Our results demonstrate that a lake area covering fewer than nine water pixels have an area error of larger than 50% (Please see the Section 4). Given the area uncertainty of glacial lakes and the spatial resolution of Landsat data, in this study, glacial lakes larger than nine pixels ($\geqslant$ 0.0081 km$^2$) were considered as the minimum mapping unit;==

*52. L127: 'polygon include' instead of 'vector includes'?*

Response:

Thanks for this suggestion. We have replaced 'vector includes' by 'polygon include' in the revised manuscript.

*53. L128: 'distance to the nearest glacier terminus': how is this distance measured? Along the flow path or the Euclidian distance? What happens if there is no glacier upstream, given that the buffer around the glacier can extend into adjacent, yet glacier-free catchments?*

Response:

We greatly appreciate the Reviewer's valuable comments very much. The distance from a glacial lake to the nearest glacier is calculated using the 'near' tool embedded in ArcGIS 10.6, which specifies a radius to search for candidate near glacier polygons, and then estimates the nearest Euclidian distance among them. For the glacial lakes that have close relationship with their parent glaciers, such as proglacial lakes, supraglacial lakes and ice-marginal lakes, the calculated distance to the nearest glacier using this analysis tool is reasonable since there is glacier in the upstream and the parent glacier is the nearest glacier to the glacial lake in the current catchment. For the unconnected glacial lakes that are not directly contacted with glacier, their interaction with glacier is also not that close compared with other types of glacial lakes, the lake water source are partly provided by glacier and partly by rainfall. Sometimes, maybe there is no glacier in the upstream, the distance calculated by 'near' tool is likely to be the distance to the nearest glacier in the adjacent catchments.

Besides, in the revised manuscript, we have modified related text to clearly show how to measure the distance between glacial lake and the nearest glacier. The detailed modifications are as follows:

(v) manual inspection and refinement of individual glacial lake were conducted and the related attribution (i.e., lake type, elevation, Euclidian distance to the nearest glacier terminus, area and perimeter) were added for each lake.

*54. L135-146: It remains unclear, how the authors calculated the perimeter of a given lake in a given year, if there is more than one suitable satellite image in this year. Please elaborate more clearly.*

Response:

We gratefully thank the Reviewer for his/her outstanding recommendations provided. We realize that this is a very important point that we may not explain clearly. If there are more than one suitable satellite images in a year, image with the least cloud cover will be selected for the calculation of the perimeter of a given lake. We have added some content in the revised manuscript to make it clearly. The detailed modifications are as follows:

Outputs per lake polygon include the information about lake type, elevation, Euclidian distance to the nearest glacier terminus, area and perimeter. Noted that if there are more than one suitable satellite images in a year, image with the least cloud cover will be selected for the calculation of the area and perimeter of a given lake.

*55. L136: 'versus image date': the image dates are not consistent across the study region, not least because of atmospheric disturbances. What do the authors mean here?*

Response:

We are terrible sorry for this improper expression, which easily make the readers to be misunderstood. Based on the final generated lake inventory data, in this study, the slope of

linear regression of lake area (over the grid cell of 1°×1°) **versus mapping year** was used to qualify the **yearly lake area changes**. Here the 'image date' does not represent the specific acquisition time of the image chosen for the mapping of glacial lakes, but the mapping year from 2008 to 2017. We have modified the related text in the revised manuscript to make it clearly. The detailed modifications are as follows:

Based on the final generated lake inventory data, we used the slope of linear regression of lake area (over the grid cell of 1°×1°) versus mapping year to qualify the yearly lake area changes during the study period.

*56. L137: delete 'on'? 'chooses the median'?*

Response:

Thanks for this suggestion. We have deleted 'on' and replaced 'chooses median' by 'chooses the median' in the revised manuscript.

*57. L140-141: Li et al. (2019) had reported similar variance in the total annual lake area in the Tian Shan, and I am still wondering whether these changes are real (which means that a substantial area of glacier lakes must have shrunk in some years) or whether this is the result of 'adverse weather conditions, varying lake characteristics and image quality'. Given that the authors in this study manually checked and mapped each lake, it is surprising that the current maps still demand smoothing. At least the authors should be able to explain how much of the annual variance can be accounted to any of these three effects, and other effects such as ice or shadow on lakes, which the authors have not mentioned yet.*

Response:

We greatly appreciate the suggestion of the Reviewer. In this study, we developed an annual 30-m dataset for glacial lakes in HMA region during 2008-2017. The slope of linear regression of lake area (over the grid cell of 1°×1°) versus mapping year was used to qualify the yearly lake area changes during the study period. Different linear trends estimation methods have been applied for the use in linear fitting of time series of data. Among these, Theil-Sen estimator chooses the median slope among all the derived fitted lines, can effectively represent long-term area changes due to its robustness for the trend detection and insensitivity to outliers. Although all the lake were manually checked and edited, due to the limitation of available images and other factors, the conditions of glacial lake mapping are not perfectly consistent for each year. For example, the image dates are not consistent across the whole HMA region because of atmospheric disturbances, also the influences from varying lake characteristics, image quality, ice and shadow that obscured the lakes, which all contributed to the detection error in the lake extent and their annual variation. In order to make the glacier lake area consistent year by year to the degree possible and estimate reliable annual changes in lake area, a high-performance Theil-Sen estimator is necessary for fitting and smoothing of the annual time series of data (Kumar, 1968;Song et al., 2018).

As for the **quantitative evaluation** of the influences from these three effects, and other effects such as ice or shadow on the annual variance of glacial lakes, generally, these factors do cause the errors in the detected glacial lake outlines and calculated area, which is objective and acceptable due to the nature of limited remote sensing data. While for this study, because we used long time series (10 years) data for the estimation of annual lake area change, and also

the errors only accounts for a small proportion of the total glacial lake area for each year, errors in the observed lake area due to the different effects do not apparently affect the trend statistical results. Quantitative analysis of the area error for every glacial lake caused by any of the above factors requires the examination and verification of all the dataset over a ten-year period, which is a large amount of work. Given the workload and the limitation of revision time, quantitative statistics can not be obtained at present, but a detailed analysis for measuring the area error and annual variance of glacial lakes resulted from the above factors is needed urgently in our future work.

Finally, in the revised manuscript, we have completely modified the related content to give detailed explanations about why the current maps demand smoothing and the influences from three effects, and other effects such as ice or shadow on the annual variance of glacial lakes. The detailed modifications are as follows:

Although all the lake were manually checked and edited, due to the limitation of available images and other factors, the conditions of glacial lake mapping are not perfectly consistent for each year. For example, the image dates are not consistent across the whole HMA region because of atmospheric disturbances, also the influences from varying lake characteristics, image quality (Bhardwaj et al., 2015;Thompson et al., 2012), ice and shadow that obscured the lakes, which all contributed to the detection errors in the lake extent and their annual variation. Generally, these errors are objective and acceptable as a result of the nature of the limited remote sensing data. For this study, because we used long time series (10 years) data for the estimation of annual lake area change, and also the errors only accounts for a small proportion of the total glacial lake area for each year, errors in the observed lake area caused by the different effects do not apparently affect the trend statistical results.

*58. L146: maybe 'elimination of the effect from differences', or rephrase appropriately. And what do the authors mean with 'differences in the sensor capabilities'?*

Response:

Thanks for this good suggestion. We have modified the 'elimination the effect of differences' as 'elimination of the effect from differences' in the revised manuscript. Here 'differences in the sensor capabilities' refer to the differences in the performance of different Landsat sensors for the mapping of glacial lakes, which can be manifested in many aspects such as the radiometric resolution, spectral resolution, spatial resolution and revisit time. For example, Landsat 8 has advantages in lake mapping over its ancestors. First of all, the 12-bit radiometric resolution and the improved sensitivity of Landsat 8's OLI sensor result in higher SNRs, essential to water body identification. Water bodies are dark objects with a reflectance usually lower than 10% in optical bands, and their mapping benefits greatly from these improvements. NDWI image segmentation is an effective water body mapping method, and histogram segmentation algorithms are widely used. Although NDWI enhances the difference between water bodies and land background, it is more sensitive to noise than the original green and NIR bands. Compared to Landsat 7 ETM+ (an 8-bit instrument), Landsat 8 OLI greatly improves its SNRs in the green and the NIR bands, from 37 to 304 and 34 to 201, respectively (Morfitt et al., 2015).

In the revised manuscript, we have rewritten the related sentence to make it clearly. The detailed modifications are as follows:

59. *L159: So n also contains m? Why is the perimeter not shown as an own variable?*

   *L160: remove 'areal'.*

Response:

Thanks for this good suggestion. Here *n* represents the number of pixels on the boundary of a glacial lake, which approximately equals to the ratio of the perimeter of a lake to the spatial resolution. *m* is the area of a pixel in the Landsat image (m²). Therefore, *n* does not contains *m*. For better understanding, the perimeter can be expressed as its own variable, then the equation (1) is changed into:

$$A_{er} = 100 \cdot ((p/a)^{1/2} \cdot a^2) / A_{gl}$$

Where *p* refers to the perimeter of a glacial lake, *a* is the spatial resolution of the Landsat sensor (m), $A_{gl}$ is the lake area (m²) and the factor 100 is there to convert to percentage.

However, this expression looks much more complicated in form than before and may be not ease of use. Therefore, we used the number of pixels to define the perimeter of a glacial lake, rather than the perimeter itself.

Finally, in the revised manuscript, we have modified the description and explanation of each variable in the formula to make them much clearly and easier to understand. The detailed modifications are as follows:

Where *n* refers to the number of pixels on the boundary of a glacial lake, approximated by the ratio of the perimeter length and spatial resolution, *m* is the area of a pixel in the Landsat image (m²), $A_{gl}$ is the lake area (m²) and the factor 100 is there to convert to percentage.

60. *L170: The authors could explain more clearly what they mean by 'precision' here. It seems that 'precision' is a conversion of uncertainty, so that the issue of high uncertainties for small lakes remains.*

   *L171: 'Krumwiede et al. (Krumwiede et al., 2014)': remove double reference.*

   *L173-174: Unclear what the authors wish to say here. What do they mean with 'predictably much better'? And how can they compare precision and accuracy, if these measures are not on the same scale?*

Response:

We greatly appreciate the Reviewer's valuable comments very much and we are terrible sorry for these inappropriate expressions about 'precision', 'accuracy' and 'uncertainty'. As for the first point about what is the meaning of 'precision' here, it is a conversion of area uncertainty indeed, rather than the accuracy of glacial lake mapping results. Secondly, the reference regarding Krumwiede et al., 2014 appears repeatedly in the text and citation, and should be kept one. Finally, as for this sentence 'the precision of measurement (such as ability to detect area changes) is predictably much better than the accuracy', here the 'precision' and 'accuracy' are not on the same scale. What we originally meant was that because the type of data used in

this study are all Landsat images, the accuracy of detecting annual area change of glacial lake is better than uncertainty of lake area in a single year, this is just an inference, has not been confirmed by a scientifically systematic argument yet. The related words about '**predictably** much better' are really not rigorous, also the use of 'precision' and 'accuracy' are quite confusing, which easily make the readers to be misunderstood. Based on the above analysis, we have decided to delete these speculative sentences and inaccurate expressions in the revised manuscript.

*61. L175: The authors could consider plotting the distributions of uncertainties (instead of a table) to highlight differences in the tails of uncertainties in each year? Do the differences in accuracy (e.g. lowest in 2010, highest in 2016) scale with the total area or number of glacier lakes?*

Response:

We greatly appreciate the suggestion of the Reviewer. Assuming an uncertainty of 1 pixel for the detected glacial lake boundaries, we calculated the systematic errors of each glacial lake over the whole HMA region (Table 1). For the years between 2008 and 2017, generally, statistical results of area uncertainty (%) is similar, the area uncertainty of each glacial lake ranged from 0.30% to 50%, with the mean value of all the glacial lakes falling around the 17%, and standard deviation around 11%. The maximum and mean value of area uncertainty for the glacial lakes in 2010 are the lowest, while for the year of 2016, the corresponding statistics are highest, this can be attributed to the different factors. The maximum of area uncertainty of glacial lakes is related with the shape and size of a certain lake (as can be seen from equation (1)), but its mean value is equal to the sum of the area uncertainties of each glacial lake divided by the total number, which depend on the total number of glacial lakes in a year, and also the shape and area of each lake

In order to highlight differences in the tails of area uncertainties in each year, following the Reviewer's very pertinent recommendation, in the revised manuscript we have added a new figure (Fig. 4a) that plotted the distributions of uncertainties from 2008 to 2017. Fig. 4b was also added for the better visualization of the relationship of area uncertainties against areas of all the glacial lakes in HMA (taking the results in 2017 as an example).

Besides, to give more detailed and comprehensive explanations about differences in the tails of area uncertainties of glacial lakes in each year, we have added some content in the **Section 4 Cross-validation and uncertainty estimate** of the revised manuscript. The detailed modifications are as follows:

Assuming an uncertainty of 1 pixel for the detected glacial lake boundaries, we calculated the systematic errors for the whole HMA region (Fig. 4). For the year between 2008 and 2017, the area uncertainty of each glacial lake generally ranged from 0.30% to 50%, with the mean value falling around the 17%, and standard deviation around 11% (Fig. 4a). The maximum and mean value of area uncertainty for the glacial lakes in 2010 are the lowest, while for the year of 2016, the corresponding statistics are highest, this can be attributed to the different factors. The maximum of area uncertainty of glacial lakes is related with the shape and size of a certain lake (as can be seen from equation (1)), but its mean value is equal to the sum of the area uncertainties of each glacial lake divided by the total number, which depend on the total number of glacial lakes in a year, and also the shape and area of

each lake. Besides, a close relationship can be found between the area uncertainties and the sizes of the glacial lakes (Fig. 4b). Most of the large glacial lakes (area $\geq 0.04 km^2$) have the mean area uncertainty of about 7%. This systematic errors were more significant for the small-sized glacial lakes. We measured glacial lake down to 0.0081 $km^2$ (nine pixels in Landsat imagery), where systematic errors calculated by equation (1) were ~50%.

[Figure]

**Fig. 4.** (a) Statistics of area uncertainty (%) of glacial lakes for the years from 2008 to 2017. (b) Relationship between area uncertainties and areas of all the glacial lakes in HMA in 2017.

*62. L178: The information from this sentence should go into the abstract.*

Response:

Thanks for this good suggestion. We have moved the information from this sentence into the abstract of the revised manuscript. The detailed modifications are as follows:

It can be observed that glacial lakes exhibited total area increases of 90.14 $km^2$ between 2008-2017, a +6.90% change relative to 2008 (1305.59 ±213.99 $km^2$).

*63. L179: 'A linear least-squares': Why did the authors refrain from using the Theil-Sen regression here?*

Response:

We gratefully thank the Reviewer for his/her constructive comments. Based on the final generated lake inventory data, we used the slope of linear regression of lake area (**over the grid cell of 1°×1°**) versus mapping year to qualify the yearly lake area changes during the study period. The approach to change analysis was predicted using a Theil-Sen estimator, which chooses median slope among all the derived fitted lines to smooth the annual time series of data. **For the whole HMA region**, we applied a linear least-squares fit to all the data to derive the annual increment of glacial lake area and number for the 10-year record. Due to the robustness for the trend detection and insensitivity to outliers, Theil-Sen estimator can effectively represent long-term area changes, it is also useful for the elimination of the effect

from differences in the sensor performance for the mapping of glacial lakes. Therefore, compared with conventional linear trends (such as linear least-squares), Theil-Sen estimator show better performance and is much more reliable to estimate year-to-year changes for the annual layers.

In the last version of manuscript, we are terrible sorry for our careless use of inconsistent regression fitting methods. In the **Section 5.1 Distribution of various types and sizes of glacial lakes** of revised manuscript, we have used the Theil-Sen regression method to estimate the yearly changes in lake area and number over the whole HMA region, Fig. 3a and related statistics have also been modified. The detailed modifications are as follows:

A Theil-Sen regression fit to all the data showed a mean expansion rate of 12 km$^2$ a$^{-1}$ for the 10-year record (Fig. 4a). Meanwhile, the estimated changes in glacial lake number from 2008 (12,593 lakes) to 2017 (15,348 lakes) showed an average increase of 306 lakes a$^{-1}$.

[Figure]

**Fig. 4. Annual glacial lake number and area. (a)** Total number and area of glacial lakes for HMA between 2008-2017. The annual increment is the slope of the trend of annual lake area and number. Altitudinal distribution (100-m bin sizes) of lake numbers for **(b)** all glacial lakes, **(c)** proglacial lakes, and **(d)** unconnected glacial lakes.

*64. L179 & L181: '18 km$^2$ a$^{-1}$' & '380 lakes a$^{-1}$': Please add confidence bands. In principle, all numbers and changes in Chapter 5.1 need error bounds because of the mapping uncertainty.*

Response:

We greatly appreciate the suggestion of the Reviewer. As for your first comment about adding the confidence bands for annual increase in lake area (18 km$^2$ a$^{-1}$) and number (380 lakes a$^{-1}$), we have added the computed P-value of the derived trends for the time-series glacial lake area and number to Fig. 3a. Besides, following the Reviewer's very pertinent recommendation, in the **Section 5.1 Distribution of various types and sizes of glacial lakes** of revised manuscript, we have completely modified the related statistics and trends by adding the error bars to area (both in the text and Fig. 3a), and confidence intervals to the trends and changes throughout the text. Please see the revised manuscript for more detailed information.

*65. L182: 'Fig. A3': unclear why this figure does not show all lake sizes. This could support the argument of the authors that smaller lakes have increased more in number than large lakes. On the other hand, do these two plot suggest that most of the increase in lake area is tied to the growth of large lakes? Data points in A3 could also have uncertainty bars given the mapping errors?*

Response:

We gratefully thank the Reviewer for his/her careful reading of the manuscript and for pointing out this issue. According to the response to your Comment 2, our team have been working hardly to update data of glacial lakes larger than 0.04 km$^2$ in the ten year records, and add their attribute information. Based on the new lake inventory, the related statistics, figures and analysis in the article have also been modified accordingly, which mainly focus on the relatively large lakes (area larger than 0.04 km$^2$). Meanwhile, we have weaken the analyses and conclusions about lakes that have an area less than 0.04 km$^2$ to the highest degree possible. Therefore, Fig. A3 has been deleted in the revised manuscript.

*66. L183-184: 'The increase of proglacial lakes was concentrated above 4900 m (Fig. 3c)': unclear whether this clearly follows from this plot, could be also at 4600 or 5600 m asl. Plotting the absolute differences of lake number for each elevation band could help*

Response:

We greatly appreciate the Reviewer's valuable suggestions very much. In the last version of manuscript, this conclusion may be not easily drawn from this plot indeed. Following the Reviewer's very pertinent recommendation, we have plotted the absolute differences of lake number (2008-2017) for the proglacial lakes along each elevation band, as shown in Figure III. From Figure III, we can clearly see that the number of proglacial lakes increased slightly between 3700m and 4200m during 2008-2017, the increase of was mainly concentrated above **4900 m**, indicated as the red line.

[Figure]

Figure III. Absolute differences of lake number (2008-2017) for the proglacial lakes along each elevation band. The red line indicates that the increase of lake number was mainly concentrated above 4900 m.

*67. L186: 'supraglacial lakes': why are these not shown here? Given the downwasting of debris-covered glacier in the Himalayas, for example, this type of lakes could account for a substantial amount of the increase in total lake number, no?*

Response:

We gratefully thank the Reviewer for his/her outstanding recommendations provided. With the accelerate downwasting of debris-covered glacier in the HMA region, a large amount of glacial lakes have emerged and expanded. Supraglacial lakes is one type of glacial lakes that formed in depressions on low-sloping parts of the surface of a melting glacier and are dammed by ice or the end-moraine or stagnating glacier snout. Typically, these lakes are highly dynamic, even ephemeral, and are quickly filled by glacier meltwater and quickly evaporated or drained off. They are **seasonal lakes**, which continue to change in their distribution and inundation area during a year. Mapping of supraglacial lakes creates great challenge for this study because the results largely depend on the annually available and valid observational data. If **images of different seasons or periods in a year** were used, the extraction results of supraglacial lakes are quite different.

In this study, we produced the glacier lake dataset in HMA region **in annual time steps** from 2008 to 2017. The optimal images with valid observations over the potential glacial lake area is very limited in each year. Moreover, the image dates are not consistent across the whole HMA region because of atmospheric disturbances, also the influences from image quality, ice and shadow that obscured the lakes, all greatly influence the results in the lake extent and number of supraglacial lakes. For such an annually resolved inventory, the obtained information about supraglacial lakes may be unstable. Therefore, the related statistics, figures and analysis in this section are mainly focused on the proglacial lakes and unconnected glacial lakes. As a consequence, supraglacial lakes were not shown here for the analysis of their altitudinal distribution and changes of lake numbers.

*68. L190-Fig 3a: why have some 1,000 lakes vanished between 2008 and 2009?*

Response:

We greatly appreciate the Reviewer's valuable comments very much. Based on the new lake inventory, the number of glacial lakes in the whole HMA region are 12593 for 2008, and 12698 for 2009, respectively, increased by a total of 105 glacial lakes.

*69. L194-195: Which 'non-parametric trend analysis' did the authors use?*

Response:

We gratefully thank the Reviewer for his/her careful reading of the manuscript and for pointing out this issue. Here 'non-parametric trend analysis' refers to Theil-Sen regression analysis introduced in the Section 3.3. We are so sorry for this unclear expression, and we have modified this sentence to make it much clearly. The detailed modifications are as follows:

Annual changes in glacial lakes were further analyzed spatially using a $1°\times1°$ grid over 22 mountain regions (Fig. 5a) using Theil-Sen regression analysis.

*70. L200: 'large patches': unclear formulation, please elaborate.*

Response:

Thanks for this good suggestion. We have modified the related sentence to make it clearly. The detailed modifications are as follows:

Glacial lakes in Nyainqentanglha, Gangdise Mountains exhibited area loss and gain in some regions.

*71. L201-203: What is the reason for the two bubbles with the large decrease in lake area (Fig. 4b), while all surrounding bubbles show a positive trend? Is this because one (or more) large lake(s) shrunk or burst out?*

Response:

We gratefully thank the Reviewer for his/her constructive comments. The two bubbles with the large decrease in lake area are indeed caused by the shrunk of few large glacial lakes. Subtle changes in the large glacial lakes can have a big impact on the annual area changes of glacial lakes over the grid cell of $1°\times1°$. In the last version of manuscript, some huge glacial lakes with an area of larger than $6.5km^2$ have also been mapped in the Hi-MAG database, this is because these lakes all located in the range of 10 km from the nearest glacier terminus according to the definition of glacial lake in HMA. However, in the dataset produced by Wang et al., 2020, these huge lakes were removed. For the sake of a reliable comparative analysis between different studies under the equal conditions, glacial lakes with an area of larger than $6.5km^2$ were not considered for the calculation and analyses in the revised manuscript. Based on the new lake inventory, Fig. 4 and related statistics and analysis have also been completely modified, the abnormal bubbles in Fig. 4 no longer exist. The detailed modifications of Fig.4 are as follows:

[Figure]

**Fig. 5. Glacial lake area changes and area distribution. (a)** Geographic coverage of mountain ranges in HMA. **(b)** Annual rate of change in lake area (2008-2017) on a 1°×1° grid. The size of the circle for the area in 2017. **(c)** Proportional areas of four types of glacial lakes in 2017. **(d)** Area of different sizes of glacial lakes in 2017. The terrain basemap is sourced from Esri (© Esri).

*72. L210-211: What is happening in regions with shrinking lakes? Do some of these fall dry or burst out?*

Response:

We greatly appreciate the Reviewer's valuable comments very much. As for the regions with negative lake growth, during the past ten years, many glacial lakes experienced shrinkage, dry up or outburst events. For example, some seasonal and highly dynamic lakes such as supraglacial lakes, their shapes and extent continue to change during a year. Because of the high sensitivity to regional climate and glacier runoff, their life is short, even ephemeral, and are quickly filled by glacier meltwater and quickly evaporated or drained off. Consequently, some supraglacial lakes drop during the post-monsoon and decreases to negligible during the winter months. Unconnected glacial lakes have been detached from ice contact due to glacial recession. As the interaction with the glacier gradually weakened, part of the water source supplied by glaciers is reduced. Also, combined with the effects from atmospheric warming and decrease of precipitation, regions mainly consist of unconnected glacial lakes exhibit the negative area change.

The above is just some general and global analysis. Based on the results obtained in this study, for a better understanding of how many glacial lake burst out, how many glacial lakes

decreased in area, as well as the reason and degree of shrinkage of lakes in the particular regions, it is necessary for us to conduct some more in-depth discussions and analyses in our future studies.

Finally, in the **Section 5.1 Distribution of various types and sizes of glacial lakes** of revised manuscript, we have added some content to provide more detailed information about what is happening in regions with shrinking lakes. The detailed modifications are as follows:

In the negative lake growth (shrinkage) regions of Eastern Tibetan Mountains and Hengduan Shan, the unconnected glacial lakes were dominantly occupied. As the interaction with the glacier gradually weakened, part of the water source supplied by glaciers is reduced. Also, combined with the effects from atmospheric warming and decrease of precipitation, regions mainly consist of unconnected glacial lakes show the decreasing trend in area.

*73. L211-214: Very nice findings. These numbers could find their way into the abstract?*

Response:

Thanks for this good suggestion. We have moved the information from this sentence into the abstract of the revised manuscript. The detailed modifications are as follows:

Our results demonstrate proglacial lakes are a main contributor to recent lake evolution in HMA, accounting for 62.87% of the total area increase (56.67 km$^2$). Proglacial lakes in the Himalaya ranges alone accounted for 36.27% of the total area increase (32.70 km$^2$).

*74. L215: 'the largest area growth of lakes occurred in areas with relatively large proportion of small glacial lakes': is there a way to measure this notion?*

Response:

We greatly appreciate the Reviewer's valuable comments very much. In the last version of manuscript, the related statement is not rigorous indeed, which needs further support from detailed statistics. Therefore, following the Reviewer's very permanent recommendations, we have calculated the area of different sizes of glacial lakes in 2017 for the regions with large area growth of rate (Table I). From Table I, it can be found that in the areas that show rapid lake area increases (higher than 0.23 km$^2$a$^{-1}$), glacial lakes with a size of less than 0.16 km$^2$ occupied more than 30% of total area. Especially for some of these regions, the area of small glacial lakes (≤0.16 km$^2$) even accounts for 69.47% of the total area.

In addition, we have modified this sentence by providing more statistical support in the revised manuscript. The detailed modifications are as follows:

We also noted the large area growth of lakes occurred in areas with relatively large proportion of small glacial lakes, mainly due to rapid growth of existing lakes and new lake formation (Fig. 5d). For example, in some areas of Central and Eastern Himalaya, Nyainqentanglha that have large annual increase in lake area (higher than 0.23 km$^2$a$^{-1}$), glacial lakes with a size of less than 0.16 km$^2$ occupied more than 30% of total area. Especially for the region in Nyainqentanglha, the area of small glacial lakes (≤0.16 km$^2$) even accounts for 69.47% of the total area (Table A3).

**Table A3. Area of different sizes of glacial lakes in 2017 for some regions with large area growth of rate. The unit of area is km².**

| Lake grid ID (Mountain ranges) | 69 (N) | 116 (CH) | 274 (WH) | 71 (N) | 48 (H) | 74 (N) | 72 (N) | 14 (EH) | 13 (EH) | 39 (CH) | 15 (EH) |
|---|---|---|---|---|---|---|---|---|---|---|---|
| ≤0.01km² | 0.18 | 0.28 | 0.20 | 0.16 | 0.22 | 0.16 | 0.17 | 0.23 | 0.16 | 0.32 | 0.33 |
| 0.01km²-0.02km² | 0.85 | 1.51 | 1.29 | 0.71 | 1.43 | 1.08 | 1.37 | 1.45 | 1.45 | 1.49 | 2.46 |
| 0.02km²-0.04km² | 1.69 | 2.16 | 2.22 | 1.79 | 3.24 | 2.09 | 2.29 | 2.24 | 2.06 | 2.72 | 4.14 |
| 0.04km²-0.08km² | 1.78 | 3.19 | 2.98 | 2.20 | 5.30 | 3.38 | 4.45 | 2.77 | 2.69 | 3.66 | 7.16 |
| 0.08km²-0.16km² | 1.91 | 5.38 | 3.87 | 2.86 | 4.81 | 4.03 | 5.06 | 3.75 | 4.33 | 5.00 | 13.16 |
| 0.16km²-0.32km² | 1.81 | 4.53 | 2.23 | 2.76 | 4.62 | 5.55 | 5.81 | 2.91 | 3.90 | 5.66 | 11.62 |
| 0.32km²-0.64km² | 1.01 | 5.37 | 1.77 | 1.79 | 3.88 | 1.75 | 3.81 | 5.72 | 3.99 | 7.13 | 12.37 |
| 0.64km²-1.28km² | 0.00 | 2.94 | 0.00 | 1.38 | 2.82 | 2.96 | 4.43 | 0.96 | 7.10 | 8.97 | 7.74 |
| ≥1.28km² | 0.00 | 7.22 | 4.19 | 0.00 | 11.46 | 3.17 | 2.59 | 6.07 | 1.40 | 6.06 | 12.00 |
| Total area (km²) | 9.22 | 32.58 | 18.76 | 13.66 | 37.76 | 24.17 | 29.99 | 26.10 | 27.09 | 41.00 | 70.96 |
| Total area (≤0.16km²) | 6.41 | 12.52 | 10.57 | 7.72 | 14.99 | 10.74 | 13.35 | 10.45 | 10.69 | 13.18 | 27.24 |
| Total area % (≤0.16km²) | 69.47 | 38.43 | 56.32 | 56.56 | 39.70 | 44.45 | 44.52 | 40.03 | 39.47 | 32.15 | 38.39 |
| Annual area increase (km²a⁻¹) | 0.23 | 0.28 | 0.28 | 0.29 | 0.32 | 0.41 | 0.42 | 0.49 | 0.70 | 0.74 | 0.94 |

75. *L216-217: 'many large-lake dominated areas exhibited decreased or nearly unchanged lake extent': not sure whether Fig. 4 fully supports this conclusion. In the Eastern Himalayas, for example, small glacier lakes account for less than 10 % for the entire lake area in that region (Table A4), so that most of the growth must be tied to the largest lakes, right?*

Response:

We gratefully thank the Reviewer for his/her comments and suggestion. According to the response to your Comment 71, in the last version of manuscript, some huge glacial lakes with an area of larger than 6.5km$^2$ have also been mapped in the Hi-MAG database. Subtle decrease or changes in the large glacial lakes can have a big impact on the annual area changes of glacial lakes over the grid cell of 1°×1°. Therefore, for the areas that large-lake dominantly occupied, decreased or nearly unchanged lake extent can be observed. In the updated version of Hi-MAG database, glacial lakes with an area of larger than 6.5km$^2$ were all deleted, and this conclusion is correspondingly weakened.

*76. L218-222: Unclear content in these sentences, please reformulate*

Response:

Thanks for this suggestion. Similar to the response to your Comment 75, the conclusions and content shown in these sentences are also weakened based on the updated version of lake inventory. Consequently, we have deleted these sentences in the revised manuscript.

*77. L230-Fig4b: not sure whether annual trends are useful here because, as far as I understood, the authors assume that lake area scales linearly with time in each 1°x1° bin. Yet the authors provide neither a measure of uncertainty for these annual changes, nor is it clear whether these changes occur linearly in time or that the rate of change could be affected by outliers. Total changes in glacier lake area between 2008 and 2017 could probably make more sense here.*

Response:

We greatly appreciate the Reviewer's valuable comments very much. We realize this is a very important point that we may not explain clearly in the last version of manuscript.

According to your comment about whether the glacial lake area occur linearly with time, also for the long time series of analysis of lake changes over a large scale area, Zhang et al., 2019 studied the regional differences of lake evolution across China during 1960s–2015, and confirm that multi-decadal lake measurements shown **significant linear regression trends, which was then used for the estimation of lake area change per year** (Zhang et al., 2019). In this study, annual changes in glacial lakes were analyzed spatially using a 1°×1° grid over 22 mountain regions using Theil-Sen regression analysis. Theil-Sen estimator chooses the median slope among all the derived fitted lines, can effectively represent long-term area changes due to its robustness for the trend detection and **insensitivity to outliers**. As for the measure of uncertainty for these annual changes, a **nonparametric Mann-Kendall trend test was used to detect and further confirm the statistical confidence or significance level by the linear regression results**. All the estimated trends fall within the **90% confidence intervals**. The upper and lower change estimates that satisfy the 90% confidence interval for the slope were also derived over the whole HMA region, ==please see Fig. A2== for the detailed information.

Just as the reviewer pointed out, the calculated rate of changes in the glacial lake area is the approximate estimations of the linear trend. Total changes in glacier lake area between 2008

and 2017 is measured in absolute values, and probably more accurate in quantitative degree. While in this study, we produced an annually resolved inventory, aiming to investigate the detailed evolution patterns and annual time series of glacial lake changes in the whole HMA for the past several years. The total changes was only calculated based on the two time periods (2008 and 2017), which not fully considered the mapping results of glacial lake in annual time steps from 2008 to 2017. More importantly, for the roughly weigh and comparison the **annual loss and gain** of glacial lake area in different regions, we used robust linear regression to calculate the yearly lake area changes over each grid cell of 1°×1°, which is not possible with simple two-stage lake mapping.

Finally, in the revised manuscript, we have added some content to give more detailed information about the measure of uncertainty for the annual changes in glacial lake area and significance test of the linear regression results. The detailed modifications are as follows:

==For the glacial lake area time series in each 1°×1°grid, we applied the Theil-Sen estimator to smooth the annual time series of data and derive the slope (annual change) of the trend. A Mann-Kendall trend test was used to detect and further confirm the statistical confidence by the linear regression results. All the estimated trends fall within the 90% confidence intervals. The upper and lower change estimates that satisfy the 90% confidence interval for the slope were also derived over the whole HMA region (Fig. A2).==

[Figure]

**Fig. A2. Annual changes in lake area between 2008 and 2017 on a 1°×1° grid. The (a) upper and (b) lower slopes represent the 90% confidence interval.**

*78. L203: where does Fig. A4 show that the increase in lake area is 'due to retreat and thinning of debris-covered glaciers'?*

Response:

We are terrible sorry for the mistake about the wrong location of Fig. A4. Fig. A4 shows the density (number per 100 km²) distribution of glacial lakes in 2017, and should be put after the

sentence 'exhibiting both a high density of 46 glacial lakes per 100 km$^2$ in 2017'. The detailed modifications are as follows:

Between 2008 and 2017 Central Himalaya's glacial lake area increased by 27.09 km$^2$ (Table A1), exhibiting both a high density of 47 glacial lakes per 100 km$^2$ in 2017 (Fig. A4) and rapid growth, +0.94 km$^2$ a$^{-1}$, in lake area due to retreat and thinning of debris-covered glaciers (Song et al., 2016).

*79. L207-208: Fig. 4c does not show any trends*

Response:

We are sorry for missing another figure (Fig. 4b) here. In the last version of manuscript, it can be seen from Fig. 4b and Fig. 4c that glacial lakes exhibited different expansion trends for different lake types and supraglacial and ice-marginal lakes have relative few coverage areas comparing with proglacial and unconnected lakes. In the revised manuscript, we have added Fig. 4b and Fig. 4c after this sentence to make it easy to understand. The detailed modifications are as follows:

We found that glacial lakes exhibited different expansion trends for different lake types and supraglacial and ice-marginal lakes have relative few coverage areas comparing with proglacial and unconnected lakes (Fig. 5b and Fig. 5c).

*80. L220-227: OK – and what about the Pamir Alay, Western & Eastern Kunlun, and Eastern Himalaya, where the contribution to the total growth in lake area is much smaller from small lakes compared to large lakes, according to Table A2? What I learned from this table is, that there is some variability in the lake-size distribution, which itself could be more emphasized. I have more the feeling that there are few (or no?) general rules that could help explaining the regional distribution of glacier lakes and associated growth patterns. If that feeling is wrong, I would appreciate simple correlation plots that check for such notions.*

Response:

We gratefully thank the Reviewer for his/her careful reading of the manuscript and for pointing out this issue. According to the response to your Comment 2, our team have been working hardly to update data of glacial lakes larger than 0.04 km$^2$ in the ten year records, and add their attribute information. Based on the new lake inventory, the related statistics, figures and analysis in the article have also been modified accordingly, which mainly focus on the relatively large lakes (area larger than 0.04 km$^2$). Meanwhile, we have weaken the analyses and conclusions about small glacial lakes that have an area of less than 0.04 km$^2$ to the highest degree possible. Therefore, Table A2, Table A3 and this paragraph have been deleted in the revised manuscript.

Just as the reviewer pointed out that the variability in the lake-size distribution itself could be more emphasized, rather than digging into few general rules that explaining the regional distribution of glacier lakes and associated growth patterns. Following the Reviewer's very pertinent recommendations, we have made efforts to thoroughly modify the whole text and paid more attention to analyze the spatiotemporal variability in the area and number of the glacial lake itself during the past ten years.

*81. L230: Fig 4b: Colors around zero could be changed to white and negative value to a darker shade of blue for more contrast in the bubbles.*

Response:

Thanks for this good suggestion. For better contrast between positive and negative values in the bubbles, we have modified Fig. 4b by changing the colors around zero to white and negative value to a darker shade of blue. The detailed modifications are as follows:

[Figure]

**Fig. 5. Glacial lake area changes and area distribution. (a)** Geographic coverage of mountain ranges in HMA. **(b)** Annual rate of change in lake area (2008-2017) on a 1°×1° grid. The size of the circle for the area in 2017. **(c)** Proportional areas of four types of glacial lakes in 2017. **(d)** Area of different sizes of glacial lakes in 2017. The terrain basemap is sourced from Esri (© Esri).

*82. L242-244: Could it be that the number of lakes in the smallest size class (< 0.01 km²) remains constant in most regions because these lakes climb into the next higher size class from year to year? If this holds true, could this mean that the lowest size class tells us more about the annual production rate of lakes in that region?*

*L247-248: Binning should be equal, so that size classes are comparable. Now, the smallest class spans 0,0919 km², the medium 0.01 km², and the largest 0.02 km². Linear regression should have confidence bounds, given that the positive trends in some panels (e.g. K, CT, WH, EHK) could be due to outliers. All panels could have the same Y-axes to show differences in lake abundance between regions*

Response:

We gratefully thank the Reviewer for his/her careful reading of the manuscript and for pointing out this issue. Similar to the response to your Comment 2, Comment 65 and Comment 80, based on the new lake inventory, the related statistics, figures and analysis in the article have been modified accordingly, which mainly focus on the relatively large lakes (area larger than 0.04 km$^2$). Meanwhile, we have weaken the analyses and conclusions about small glacial lakes that have an area of less than 0.04 km$^2$ to the highest degree possible. Therefore, information about **Section 5.2 Inter-annual variability of small glacial lakes for different mountain regions** have been deleted in the revised manuscript. But in our future work, our team will make great effort to supplement and improve the analysis about tiny glacial lakes, and continuously optimize and share glacial lake data with long time scales.

*83. L256: $R^2=0.53$: I couldn't find this number in Table A5. Also, which type of correlation coefficient is calculated here? The caption of table A5 also talks about R, not $R^2$.*

Response:

Apologies for this mistake. The Tables A5 and A6 contain the Pearson coefficients of correlation (R), and not coefficients of determination ($R^2$). This has now been made consistent between the text and tables, with R reported only.

*84. L257: 'larger ice-contact proglacial lakes imply larger calving-front interactions': source?*

Response:

Sentence has been clarified and supporting references provided.

*85. L259: What and how long is 'a typical glacier response time'?*

*L260-261: Content of this sentence not fully clear, please reformulate.*

Response:

Typical glacier response times cover a large range starting from 10 years, to over 150 years. We have added text on this and provided a reference.

*86. L264: 'geomorphic, topographic and climate parameters': more specifically?*

Response:

This specificity is given in the tables and figures immediately referred to in the next sentences.

*87. L266: what does this correlation tell us?*

Response:

The correlation reconfirms, and demonstrates over a much larger scale that pro and supraglacial lakes predominately develop on debris-covered glaciers. Text on this, with supporting reference, has been added.

*88. L267: Not the best statistical approach to exclude data points that we don't like.*

Response:

The point being made here is that we would expect the Karakorum to behave differently as it is a known anomaly. A sentence explaining this came later, but has now been moved to make this point earlier. We deliberately report statistical results both with/without including the Karakorum, so we don't believe the reviewers criticism is justified.

*89. L267-269: What has the correlation between debris cover and glacier length to do with glacier lakes? Where is a proof for the notion that 'low-gradient glaciers favour supraglacial and proglacial lake formation'?*

Response:

In the previous paragraph we highlight the link between glacier lakes and debris-covered glaciers. Here we take that one step further, and note that debris cover (and therefore lake growth) is most pronounced on long, flat glaciers. This is not a new finding, but is worth noting here as we can use our large-scale data to quantitatively demonstrate these relationships. We have added references to papers that have also noted this association. A statistically significant correlation is also seen between glacier length and lake area (Table A6).

*90. L272: 'Some adjacent regions': which ones exactly? 'comparable', 'large', 'substantial': please be more specific.*

Response:

This statement was referring to data given in Table A6. The sentence has been revised, and specific examples from Table A6 are now given.

*91. L273: 'longer-term climate trends': on which scales? Years, decades, centuries?*

Response:

The statement refers to the trends shown in Figure 6, i.e, multi-decadal. The text has been clarified.

*92. L274: 'rapid warming': warming seems to be present across the entire study region except the Karakoram. 'decreased precipitation': precipitiation is unchanged in the Eastern and Western Himalaya, where lakes have grown most in the study period. The line of argumentation is not clear here.*

Response:

The warming has not been uniform, and the point made in the previous sentences is that in some areas where warming has been most rapid (e.g. Eastern Himalaya), lakes are large despite other factors not being so ideal (e.g., a lower abundance of debris covered glacier tongues). Conversely, in Western Himalaya we see the opposite – less warming and less lakes, despite lots of long debris covered glaciers.

*93. L275-277: Unclear, please be more specific ('further west and north', 'similar'?) and reformulate.*

Response:

A specific example has been added to the text. The reader can refer to the cited figures for further examples.

*94. L279-282: Vague conclusion – these arguments could explain anything, in the light of the high regional variability in all the predictors that the authors have shown. There is a large body of literature that has worked on the size distribution and regional growth pattern of glacier lakes (e.g. Song et al., 2016, 2017). However, this literature is largely left untouched here. By the way, until here, no references have been given for the datasets used in the correlations.*

Response:

Concluding statement for this section has been rewritten. The point that we try to highlight is the close association between surpa- and proglacial lake formation with large debris covered glaciers. This is not a new finding, but our large dataset enable us to confirm observations previously reported at regional scales. We now refer to the earlier work of Song et al., and in particular the processes of lake enlargement on debris covered tongues that were reported three.

References/sources for all datasets used in the correlations are now provided in Table A5. In addition, the primary glacial datasets are cited directly in the text.

*95. L282: Dan and CLAGUE, 2011: wrong citation.*

Response:

Citation has been removed.

*96. L291: 'datasets'.*

Response:

Thanks for this suggestion. We have replaced 'dataset' by 'datasets' in the revised manuscript.

*97. L293: compared to which dataset? Reminds me also a bit of comparing apples with oranges, given these studies used different minimum mapping units, sizes of the study regions, buffers around lakes, types of lakes analysed and so on.*

Response:

We greatly appreciate the Reviewer's valuable comments very much.

Firstly, we are so sorry for the wrong location of the citation again. Compared with the results from Zhang et al., 2015, Hi-MAG lake number was 7268 higher and area was 644.26 $km^2$ higher than the estimation for the Tibetan Plateau. We have modified this sentence in the revised manuscript to make it clearly. The detailed modifications are as follows:

Hi-MAG lake number was 7268 higher and area was 644.26 $km^2$ higher than the estimation for the Tibetan Plateau (Zhang et al., 2015).

Secondly, when we used to produce the glacial lake database and wrote the last version of manuscript, no relevant data were released publicly. Because the dynamic cross-validation procedures are essential to maximize the quality of glacial lake change detections, we have

only made a rough comparison of the statistical results between our Hi-MAG database and other studies. However, at present, there are no uniform definition and mapping standard for the glacial lakes yet, which result in the minimum mapping units, buffer distance around lakes and types of lakes used are different for these studies. Also the extent of the study regions are not completely consistent, just as the Reviewer pointed out, the comparison with other lake dataset is not strict indeed.

To clearly show benefits and challenges remaining in this study, and also for more rigorous cross-validation, in the **Section 6.1** of revised manuscript, we have conducted a deep analysis and comparison between our Hi-MAG dataset and glacial lake inventory from Wang et al., 2020, who mapped glacial lake larger than 0.0054 $km^2$ in a much larger High Mountain Asia region in two time periods (1990 and 2018), and made their data public recently. The comparative discussions with other studies (e.g. Nie et al., 2017;Zhang et al., 2015;Pekel et al., 2016a) can be used as a general reference. Please see the revised manuscript for more details.

*98. L296: 'a much larger lake area': how much larger?*

Response:

We are so sorry for this inappropriate expression. We have deleted this sentence in the revised manuscript.

*99. L300: How did the authors convert the gridded data from GSW to polygons that match the Hi-MAG database?*

Response:

We gratefully thank the Reviewer for his/her careful reading of the manuscript and for pointing out this issue. GSW and Hi-MAG database have different data formats, which are grid and vector, respectively. To match the formats of the two datasets, we have converted the lake polygons in the Hi-MAG dataset to the gridded data. In the **Section 6.1 Comparison with other lake datasets** of revised manuscript, we have added some text to explain the conversion between two dataset clearly. The detailed modifications are as follows:

For the sake of a reliable comparative analysis, lake polygons in the Hi-MAG dataset were converted into the grid format, and glacial lakes in the GSW were further extracted using the range of glacier buffer (10 km).

*100. L300-301: 'more glacial lakes in the Himalaya, Eastern Hindu Kush, and Tien Shan, and fewer in Eastern Pamir and Western Kunlun Shan': please be more specific.*

Response:

Thanks for this good suggestion. According to the response to your Comment 97, because the minimum mapping units, buffer distance around lakes and types of lakes used are not completely consistent for the different studies, we have made a **qualitative comparison** of the statistical results between our Hi-MAG database and study from Pekel et al., 2016. In the **Section 6.1** of revised manuscript, more detailed **quantitative comparisons** were implemented between Hi-MAG database and glacial lake inventory produced by Wang et al., 2020. Besides, we have also modified this sentence to make it formulated clearly. The detailed modifications are as follows:

In addition, we qualitatively compared the lake extent between publicly available high-resolution Global Surface Water (GSW) dataset and our Hi-MAG database summed by mountain range in 2015.

Hi-MAG detected more glacial lakes in the Himalaya, Eastern Hindu Kush, and Tien Shan, and fewer in Eastern Pamir and Western Kunlun Shan.

*101.    L304: 'significant number': how do the authors measure significance here?*

Response:

We are terrible sorry again for the inappropriate use of 'significant', which may confuse the readers. The word 'significant' here is different from the concept of statistical significance, it means that 'numerous' or 'a large amount of' glacial lakes. In the revised manuscript, we have modified this expression to make it easy to understand. The detailed modifications are as follows:

While there are numerous glacial lakes from an open water perspective,

*102.    L305: 'only part of lakes are formed by glacier meltwater': how do the authors measure the contribution of glacier meltwater to the total volume of a glacier lake?*

Response:

We greatly appreciate the Reviewer's valuable comments very much. We realize this is a very important point we may not explain clearly. In the last version of manuscript, what we originally meant was that although GSW detected more glacial lakes in some sub-regions of HMA, actually part of them are river segments, which is not formed by glacier meltwater. We have completely modified this sentence in the revised manuscript to make it clearly. The detailed modifications are as follows:

While there are numerous glacial lakes from an open water perspective, actually part of them are river segments.

*103.    L304: the GSW does not 'define' glacier lakes per se, but maps surface water. Please clarify.*

Response:

Thanks for this good suggestion. We have carefully modified this sentence in the revised manuscript to clarify that GSW does map surface water rather than glacial lakes. The detailed modifications are as follows:

The glacial lake area observed in our lake dataset in the Eastern Pamir and Western Kunlun Mountains does not conform to the mapped surface water in the GSW for these sub-regions.

*104.    L310: 'errors could exist in either dataset': a surprising notion given that 10 independent experts have mapping glacier lakes for the Hi-MAG database. Also, what causes these 'similar reflectance from the adjacent land surfaces' to be so special in the Tien Shan compared to all other regions?*

Response:

We gratefully thank the Reviewer for his/her comments and suggestion.

Tien Shan Mountains, which are located inland, are the only large mountains in the world that surrounded by huge deserts. It has a temperate continental arid climate, in the cold season, mountains, basins and valleys are all covered by deep snow below the altitude of 3000m above sea level, while in summer, it favors the formation of rainy and snowy climate in the region above 3000 m (Wang et al., 2013a;Shen et al., 2009). Being in this frost zone, large amounts of ancient glacial deposits have also been accumulated (Glazirin, 2010). The unique climatic and terrain conditions have developed glacial lakes that are abundant at small sizes. Also because of the characteristics of the source glacier and lake bed, and due to the water depth and sediment influx, these lakes appear turbid over a wide spectral range, this produces heterogeneous reflectance from the lakes and adjacent land surfaces. The regional heterogeneity in image bring great challenges for automated detection of glacial lakes. Errors could exist in datasets produced by automatic classification. For the further refinement of individual glacial lake, in this study, a detailed manual editing was conducted by ten independent experts, serving as an effective supplement to initial glacial lake extraction. So we were not relying exclusively on the results of automatic classification.

However, for the generation of such an annually resolved glacial lake inventory, the optimal images with valid observations over the potential glacial lake area is very limited in each year. Also the influences from image quality, ice and shadow that obscured the lakes, all greatly influence the extraction results of the lake extent and thus inevitably produced the detection errors, especially for the problematic regions like Tien Shan Mountains, where glacial lakes have outlines that remain unclear and are easily mixed with the glacier snow/ice cover. By combining with other types of optical or SAR data during the same periods, further improvement of the extraction results of glacial lakes in Tien Shan will be an urgent issue for us to consider in the next step.

Finally, in the revised manuscript, we have rewritten the related content to give more information about why the errors could exist in either dataset and why Tien Shan is a special region that glacial lakes show similar reflectance with the adjacent land surfaces. The detailed modifications are as follows:

There is little agreement for Tien Shan, where the weather is rainy and snowy in the region above 3000 m, and large amounts of ancient glacial deposits have been accumulated. Here glacial lakes are featured by small size, due to the influence of source glaciers and lake beds, as well as the water depth and sediment inflow, glacial lakes appear heterogeneous reflectance in the image. Errors could exist in datasets produced by automated classification, but we also did a detailed manual editing, so we were not relying exclusively on automatic classification.

*105. L311-312: 'Karakoram regions have fewer glacial lakes in our estimate': because? The GSW has overestimates surface water on debris covered glaciers, may this helps explaining the difference?*

Response:

We gratefully thank the Reviewer for his/her valuable comments and suggestions. In the GSW dataset, the transient melt ponds and streams on the debris-covered glaciers are also identified here as permanent water surfaces, result in the overestimation of the surface water in the Karakoram regions. Therefore, it looks like that our Hi-MAG dataset represented fewer glacial lakes in the Karakoram regions. In the revised manuscript, we have added some text to give

more explanations about why Karakoram regions have fewer glacial lakes in our estimate. The detailed modifications are as follows:

==Karakoram regions seem to have fewer glacial lakes in our estimate, owing to the overestimation of surface water on debris covered glaciers in the GSW dataset.==

*106.   L343-346: 'relatively good performance for areas having simple lake characteristics and environmental backgrounds'; 'diverse climatic conditions, physical properties and surrounding environments': please be more specific.*

Response:

Thanks for this good suggestion. To make the related expressions much more specific and clearly, we have rewritten these sentences in the revised manuscript. The detailed modifications are as follows:

==Such methods result in relatively good performance for lake areas that remain clear and having homogeneous reflectance in the image, but do not allow for continental-scale glacial lake mapping that have spectral interference from the other objects such as glaciers, snow, clouds, turbidity and sedimentation characteristics of glacial lake itself, or the atmospheric interference and terrain effects.==

*107.   L360: Please add the year when these images were shot. In each panel, the authors could also show the lake polygons mapped for this year for each lake type.*

Response:

Thanks for this good suggestion. We have added the year of these images in the caption of Fig. A1, and modified Fig. A1 by overlaying lake polygons for each lake type. The detailed modifications are as follows:

[Figure]

(a)                    (b)                    (c)                    (d)

**Fig. A1. Examples of the various types of glacial lake found in the HMA: (a) pro-glacial lakes, which are connected to the parent glacier and usually impounded by a debris dam (usually a moraine or ice-cored moraine); (b) supraglacial lakes (denoted by the red rectangle) which develop on the glacier surface; (c) unconnected glacial lakes; and (d) ice-marginal lakes that distributed on the edge of a glacier. ==Background images were acquired from © Google Earth, and were shot in 2009, 2011, 2012 and 2014, respectively. Glacial lake outlines for each type are shown in blue color.==**

*108. L366: Please avoid using rainbow color scales*

Response:

Thanks for this good suggestion. We have modified Fig. A2 by using another color scale. The detailed modifications are as follows:

[Figure]

**Fig. A2. Annual changes in lake area between 2008 and 2017 on a 1°×1° grid. The (a) upper and (b) lower slopes represent the 90% confidence interval.**

*109. L406: Why does this need to be acknowledged given that these researchers are coauthors in this study?*

Response:

We are terrible sorry for this mistake. These researchers are co-authors indeed, and do not need to be acknowledged in this section. We have deleted this sentence in the revised manuscript.

**Response to Comments by Reviewer 2:**

*Chen et al. used Google Earth Engine to map glacial lakes in High Mountain Asia (HMA) from 2008 to 2017 with Landsat imagery. Their data is given in annual time steps, which so far is the highest temporal resolution of glacial lake inventory for the HMA. Thus, this kind of dataset if with fine quality could be particularly useful for scientific researches in changes of the cryosphere of the HMA as well as for related assessments and evaluations on the hydrological responds and glacial lake outburst flood risks under a changing climate in the HMA.*

We greatly appreciate the suggestions of the Reviewer for his/her accurate summary of the main contributions of our work and for the outstanding recommendations provided. Following the Reviewer's very pertinent recommendations, in the revised manuscript and the following text we have made modifications according to these questions. We thank again the Reviewer for his/her careful handling of our manuscript and for the constructive suggestions provided, which greatly helped us improve the technical quality and presentation of our manuscript.

1. *Noticed that recently there published a similar dataset produced by Wang et al. (2020), which includes two periods (1990 and 2018) of glacial inventory for the HMA and currently is also under review for the ESSD. The later one used a more traditional method and probably involved more extensive manually inspection during their investigations. When comparing these two datasets for the closest period (2017 of Cheng et al. and 2018 of Wang et al. 2020), I found there is a very large discrepancy (Table 1) in their results, although they have claimed that they used similar (not the same) area threshold (0.0081 $km^2$ of Chen et al. 2020 and 0.0054 $km^2$ of Wang et al. 2020) and distance threshold (within a 10 km from the nearest glacier terminus) for the lake mapping.*

Table 1: Differences between two datasets of HMA glacial lake inventory

| Source | Year | Total Numbers | Area_Sum (km²) | Area_Max (km²) | Area_Mean (km²) | Area_Min (m²) | Notes |
|---|---|---|---|---|---|---|---|
| Wang et al. 2020 | 2018 | 28953 | 1955.939 | 6.465 | 0.067 | 5400 | Altai mountains included |
| Wang et al. 2020 | 2018 | 22219 | 1672.479 | 6.465 | 0.075 | 8100 | Altai mountains excluded and area larger than 8100 m² |
| Chen et al. 2020 | 2017 | 14477 | 1635.939 | 26.598 | 0.113 | 8100 | Altai mountains excluded |

*In general, dataset of Chen et al. has missed a quite number of glacial lakes when comparing their results with the Wang et al. 2020's. In addition, they excluded the Altay and Sayan mountains, which actually should be included for an inventory study for the HMA. Even if excludes Altay form Wang et al.'s result, there still exist remarkable discrepancy in total numbers (nearly 7800 lakes) and total area (~37 $km^2$) between each other (Table 1). I agree with the Reviewer #1 that the missing inventory by Chen et al. is far from the range of uncertainty, but it was indeed errors due to the lack of systematic experts' check through the results, which were greatly depended on the methods they used when applying their procedures in Google Earth Engine.*

*The authors should pay more attention to how to control the quality of their dataset. Although the Google Earth Engine offer the opportunity to calculate lake inventory with a higher temporal resolution, I still strongly recommend they check through their result for*

*each year, or maybe improve their methods to avoid errors as much as possible. Then a comparison between their results with existing datasets (such as datasets published by Zhang et al. 2015 or Wang et al. 2020) is necessary for audients judgement of the data quality. They did have do some comparison between their results with the Global Surface Water (GSW) dataset, but both these two were calculated by Google Earth Engine.*

Response:

We greatly appreciate the Reviewer's valuable comments and suggestions. This paper focus on obtaining the comprehensive knowledge of the distribution, area of glacial lakes, and also quantification of variability in their size and type at high resolution in HMA. We develop a HMA Glacial Lake Inventory (Hi-MAG) database to characterize the annual coverage of glacial lakes from 2008 to 2017 at 30 m resolution. This is the first glacial lake inventory across the HMA with annual temporal resolution, it can provide details for different types of glacial lakes and evolution patterns. Although the method of lake mapping was automatic, for quality control every lake polygon was inspected and was manually edited where needed. The related attribution were also added for each lake. This is a huge amount of work for the mapping of nearly 140,000 glacial lakes over ten time periods from 2008 to 2017.

In the last version of manuscript, the mapping work still have some deficiencies and was not fully reflected in the quality of the manuscript indeed. Therefore, we have made considerable efforts to fill these defects, which is mainly manifested in the four aspects including **supplement of glacial lake data, detailed comparison with other lake inventories, quantitative uncertainty analysis of lake area and adding confidence intervals for trends, details about the whole mapping procedure.** We have made the following improvements in our study:

**1) We have been working hardly to update data of glacial lakes in the ten year, and add their attribute information. Based on the new lake inventory, we have also carefully modified the tables, figures, the related analysis and made reliable conclusions**.

In this study, we produced the glacial lake dataset in High Mountain Asia in annual time steps from 2008 to 2017. The number of glacial lakes in each year is more than 12,000 totally, the ten year records have nearly 140,000 lake polygons. To improve work efficiency, obtain the accurate boundary of the glacial lakes, and meanwhile, minimize the subjective judgment errors of the operatives, we applied a systematic glacial lake detection method that combined two steps from initial glacial lake extraction and subsequently manual refinement of lake mapping results.

The initial glacial lake extraction using GEE can make sure that approximately half of all the lakes are automatically extracted. To make annual data more complete and accurate, manual inspection and refinement of individual glacial lake is necessary to supplement some missing lakes and correct the mapping errors such as mountain shadows and river segments. The glacial lake vector over the entire HMA for the years from 2008 to 2017 has been rechecked and reedited individually through dynamic cross-validation by ten trained experts. Besides, the related attribution (i.e., lake type, elevation, distance to the nearest glacier terminus, area and perimeter) were manually added for each lake. It should be noted that the type of each individual glacial lake was carefully judged according to the different formation mechanisms or growth stages. Thus the whole processing is time consuming and require a considerable amount of work. We are terrible sorry for the large discrepancy compared with dataset from

Wang et al., 2020, which may contribute to the difference in the calculated statistics, and the conclusions drawn in some sub-regions.

Therefore, following the Reviewer's very pertinent recommendations, we have carefully examined the lake data for the ten year by supplementing glacial lakes that have an area of greater than 0.04 km². The area of 0.04 km² is chosen as the threshold for re-revision of the glacial lake dataset for three reasons: **i)** glacial lakes with small sizes are more likely to be confused with surroundings due to its less effective spectral, textural and spatial information in comparison with those of relatively large glacial lakes, which results in the **large uncertainty of area** (Please see the Section 4, small glacial lakes with the area of less than 0.04km² have the mean area uncertainty of about 25.7 %). **ii)** small glacial lakes are highly variable in their locations, shapes and size, and also the optimal images with valid observations over the potential glacial lake area is very limited **in each year**. The image dates are not consistent across the whole HMA region because of atmospheric disturbances, also the influences from image quality, ice and shadow that obscured the lakes, this creates a great deal of uncertainty about the number and extent of small glacial lakes **for the such an annually resolved inventory**. **iii)** given the amount of work required to update the inventory and also revise our manuscript before the deadline.

Moreover, as for the definition of the geographical location of the study area, generally, the term HMA refers to a **broad high-altitude region in South and Central Asia** that covers the whole Tibetan Plateau and adjacent mountain ranges, including the Eastern Hindu Kush, Western Himalaya, Eastern Himalaya, Central Himalaya, Karakoram, Western Pamir, Pamir Alay, Northern/Western Tien Shan, Dzhungarsky Alatau, Western Kunlun Shan, Nyainqentanglha, Gangdise Mountains, Hengduan Shan, Tibetan Interior Mountains, Tanggula Shan, Eastern Tibetan Mountains, Qilian Shan, Eastern Kunlun Shan, Altun Shan, Eastern Tien Shan, Central Tien Shan, Eastern Pamir **(Yoon et al., 2019;Brun et al., 2017a;Zhao et al., 2014)**. It extends from 26°N to 45°N and from 67°E to 105°E, the altitude of the plateau is about 4500 m on average.

Until now, there are still not a uniform standard about the spatial location of HMA region. When we first started working on the glacial lake mapping in the HMA region and required accurate mountain ranges division data, Tobias Bolch kindly provided us the mountain boundary shapefile in High Mountain Asia, which excluded Altay and Sayan mountains, can be downloaded from the website geo.uzh.ch/~tbolch/data/regions_hma_v03.zip. Just as the Reviewer pointed out that, the generalized concept of HMA could contain the Altay and Sayan mountains, and a comprehensive delineation of glacial lakes in these regions over the ten year periods is needed urgently in our future work.

Our team have been working hardly to update data of glacial lakes larger than 0.04 km² in the ten year, and add their attribute information. Correspondingly, the related statistics, figures and analysis in the article have also been modified based on the new lake inventory. We tried our best to improve the data and manuscript, and hope that the revised manuscript will meet with approval. As an annual time steps data over the HMA region from 2008 to 2017, the produced data may not be complete and perfect enough, yet our team will continuously update and share more and better glacial lake data in the future.

Finally, to provide more information about our study area, in the **Section 2.1 Study area** of the revised manuscript, we have rewritten and expanded this chapter by adding the location map of the HMA and comprehensive descriptions about the climate and topography. Please see the revised manuscript for more details.

**2) We have comprehensively investigated the existing works about glacial lake inventory, and conducted deep comparisons between our Hi-MAG dataset and glacial lake inventory from Wang et al., 2020, and other studies from Nie et al., 2017;Zhang et al., 2015;Pekel et al., 2016a.**

To clearly show benefits and challenges remaining in this study, we firstly comprehensively investigated the existing works about glacial lake inventory.

Many previously published researches have devoted to the glacial lakes mapping with remotely sensed data over the different regions of HMA. Some works mainly focus on the investigation of the development of relatively large glacial lakes. Rounce et al. identified 131 glacial lakes in Nepal in 2015 that are greater than 0.1 $km^2$ (Rounce et al., 2017). Li et al. compiled an inventory of glacial lakes (≥0.01 $km^2$) with a spatial resolution of 30 m in the Karakoram mountains (Li et al., 2020). Aggarwal et al. shared a new dataset of glacial and high-altitude lakes that have an area > 0.01 $km^2$ for Sikkim, Eastern Himalaya from 1972–2015 (Aggarwal et al., 2017). Ukita et al. constructed a glacial lake inventory of Bhutan in the Himalaya from the period 2006-2010 based on high-resolution PRISM and AVNIR-2 data from ALOS. Considering small lakes present less of a GLOF risk. They set 0.01 $km^2$ as the minimum lake size (Ukita et al., 2011). Ashraf et al. used Landsat-7 ETM+ images for the 2000-2001 period to delineate glacial lakes greater than 0.02 $km^2$ in the Hindukush-Karakoram-Himalaya (HKH) Region of Pakistan (Ashraf et al., 2012). Because small glacial lakes experience highly variable in their shape, location, and occurrence, and were clearly sensitive to the warming climate and glacier wastage, a growing number of scholars have paied attention to the abundance of small glacial lakes. Salerno et al. provided a complete mapping of glacial lakes (including lake size less than 0.001 $km^2$) and debris-covered glaciers with 10-m spatial resolution in the Mount Everest region in 2008 (Salerno et al., 2012). Wang et al. utilized Landsat TM/ETM+ images for the years 1990, 2000 and 2010 to map glacial lakes with area more than 0.002 $km^2$ in Tien Shan Mountains (Wang et al., 2013a). Luo et al. examined glacial lake changes (lake area >0.0036 $km^2$) for the entire western Nyainqentanglha range for the five periods between 1976 and 2018 using multi-temporal Landsat images (Luo et al., 2020). International Centre for Integrated Mountain Development (ICIMOD) provided comprehensive information about the glacial lakes (greater than or equal to 0.003 $km^2$) of five major river basins of the Hindu Kush Himalaya (HKH) using Landsat images for the year 2005 (Sudan et al., 2018). Nie et al. mapped the distribution of glacial lakes across the entire Himalaya in the year of 2015 using a total of 348 Landsat images at 30 m resolution. They set the minimum mapping unit to 0.0081 $km^2$ (Nie et al., 2017). Zhang et al. presented a database of glacial lakes larger than 0.003 $km^2$ in the Third Pole for the years ~1990, 2000, and 2010 (Zhang et al., 2015). All these researches greatly help to fill the data gap of glacial lakes information in the HMA region. At the global scale, Pekel et al. used millions of Landsat satellite images to record global surface water over the past 32 years at 30-metre resolution (Pekel et al., 2016b), many large and visible glacial lakes were also included. More recently, Shugar et al. mapped glacial lakes with areas >0.05 $km^2$ around the world using 254,795 satellite images from 1990 to 2018 (Shugar et al., 2020). Wang et al. developed a glacial lake inventory (with size of larger than 0.0054 $km^2$) across the High Mountain Asia at two time periods (1990 and 2018) using manual mapping on 30 m Landsat images (Wang et al., 2020b). They firstly introduce glacial lake inventory at such a large-scale, the data shared will be served as a baseline for the further studies related to water resource assessment or glacier hazard risk.

For more rigorous cross-validation and assessment of the data quality, in the **Section 6.1** of revised manuscript, we have conducted a deep analysis and comparison between our Hi-MAG

dataset and glacial lake inventory from Wang et al., 2020, who mapped glacial lake larger than 0.0054 km$^2$ in a much larger High Mountain Asia region in two time periods (1990 and 2018), and made their data public recently. The comparative discussions with other studies (e.g. Nie et al., 2017;Zhang et al., 2015;Pekel et al., 2016a) were also used as a general reference.

**Firstly, we compared our dataset with that of Wang et al., 2020 for the closest period (2017 from our Hi-MAG database and 2018 from Wang et al. 2020) over the spatial extent of our HMA region.** It should be noted that in the last version of database, some huge glacial lakes with an area of larger than 6.5km$^2$ have also been mapped in the Hi-MAG dataset, this is because these lakes all located in the range of 10 km from the nearest glacier terminus according to the definition of glacial lake in HMA. However, in the dataset produced by Wang et al., 2020, these huge lakes were removed. For the sake of a reliable comparative analysis between different studies under the equal conditions, glacial lakes with an area of larger than 6.5km$^2$ were deleted in our new lake inventory. Besides, **serval hundreds of glacial lakes in the dataset from Wang et al. are located outside of 10km buffer to the glacier terminus, which were also excluded for the comparison**.

The differences in the total number and area between these two dataset are 6206 and 223.97 km$^2$, respectively. **We also found that 2077 glacial lakes with a total area of 178.77km$^2$ in our Hi-MAG dataset were not detected by Wang et al.** As for the lakes we have not mapped, 96.1% of the glacial lakes are smaller than 0.04km$^2$, which means that 3.9 % (323 glacial lakes) of the difference in the total number is composed by the glacial lakes larger than 0.04 km$^2$. The main reasons for these missed 323 glacial lakes in our dataset are because the interference of some bad observations (cloud or snow), glacial lake dried up or outburst, or located in the middle of the river, which are summarized and explained in the following:

i) The limited or lack of high-quality images in a whole year. Many glacial lakes were always covered by cloud or snow, and no effective observation data are available, as shown in Figure I.

[Figure]

Figure I. Some glacial lakes that are always covered by cloud (first row) or snow (second row) in the true color composites (Bands:7, 4, 3) of the Landsat 8 OLI images for the year of 2017. The red contours refer to the 2018 glacial lake outlines digitized by Wang et al.

ii) Many glacial lakes dried up or outburst, and thus vanished in the image of 2017 (Figure II).

[Figure]

Figure II. Some glacial lakes dried up or outburst in 2017. Background images are the true color composites (Bands:7, 4, 3) of the Landsat 8 OLI images for the year of 2017. The red contours refer to the 2018 glacial lake outlines digitized by Wang et al.

iii) Lakes or ponds located in the middle of the river, which were not judged to be glacial lakes in our Hi-MAG database (Figure III).

[Figure]

Figure III. Lakes or ponds located in the middle of the river, and were not judged to be glacial lakes in our inventory. Background images are the true color composites (Bands:7, 4, 3) of the Landsat 8 OLI images for the year of 2017. The red contours refer to the 2018 glacial lake outlines digitized by Wang et al.

To test the spatial correlation of glacial lakes distribution in two datasets, we compared the statistics in glacial lake number and area aggregated on a 0.1°×0.1° grid for the HMA regions. The results for the total glacial lake area, areas for glacial lakes larger than 0.04km$^2$, and number for glacial lakes larger than 0.04km$^2$ are depicted in Figure IV. A clear and strong correlation can be observed for all the statistics between our Hi-MAG dataset and glacial lake data by Wang et al. Most of the points being distributed around the 1:1 line, which shows that there is great consistency in the results.

[Figure]

Figure IV. Comparison the results of (a) total glacial lake area, (b) areas for glacial lakes larger than 0.04km$^2$, and (c) number for glacial lakes larger than 0.04km$^2$ summed over a 0.1°×0.1° grid between Hi-MAG database and inventory by Wang et al., 2020.

In order to quantitatively and systematically evaluate the accuracy of our product, we implemented a stratified random sampling (Song et al., 2017b;Stehman and SV, 2012), where the glacial lakes were divided into four strata. The sample sizes are the spatial resolution (30 m) of the data, and the strata are designed as: C0W0. both the results are non glacial lakes;

C0W1. non glacial lake for Chen's and glacial lake for Wang's; C1W0. glacial lake for Chen's and non glacial lake for Wang's; C1W1. Both the results are glacial lakes.

A total of 4,000 points were randomly selected, as shown in Figure V. The sample number for C1W1 and C1W0 are 1300 and 700, respectively, which is almost the same ratio between the total areas for the two strata (1450.50 km$^2$ vs 732.77 km$^2$). Because of the approximate total area with C1W0, we also randomly selected 700 samples from stratum C0W1. The rest 1300 samples are from C0W0.

[Figure]

Figure V. Distribution of validation samples selected using stratified random sampling. Blue polygons are glacier outlines taken from the Randolph Glacier Inventory (RGI v5.0). Yellow polygons refer to buffer area within 10 km of glacier terminals.

Every validation sample was visually examined using Landsat imagery and higher-resolution imagery in Google Earth. Sample pixels were interpreted by a regional glacial lake mapping expert, and ambiguous samples were cross-validated by a second observer. If a sample is difficult to interpret, it was marked as ambiguous sample and excluded for the accuracy assessment. The sample number estimates were produced for each of the four strata (Table I), and these strata totals were then summed to obtain the total accuracy.

For the 1300 pixel samples that were considered to be non glacial lake by both datasets, after the pixel by pixel verification, 1215 were indeed non-glacial lakes, while 37 were the missed glacial lakes. In contrary, 1260 out of 1300 pixels belongs to the class of glacial lake, and 25 pixels were misclassified as glacial lake by the two inventories. 307 error pixels were found in the results from Wang et al., constitute about half of the total validation number. For the glacial lakes identified only by our inventory, 678 out of 700 were corrected classified. Our results yielded high overall classification accuracy (88%), user's accuracy (97%), and producer's accuracy (82%) for glacial lake classification using Landsat data, which further confirm the improved quality of the Hi-MAG database.

Table I. Statistical results of stratified random sampling.

| Strata | Total pixel number | Total area (km²) | Sample number | Sample No. of non glacial lake | Sample No. of glacial lake | No. of ambiguous sample |
|---|---|---|---|---|---|---|

| | | | | | | |
|---|---|---|---|---|---|---|
| C0W0 | 2,022,448,650 | 1,820,203.78 | 1300 | 1215 | 37 | 48 |
| C0W1 | 925,449 | 832.90 | 700 | 307 | 362 | 31 |
| C1W0 | 814,196 | 732.77 | 700 | 21 | 678 | 1 |
| C1W1 | 1,611,668 | 1,450.50 | 1300 | 25 | 1260 | 15 |

**Based on the new version of glacial lake inventory, we also made comparisons of the statistical results between our Hi-MAG database and other studies from Nie et al., 2017;Zhang et al., 2015;Pekel et al., 2016a.** It should be noted that these inventory data have not been released publicly yet, except for GSW dataset produced by Pekel et al. Pekel et al. used millions of Landsat satellite images to map global surface water, many transient melt ponds and streams on the debris-covered glaciers were also identified here as permanent water surfaces, meanwhile, large quantity of small glacial lakes were missing. For the sake of a reliable comparative analysis, glacial lakes in the GSW were further extracted using the range of glacier buffer (10 km). We therefore conducted a rough comparisons with statistical results from Nie et al., 2017;Zhang et al., 2015, and qualitative comparison of the lake extent between GSW dataset and our Hi-MAG database summed by mountain range in 2015.

Hi-MAG lake number was 7268 higher and area was 644.26 km$^2$ higher than the estimation for the Tibetan Plateau (Zhang et al., 2015). The largest discrepancy is in the Gangdise, Himalaya and Nyainqentanglha Mountains in 2010. Across the Himalaya, we found 476.09 km$^2$ of glacial lakes, 4.57% more than previous estimates in 2015 (Nie et al., 2017). In addition, we qualitatively compared the lake extent between publicly available high-resolution Global Surface Water (GSW) dataset and our Hi-MAG database summed by mountain range in 2015. GSW data can be accessed at https://global-surface-water.appspot.com/download. For the sake of a reliable comparative analysis, lake polygons in the Hi-MAG dataset were converted into the grid format, and glacial lakes in the GSW were further extracted using the range of glacier buffer (10 km). Hi-MAG detected more glacial lakes in the Himalaya, Eastern Hindu Kush, and Tien Shan, and fewer in Eastern Pamir and Western Kunlun Shan. Fig. A5 illustrates the differences between our Hi-MAG glacial lake results and GSW-derived lake area for the whole HMA region.

The glacial lake area observed in our lake dataset in the Eastern Pamir and Western Kunlun Mountains does not conform to the mapped surface water in the GSW for these sub-regions. While there are numerous glacial lakes from an open water perspective, actually part of them are river segments. Additionally, the Himalaya, Eastern Hindu Kush, and some other Tien Shan host thousands of glacial lakes that are not readily observable in the GSW product. Large discrepancies in mountainous glacial lake estimates preclude a significant consistency between GSW and our Hi-MAG lake data over the HMA region. The region with the highest consistency between GSW and Hi-MAG product is interior Tibet. There is little agreement for Tien Shan, where the weather is rainy and snowy in the region above 3000 m, and large amounts of ancient glacial deposits have been accumulated. Here glacial lakes are featured by small size, due to the influence of source glaciers and lake beds, as well as the water depth and sediment inflow, glacial lakes appear heterogeneous reflectance in the image. Errors could exist in datasets produced by automated classification, but we also did a detailed manual editing, so we were not relying exclusively on automatic classification. Karakoram regions seem to have fewer glacial lakes in our estimate, owing to the overestimation of surface water on debris covered glaciers in the GSW dataset.

Finally, we have added the related content in the **Section 1. Introduction**, to give a comprehensive introduction about the existing glacial lake inventories. In the **Section 6.1** of revised manuscript, we have conducted a deep analysis and comparison between our Hi-MAG

**3) A thorough and quantitative uncertainty analysis of lake area was added in the Section 4 of revised manuscript, the error bars for the lake area, and confidence intervals for the estimated trends were also added throughout the paper.**

A thorough and quantitative uncertainty analysis of lake area was added in the Section 4 of revised manuscript. In order to highlight differences in the tails of area uncertainties in each year, we have added a new figure (Fig. 4a) that plotted the distributions of uncertainties from 2008 to 2017. Fig. 4b was also added for the better visualization of the relationship of area uncertainties against areas of all the glacial lakes in HMA (taking the results in 2017 as an example).

Assuming an uncertainty of 1 pixel for the detected glacial lake boundaries, we calculated the systematic errors for the whole HMA region (Fig. 4). For the year between 2008 and 2017, the area uncertainty of each glacial lake generally ranged from 0.30% to 50%, with the mean value falling around the 17%, and standard deviation around 11% (Fig. 4a). The maximum and mean value of area uncertainty for the glacial lakes in 2010 are the lowest, while for the year of 2016, the corresponding statistics are highest, this can be attributed to the different factors. The maximum of area uncertainty of glacial lakes is related with the shape and size of a certain lake (as can be seen from equation (1)), but its mean value is equal to the sum of the area uncertainties of each glacial lake divided by the total number, which depend on the total number of glacial lakes in a year, and also the shape and area of each lake. Besides, a close relationship can be found between the area uncertainties and the sizes of the glacial lakes (Fig. 4b). Most of the large glacial lakes (area $\geq 0.04 km^2$) have the mean area uncertainty of about 7%. This systematic errors were more significant for the small-sized glacial lakes. We measured glacial lake down to 0.0081 $km^2$ (nine pixels in Landsat imagery), where systematic errors calculated by equation (1) were ~50%.

[Figure]

**Fig. 4.** (a) Statistics of area uncertainty (%) of glacial lakes for the years from 2008 to 2017. (b) Relationship between area uncertainties and areas of all the glacial lakes in HMA in 2017.

Besides, given the mapping uncertainty of glacial lakes and thus for the further improvement of reliability of calculated statistics, the error bars for the lake area, and confidence intervals for the estimated trends, and measure of uncertainty for the annual changes were also added throughout the paper.

**4) Detailed descriptions about the key procedures for glacial lake mapping were provided in the Section 3.2, such as advantages of automated mapping, and the amount of work required for the manual editing.**

For the development of HMA Glacial Lake Inventory (Hi-MAG) database, we applied a systematic glacial lake detection method that combined two steps from initial glacial lake extraction and subsequently manual refinement of lake mapping results. As for the automatic classification using GEE, there are four main procedures including (i) the clipped Landsat images by the extent of the glacier buffers and assembled into a time-series dataset; (ii) identified and masked some poor quality observations such as cloud, cloud shadow, topographic shadow; (iii) calculated MNDWI; and (iv) extracted the potential glacial lake areas by applying adaptive MNDWI threshold. Based on the automated processing, nearly 60% glacial lakes in each year can be correctly classified, of the other lakes that were not properly classified, 30% were missed and 10% were misclassified (mainly mountain shadows and river segments). For such a large-scale area that characterized by various and complex climatic, geological and terrain conditions, this classification method is simple but effective, the results are also reasonable since it provides very low commission errors. As shown in Figure VI, glacial lake outlines extracted using the automated classification method in our study fit the real boundary of the glacial lake very well, while manually delineated glacial lake outlines are largely influenced by people's subjective experience and manual operation, resulting in overestimation for most part of a glacial lake region, and underestimation for a few areas. Moreover, results from manual digitization show poor performance for the delineation of the some glacial lakes with complex curved shapes.

[Figure]

☐ Manually delineated lake outlines from Wang et al. 2020
☐ Automatically extracted glacial lake outlines

Figure VI. Examples of extraction of glacial lake outlines using the automated classification method in our study and manual digitization from Wang et al. 2020.

The estimated 30% errors in the initially classified map were from the missing small glacial lakes. Glacial lakes with small sizes are more likely to be confused with surroundings due to its less effective spectral, textural and spatial information in comparison with those of relatively large glacial lakes. Because the automated method is mainly based on the spectral features for the glacial lake mapping over the large mountainous area, the spectral information provided by the small glacial lakes will be incomprehensive and insufficient for the accurate detection of this kind of glacial lakes under various and complex environmental conditions. Therefore, visual interpretation and manual editing is still an effective way to ensure the high accuracy of lake inventories. In the updated version of dataset, we have carefully examined the lake data for the ten year by supplementing glacial lakes that have an area of greater than 0.04 km$^2$, and

add their attribute information. We tried our best to improve the quality of glacial lake dataset, and make sure it is greatly optimized.

   Finally, in the **Section 3.2 Adaptive glacial lake mapping method** of revised manuscript, we have added some text to give detailed explanations about the whole mapping procedures. The amount of work required for the manual editing were also described clearly. Please see the revised manuscript for more details.

2. *The current attribute table of this inventory is too sample, that it even did not give an ID for each lake. Glacial lakes should be indexed with unique ID that could be used to connect with RGI or GLIMS glacier inventory dataset. In addition, the abbreviations used in the dataset (PGL, UCL and SGL) were totally not mentioned in the main text of the paper, we don't know what they meant.*

Response:

We are terrible sorry for the incomplete attribute information of our inventory. We have carefully modified the attribute table of the new version of inventory by adding the unique ID for each glacial lake (attribute item GL_ID), for example, GL075720E40943N that formed by 'GL'+ 'longitude of centroid' + 'latitude of centroid', and retain three decimal places. Besides, the abbreviations of glacial lake type including PGL, UCL, SGL and IGL were all replaced by their full name, i.e. proglacial lake, unconnected glacial lake, supraglacial lake and ice-marginal lake.

3. *For a dataset paper, it should avoid including any further analysis on the data (for example, the inter‐annual variability of lake area presented in the section 5.2), especially when their results exist large uncertainty or errors.*

Response:

We greatly appreciate the suggestion of the Reviewer. According to the response to your Comment 1, the dataset was updated by supplementing glacial lakes that have an area of greater than 0.04 $km^2$ for the ten year. As for the estimation of area error, most of the large glacial lakes (area $\geqslant$ 0.04$km^2$) have the mean relative error of about 7%. We also measured glacial lake down to 0.0081 $km^2$ (nine pixels in Landsat imagery), where relative errors calculated were ~50%. It can be found that this systematic error was more significant for the small-sized glacial lakes. Given the large uncertainty in the area and number of small glacial lakes, we should avoid including any further analysis on the data, especially for small glacial lakes that have large uncertainty. Therefore, we have carefully modified the whole manuscript to make sure the statistics and related analyses all focus on large-sized and all sizes of glacial lakes over the whole HMA region or different mountain ranges to discuss spatial and temporal variability for the 10-year record, and meantime, to avoid analysing and explaining the trends and number for small-sized glacial lakes. Fig. 5 shows the inter-annual variations in the number of small glacial lakes for different HMA mountain regions, we have removed the Section 5.2 in the revised manuscript.

[revised manuscript text omitted]

Chen, F., Zhang, M., Guo, H., Allen, S., Kargel, J., Haritashya, U., Watson, S.: Annual 30-meter Dataset for Glacial Lakes in High Mountain Asia from 2008 to 2017 (Hi-MAG) [Dataset], Zenodo, https://doi.org/10.5281/zenodo.4059181.

715

---

## Author Response (AR2)

**Response to Editor's comments to manuscript essd-2020-57**

**"Annual 30-meter Dataset for Glacial Lakes in High Mountain Asia from 2008 to 2017"**

**Dear Editor:**

Thank you a lot for your kind and careful reviewing. Your suggestions give us important and constructive perspective on this manuscript, and help to improve the manuscript greatly. We have fully considered all the comments of you, and have substantially revised our manuscript according to your comments. A point-by-point response to the outstanding comments raised is attached to this manuscript. The major changes are summarized as follows:

1. We have been looking for a native speaker and a special Polish Company: Charlesworth to modify the paper, and to reduce the language and grammar errors as possible.

2. The full term of every acronym has been introduced the first time it was used in the manuscript.

3. We have modified the data description document by adding the content about data sources and methods, spelling out all the abbreviations of the table caption, and adding the ORCID details of all the authors.

4. The CRS of all the prj files of our dataset have been transformed into a standard projection Asia_North_Albers_Equal_Area_Conic (ESRI: 102025) according to the spatialreference.org.

The changes can be tracked in the revised manuscript. In the following, we provide point by point responses to the outstanding comments and suggestions provided by the Topical editor. We are indebted to you for your outstanding and constructive comments, which greatly helped us to improve the technical quality and presentation of our manuscript. Once again, thank you very much for your comments and suggestions.

**Response to Comments by Topical editor:**

*1. The published time series on Glacial Lakes in High Mountain Asia is of importance. The manuscript, the data description and data publication do not yet fulfill the requirements of ESSD and improvements are needed. A major revision of the manuscript and a minor revision of the dataset is needed.*

*First, the manuscript requires editing by a professional English language editing service. SRTM, SLC, GAMDAM and many more acronyms are not introduced. Introduce every acronym before using it in the text. The first time you use the term, put the acronym in parentheses after the full term. Please upload the revised version of your manuscript with track changes included.*

Response:

We greatly appreciate the suggestions of the Topical editor for his/her accurate summary of the main contributions of our work and for the outstanding recommendations provided. Following the Topical editor's very pertinent recommendations, in the revised manuscript and the following text we have made modifications according to these questions. We thank again the Topical editor for his/her careful handling of our manuscript and for the constructive suggestions provided, which greatly helped us improve the technical quality and presentation of our manuscript.

We have been looking for a native speaker and a special Polish Company: Charlesworth to modify the paper, and to reduce the language and grammar errors as possible. Then we have corrected all the acronyms by introducing the full term first and put the acronym in parentheses. Besides, the revised version of manuscript is uploaded with track changes included.

2. *Please also optimize the data set abstract in Zenodo, please include data sources and methods in the abstract.*

   *Please also include data sources and methods in the data description document. Please add the overview on the dBASE table in the overview text, please spell out abbreviations, e.g. GL_type in the table caption. Please add your ORCID in the data description document.*

Response:

Thanks for these good suggestions. Firstly, we have optimized the data set abstract in Zenodo by adding the content about data sources and methods. The detailed modifications are as follows:

We developed a High Mountain Asia (HMA) Glacial Lake Inventory (Hi-MAG) database to characterize the annual coverage of glacial lakes from 2008 to 2017 at 30 m resolution. For the development of the Hi-MAG database, a total of 40,481 satellite images including Landsat 5 TM, Landsat 7 ETM+ and Landsat 8 OLI were used, and a systematic glacial lake detection method that comprised the automated processing using GEE and subsequent manual refinement of these lake mapping results were applied. This is the first glacial lake inventory across the HMA with annual temporal resolution, it can provide details for different types of glacial lakes and evolution patterns. It can be used for studies of the complex interactions between glaciers, climate, and glacial lakes, and GLOFs, potential downstream risks, and water resources.

Then the information about the data sources and methods have also been added in **the Section of Brief introduction** of the data description document.

Finally, we have modified the data description document by adding the overview about the dBASE table, spelling out all the abbreviations of the table caption, and adding the ORCID details of all the authors. The detailed modifications are as follows:

2) Hi-MAG database.zip: it contains 50 files in total for the ten years. Each phase consists of five files, including *.shp (the main file that stores the feature geometry), *.shx (the index file that stores the index of the feature geometry), *.dbf (the dBASE table that stores the attribute information of features, including **GL_ID**: the coding of glacial lake, **GL_type**: the type of glacial lake, **GL_Area**: the area of glacial lake ($m^2$), **GL_Year**: the year of mapped glacial lake, **GL_Elev**: the elevation of glacial lake (m), **GL_SubR**: the sub-region of glacial lake in the HMA, **GL_Peri**: the perimeter of glacial lake (m),

**Distance**: the euclidian distance from glacial lake to the nearest glacier terminus (m)), *cpg (character encoding file, describes a set of characters for displaying text in shapefiles; helps localize maps for specific languages), and *.prj (the file that stores the coordinate system information). In each shape file, the polygons are boundaries of each glacial lake.

*Authors:*

Fang Chen (chenfang@radi.ac.cn) https://orcid.org/0000-0002-3245-2584

Meimei Zhang (zhangmm@radi.ac.cn) https://orcid.org/0000-0001-9621-4879

Huadong Guo (hdguo@radi.ac.cn) https://orcid.org/0000-0003-0337-1862

Simon Allen (simon.allen@geo.uzh.ch) https://orcid.org/0000-0002-4809-649X

Jeffrey S. Kargel (jeffreyskargel@hotmail.com) https://orcid.org/0000-0002-5506-1797

Umesh K. Haritashya (uharitashya1@udayton.edu) https://orcid.org/0000-0001-9527-954X

C. Scott Watson (cswatson@email.arizona.edu) https://orcid.org/0000-0003-2656-961X

3. *ESSD policy requires the re-usability of the published data sets using open source: Please list open-source GIS software in the list of software. Using open source GIS software, e.g. QGIS, the CRS of Glacial Lakes in High Mountain Asia is not readily recognized, eventually because it is no standard projection. Could you check back with spatialreference.org, eventually change your prj file.*

   *e.g.https://essd.copernicus.org/articles/10/2275/2018/essd-10-2275-2018.pdf Carlson and Oda put the example of QGIS-compatible shapefiles.*

Response:

We gratefully thank the Topical editor for his/her comments and suggestions. We have listed the open-source GIS software in the **Section 6. Recommended software** of the data description document. The detailed modifications are as follows:

**6. Recommended software**

All the vector (shape) files of this dataset can be easily opened and processed with QGIS, gvSIG or other open-source remote sensing and geographic information system software.

   Besides, the CRS of our last version of dataset is not recognized by QGIS software indeed, which is due to the fact that it is a custom coordinate system, rather than a standard projection. Therefore, following the Topical editor's very pertinent recommendations, we have carefully changed all the prj files by transforming their CRS to standard Asia_North_Albers_Equal_Area_Conic (ESRI: 102025) according to the spatialreference.org.

[revised manuscript text omitted]